# Cascaded Flow Matching for Heterogeneous Tabular Data with Mixed-Type Features

Markus Mueller [1]   Kathrin Gruber [1]   Dennis Fok [1]

## Abstract

Advances in generative modeling have recently been adapted to tabular data containing discrete and continuous features. However, generating mixed-type features that combine discrete states with an otherwise continuous distribution in a single feature remains challenging. We advance the state-of-the-art in diffusion models for tabular data with a cascaded approach. We first generate a low-resolution version of a tabular data row, that is, the collection of the purely categorical features and a coarse categorical representation of numerical features. Next, this information is leveraged in the high-resolution flow matching model via a novel guided conditional probability path and data-dependent coupling. The low-resolution representation of numerical features explicitly accounts for discrete outcomes, such as missing or inflated values, and therewith enables a more faithful generation of mixed-type features. We formally prove that this cascade tightens the transport cost bound. The results indicate that our model generates significantly more realistic samples and captures distributional details more accurately, for example, the detection score improves by 51.9%. Code is available at https://github.com/muellermarkus/tabcascade.

## 1. Introduction

Advancements in the field of generative modeling – rooted in seminal contributions on diffusion models (Sohl-Dickstein et al., 2015; Ho et al., 2020), score-based modeling (Song et al., 2021) and flow matching (Albergo & Vanden-Eijnden, 2023; Lipman et al., 2023; Liu et al., 2023) – have yielded state-of-the-art results for high-dimensional modalities that admit a homogeneous underlying representation such as images, audio, and text. Diffusion-based models for heterogeneous tabular data generation, i.e., datasets with categorical, continuous or mixed categorical-continuous features in each sample, (Kim et al., 2023; Kotelnikov et al., 2023; Zhang et al., 2024b; Lee et al., 2023; Mueller et al., 2025; Shi et al., 2025) largely inherit this design choice of a shared generative objective across feature types. However, categorical and continuous feature types rest on different structural assumptions, such as discrete versus continuous support, probability mass versus density formulations, differing perturbation or noise models, and therefore require distinct representations and generative mechanisms. Combining distinct feature types under a unified training objective leads to implicit feature reweighting such that some features dominate learning. The intermediate case of *mixed-type* features, i.e., features whose marginal distributions combine discrete point masses with continuous densities, lacks a dedicated representational and generative treatment, which degrades the realism of the joint distribution.

In this paper, we propose TabCascade, a novel cascaded flow matching framework for heterogeneous tabular data. Within this cascaded structure, numerical details are generated conditional on a coarse-grained representation of the high-fidelity data. Accordingly, we conceptualize categorical and numerical features as low- and high-resolution representations of a tabular data row. We explore discretization methods such as distributional regression trees and Gaussian mixture models to construct a categorical, i.e., low-resolution, approximation of the numerical features. TabCascade first learns the low-resolution joint distribution of categorical and discretized numerical data. Then, TabCascade generates numerical data, i.e., the high-resolution signal, conditionally on the low-resolution model's output. In this second step, TabCascade focuses its capacity on where it is most needed: generating details, as opposed to coarse categorical data, which we show is relatively easy to learn. We design the high-resolution model from a conditional probability path guided by low-resolution information, thereby introducing a data-dependent coupling that reduces the transport costs between source and target distributions of high-resolution data. Further, we allow for the paths to be non-linear by utilizing learnable time schedules conditioned on low-resolution

---

[1]Econometric Institute, Erasmus University Rotterdam, Rotterdam, The Netherlands. Correspondence to: Markus Mueller <mueller@ese.eur.nl>.

*Proceedings of the 43rd International Conference on Machine Learning*, Seoul, South Korea. PMLR 306, 2026. Copyright 2026 by the author(s).

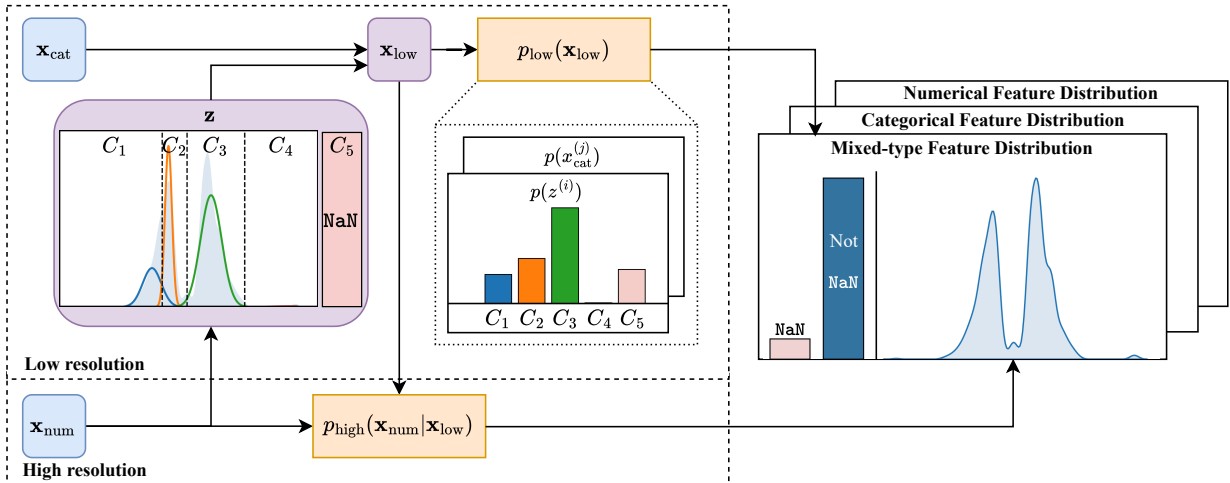

*Figure 1.* Overview of TabCascade for the missing value generation task. We derive a categorical, low-resolution representation $\mathbf{z}$ from $\mathbf{x}_{\text{num}}$, form $\mathbf{x}_{\text{low}} = (\mathbf{x}_{\text{cat}}, \mathbf{z})$ and then learn $p_{\text{low}}(\mathbf{x}_{\text{low}})$. We then learn the high-resolution distribution $p_{\text{high}}(\mathbf{x}_{\text{num}}|\mathbf{x}_{\text{low}})$ conditional on $\mathbf{x}_{\text{low}}$. This reduces the transport cost bound and simplifies the learning task. The discrete state $\mathbf{z}$ enables the model to naturally handle mixed-type feature distributions at generation time. This approach generalizes to arbitrary (and multiple) discrete states.

information. We choose the categorical part of CDTD (Mueller et al., 2025) as our low-resolution component.

The cascaded formulation also provides a natural mechanism to model *mixed-type* features (Li et al., 2025). Among others, such features arise in censored, zero- or one-inflated, and missing-value–augmented variables, where these different discrete outcomes can carry meaningful information (Little & Rubin, 1987). Realistic synthesis therefore requires the model to generate discrete states (including missingness) as part of the data-generating process. By separating coarse discrete structure from continuous refinement, TabCascade directly accommodates these mixed-type features within a unified generative process. Our results show that this substantially benefits the realism of the generated samples and that TabCascade learns the details of the distributions much more accurately than the current state-of-the-art methods.

In sum, our key contributions are:

- To the best of our knowledge, we propose the first cascaded diffusion model for tabular data as well as the first diffusion model to address mixed-type feature generation.

- We decompose the generation task into low- and high-resolution parts and propose a novel cascaded flow matching framework. We design a guided conditional probability path to model the high-resolution details.

- The use of feature-type–tailored models sidesteps the challenge of balancing type-specific losses, thereby preventing the unintended weighting of features during training that is prevalent in previous work.

- We demonstrate state-of-the-art results: the detection score improves by over 50%, the Wasserstein distance by 50% and the machine learning efficiency by 30%.

## 2. Related Work

**Diffusion models for tabular data.** The main challenge for tabular data generation is the effective integration of heterogeneous (i.e., numerical and categorical) feature sets. TabDDPM (Kotelnikov et al., 2023) and CoDi (Lee et al., 2023) combine multinomial diffusion (Hoogeboom et al., 2021) with DDPM (Sohl-Dickstein et al., 2015; Ho et al., 2020); STaSY (Kim et al., 2023) treats one-hot encoded categorical data as numerical; and TabSyn (Zhang et al., 2024b) adopts latent diffusion to embed both feature types into a continuous space. Despite its popularity in other domains, latent diffusion has proven less effective for heterogeneous tabular data compared to models defined directly in data space (Mueller et al., 2025). More recent models, such as TabDiff (Shi et al., 2025) and CDTD (Mueller et al., 2025) learn noise schedules alongside the diffusion model to accommodate the feature heterogeneity in tabular data. These models integrate score matching (Song et al., 2021; Karras et al., 2022) with either masked diffusion (Sahoo et al., 2024a) or score interpolation (Dieleman et al., 2022), respectively. While most of these models can be easily adapted to be *trainable* on data containing missing values, in their original state none of them can *generate* missing values in numerical features.

**Exploitation of low-resolution information.** Outside the domain of tabular data generation, several approaches exist to leverage low-resolution information. Cascaded diffusion models (Ho et al., 2022) for super-resolution images define a sequence of diffusion models, where higher resolution models are conditioned on the lower resolution model's outputs. This divide-and-conquer strategy has been successfully used in Google's Imagen model (Saharia et al., 2022)

for the generation of high-fidelity images, and can be further refined with data-dependent couplings (Albergo et al., 2024). Similarly, Tang et al. (2024) improve sample quality by encoding images into categorical and continuous tokens, which are modeled separately by an autoregressive model and a diffusion model, respectively. Sahoo et al. (2024b) introduce auxiliary latent variables to learn a latent lower resolution structure among images to learn pixel-wise conditional noise schedules. This allows the model to adjust the noise in the forward process dependent on low-resolution information of an image. Neural flow diffusion models (Bartosh et al., 2024) generalize this by learning the entire forward process. More generally, Pandey et al. (2022) and Kouzelis et al. (2025) show that combining low-level image details with high-level semantic features improves training efficiency and sample quality. However, the concept of resolution in images cannot be transferred to tabular data. This prevents the models above from being applicable to our case. To close this gap, we introduce a novel conceptualization of resolution in the context of tabular data and propose the first cascaded model for high-fidelity tabular data generation.

## 3. Problem Statement and Motivation

**Goal.** Let $\mathcal{D}_{\text{train}} = \{\mathbf{x}_i\}_{i=1}^N$ denote a tabular dataset with i.i.d. observations $\mathbf{x} = (\mathbf{x}_{\text{cat}}, \mathbf{x}_{\text{num}})$ drawn from an unknown distribution $p_{\text{data}}(\mathbf{x}_{\text{cat}}, \mathbf{x}_{\text{num}})$. Further, let $\mathbf{x}_{\text{cat}} = (x_{\text{cat}}^{(j)})_{j=1}^{K_{\text{cat}}}$ with $x_{\text{cat}}^{(j)} \in \{0, \ldots, C_j\}$ represent the $K_{\text{cat}}$ categorical (including binary) features; and $\mathbf{x}_{\text{num}} = (x_{\text{num}}^{(i)})_{i=1}^{K_{\text{num}}} \in \mathbb{R}^{K_{\text{num}}}$ the $K_{\text{num}}$ numerical features. The objective is to learn a (parameterized) joint distribution $p^{\boldsymbol{\theta}}(\mathbf{x}_{\text{cat}}, \mathbf{x}_{\text{num}}) \approx p_{\text{data}}(\mathbf{x}_{\text{cat}}, \mathbf{x}_{\text{num}})$ to generate new samples $\mathbf{x}^* = (\mathbf{x}_{\text{cat}}^*, \mathbf{x}_{\text{num}}^*) \sim p^{\boldsymbol{\theta}}(\mathbf{x}_{\text{cat}}, \mathbf{x}_{\text{num}})$ that match the statistical properties of the training data. Some elements of $\mathbf{x}_{\text{num}}$ may have a continuous marginal density. However, we explicitly allow for features with missing, inflated or censored values, where a given $x_{\text{num}}^{(i)}$ is of *mixed-type*. Its distribution combines a continuous density and discrete point masses, and thus differs considerably from the purely continuous distributions typically considered in diffusion-based generative models.

**Inflated values.** Consider a mixed-type feature $x_{\text{mixed}}$ with a single inflated value at $v$ and univariate density $p(x_{\text{mixed}}) = \pi_v \cdot \delta_v(x_{\text{mixed}}) + (1 - \pi_v) \cdot p_{\text{cont}}(x_{\text{mixed}})$, where $\pi_v$ is the probability mass at $v$, $p_{\text{cont}}$ is a continuous density, and $\delta_v$ is the Dirac delta function centered at $v$. Zero-inflated features ($v = 0$) are common in practice and often carry contextual information: a working time of zero hours in economic survey data may indicate unemployment; in medical data, a drug dosage of zero may indicate the absence of treatment. In both cases, the excess mass at zero represents a distinct participation state. While existing diffu-

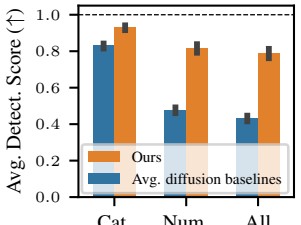 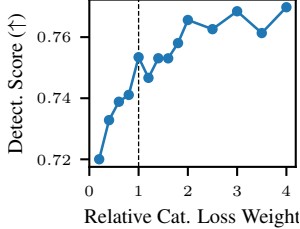

*(a)* Avg. detection scores over datasets, training and sampling seeds computed on only cat., only num., and all features.

*(b)* Detection score as a function of the relative loss weight of cat. features (from the `adult` dataset) for CDTD.

*Figure 2.* Motivational results

sion models can, in principle, generate such inflated values, they do not explicitly account for this structure. As the distribution becomes more complex, assigning precise probability mass exactly at $v$ becomes increasingly difficult. This setup naturally extends to multiple inflated values, making the discrete part of the distribution categorical instead of binary.

**Missing values.** Likewise, the discrete state in a mixed-type feature can represent missingness. Let $m = 1$ if feature $x_{\text{mixed}}$ is missing, and $m = 0$ otherwise. Then, the observed data is $x_{\text{mixed}} = (1 - m) \cdot x_{\text{num}}^{(\text{latent})} + m \cdot \texttt{NaN}$ with a latent variable $x_{\text{num}}^{(\text{latent})}$. Generally, the missingness indicator $m$ may depend on both observed and unobserved parts of the data row. The generative model must therefore also be able to infer $p(m|\mathbf{x}_{\text{num}}, \mathbf{x}_{\text{num}}^{(\text{latent})})$ for all features (Little & Rubin, 1987). This formulation is particularly relevant in domains where missing values carry information: missing answers in psychological questionnaires may point towards certain personality traits; missing values in medical datasets might indicate reluctance to disclose information. Thus, the ability to generate realistic missing values is a requirement whenever they represent informative signal required for a downstream task. Examples include the development of imputation techniques and other models that handle the missingness explicitly (Daniels & Hogan, 2008; Molenberghs et al., 2014). Previous diffusion models for tabular data can be *trained* on numerical features with missing values, but are not designed to *generate* such instances.

**The comparative ease of learning categorical features.** The premise of existing models for tabular data is to generate $\mathbf{x}_{\text{cat}}$ and $\mathbf{x}_{\text{num}}$ jointly. However, the generation performance is not equal across the two feature types. Empirical evidence in Figure 2a shows that the detection score (averaged over all datasets and diffusion-based models) estimated only on $\mathbf{x}_{\text{cat}}$ substantially exceeds the score obtained for only $\mathbf{x}_{\text{num}}$. Thus, on average, $\mathbf{x}_{\text{num}}$ is more difficult to learn and accurately generate than $\mathbf{x}_{\text{cat}}$. Figure 23 in the Appendix shows the detailed results per model. This observation motivates the divide-and-conquer approach of our model: first gen-

erating the easier component, $\mathbf{x}_{\text{cat}}$, and afterwards the more difficult part $\mathbf{x}_{\text{num}}$ conditional on $\mathbf{x}_{\text{cat}}$ to improve sample quality, resulting in improved detection scores in Figure 2a.

**The pitfall of imbalanced losses.** The heterogeneity of features requires alignment of their respective losses to avoid implicit feature importance weighting (Ma et al., 2020). For tabular data, Mueller et al. (2025) aim to achieve a proper balance from first principles as part of their CDTD model. Yet, importance parity between $\mathbf{x}_{\text{cat}}$ and $\mathbf{x}_{\text{num}}$ does not necessarily translate into better overall sample quality. For illustration, we train CDTD on the `adult` data using a grid of 14 relative loss weights for the average categorical feature loss. Figure 2b shows improvement of the detection score by increasing the relative weight of the categorical losses. In practice, however, models tend to be too large to effectively tune such hyperparameters. Our novel cascaded flow matching model avoids such balancing issues entirely, without requiring any tuning of relative loss weights.

# 4. Cascaded Flow Matching for Tabular Data

Next, we introduce TabCascade, a cascaded flow matching model for heterogeneous tabular data including mixed-type features. An overview of our approach is given in Figure 1. We outline the general framework and motivate the proposed decomposition into low- and high-resolution information in Section 4.1. We next leverage the low-resolution structure to learn feature-specific probability paths to improve the generation of $\mathbf{x}_{\text{num}}$ (Section 4.2). In addition to a high-resolution flow matching model, we adopt an efficient low-resolution model and demonstrate how a low-resolution representation of $\mathbf{x}_{\text{num}}$ can be obtained in practice (Section 4.3).

## 4.1. Cascaded Framework

**Tabular data resolution.** In images, resolution refers to the level of visual detail, typically expressed in terms of the total number of pixels. Tabular data lacks a comparable notion of resolution. Building on Figure 2a and the idea that coarse information is easier to learn than details, we link data resolution in tabular datasets to feature types, that is, we treat $\mathbf{x}_{\text{cat}}$ as low-resolution information and $\mathbf{x}_{\text{num}}$ as high-resolution information. We assume that each $x_{\text{num}}^{(i)}$ has a latent low-resolution representation $z^{(i)}$. For each data row, $\mathbf{x} = (\mathbf{x}_{\text{cat}}, \mathbf{x}_{\text{num}})$, we construct a low-resolution counterpart, $\mathbf{x}_{\text{low}} = (\mathbf{x}_{\text{cat}}, \mathbf{z})$, where $\mathbf{z} = (z^{(i)})_{i=1}^{K_{\text{num}}}$ and each $z^{(i)}$ is a categorical, low-resolution representation of $x_{\text{num}}^{(i)}$.

**Cascaded structure.** Given $\mathbf{z}$, we define the cascaded pipeline (Ho et al., 2022) as a low-resolution model followed by a high-resolution model:

$$p^{\boldsymbol{\theta}}(\mathbf{x}_{\text{cat}}, \mathbf{x}_{\text{num}}) = \sum_{\mathbf{z} \in \mathcal{Z}} p_{\text{high}}^{\boldsymbol{\theta}}(\mathbf{x}_{\text{num}} | \mathbf{z}, \mathbf{x}_{\text{cat}}) \, p_{\text{low}}^{\boldsymbol{\theta}}(\mathbf{z}, \mathbf{x}_{\text{cat}}). \quad (1)$$

Thus, we resemble a latent variable model, with the latent variable $\mathbf{z}$ generated jointly with $\mathbf{x}_{\text{cat}}$. This factorization simplifies learning the joint distribution: The generation of $\mathbf{x}_{\text{cat}}$ is informed by coarse information about $\mathbf{x}_{\text{num}}$, and enables the model to capture dependencies across feature types effectively. Additionally, conditioning on the information in $\mathbf{z}$ eases learning $p_{\text{high}}$ and generating $\mathbf{x}_{\text{num}}$. From the chain rule of entropy, we know that $\mathbb{H}(\mathbf{x}_{\text{num}} | \mathbf{z}, \mathbf{x}_{\text{cat}}) < \mathbb{H}(\mathbf{x}_{\text{num}} | \mathbf{x}_{\text{cat}})$ if $\mathbf{x}_{\text{num}} \not\perp \mathbf{z}$. We therefore aim to infer an informative $\mathbf{z}$ such that $p(\mathbf{x}_{\text{num}} | \mathbf{x}_{\text{low}})$ and $p(\mathbf{x}_{\text{low}})$ are easier to learn than the joint distribution $p(\mathbf{x}_{\text{num}}, \mathbf{x}_{\text{cat}})$.

**Mixed-type features.** We sample from $p^{\boldsymbol{\theta}}(\mathbf{x}_{\text{cat}}, \mathbf{x}_{\text{num}})$ with ancestral sampling: we first sample $\mathbf{z}, \mathbf{x}_{\text{cat}} \sim p_{\text{low}}^{\boldsymbol{\theta}}(\mathbf{z}, \mathbf{x}_{\text{cat}})$, and $\mathbf{x}_{\text{num}} \sim p_{\text{high}}^{\boldsymbol{\theta}}(\mathbf{x}_{\text{num}} | \mathbf{z}, \mathbf{x}_{\text{cat}})$ afterwards. The distribution $p_{\text{high}}^{\boldsymbol{\theta}}(\mathbf{x}_{\text{num}} | \mathbf{z}, \mathbf{x}_{\text{cat}})$ is a deterministic mixture of discrete states of interest and truly continuous values. For instance, let `NaN` and $v_{\text{infl}}$ be the missing and inflated states of $x_{\text{num}}^{(i)}$, both encoded as separate categories $c_{\text{miss}}$ and $c_{\text{infl}}$ in $z^{(i)}$. Accordingly, if $z^{(i)} = c_{\text{miss}}$, then $p_{\text{high}}^{\boldsymbol{\theta}}(\mathbf{x}_{\text{num}} | \mathbf{z}, \mathbf{x}_{\text{cat}})$ generates `NaN` with probability 1. Similarly if $z^{(i)} = c_{\text{infl}}$, then $v_{\text{infl}}$ is generated. Otherwise a continuous value is generated from a learned distribution.

Intuitively, the model first decides on the coarse structure and only fills in the details when necessary. Therefore, $p_{\text{low}}^{\boldsymbol{\theta}}$ entirely determines inflatedness and missingness. We can thus mask the corresponding instances when training $p_{\text{high}}^{\boldsymbol{\theta}}$ to free up model capacity. This setup trivially extends to any arbitrary mixed-type structure, for instance, with multiple inflated values.

## 4.2. High-Resolution Model

We first discuss the high-resolution component in the cascade in the absence of discrete states. At the end of this section we explain how these discrete states can easily be incorporated in this part of the model. To learn $p_{\text{high}}^{\boldsymbol{\theta}}$, we rely on flow matching (Lipman et al., 2023; Albergo & Vanden-Eijnden, 2023; Liu et al., 2023). For $t \in [0, 1]$, we define an ODE $\mathrm{d}\mathbf{x}_t = \mathbf{u}_t(\mathbf{x}_t | \mathbf{x}_1, \mathbf{x}_{\text{low}}) \mathrm{d}t$ with a time-dependent *guided* conditional vector field $\mathbf{u}_t(\mathbf{x}_t | \mathbf{x}_1, \mathbf{x}_{\text{low}})$ to transform samples from a source distribution $\mathbf{x}_0 \sim p_0$ to the distribution of interest $\mathbf{x}_1 \sim p_1 = \sum_{\mathbf{x}_{\text{low}} \in \mathcal{X}_{\text{low}}} p_{\text{data}}^*(\mathbf{x}_1, \mathbf{x}_{\text{low}})$ via a probability path $p_t(\mathbf{x}_t | \mathbf{x}_1, \mathbf{x}_{\text{low}})$. Averaged over the data, the vector field generates a flow $\Psi_t(\mathbf{x}_0 | \mathbf{x}_{\text{low}}) = \mathbf{x}_t \sim p_t$ such that $\Psi_0(\mathbf{x}_0 | \mathbf{x}_{\text{low}}) = \mathbf{x}_0 \sim p_0$ and $\Psi_1(\mathbf{x}_0 | \mathbf{x}_{\text{low}}) = \mathbf{x}_1 \sim p_1$.

**Guided conditional probability path.** The ODE construction requires the design of an appropriate probability path. Here, the linear path, $\mathbf{x}_t = t\mathbf{x}_1 + (1-t)\mathbf{x}_0$ with $\mathbf{x}_0 \sim \mathcal{N}(\mathbf{0}, \mathbf{I})$, is particularly popular. To account for the high feature heterogeneity and to exploit our knowledge of $\mathbf{x}_{\text{low}}$, we introduce a novel conditional probability path, guided

by feature-specific time schedules and source distributions.

First, we define a time schedule $\boldsymbol{\gamma}_t(\mathbf{x}_{\text{low}}) : t \to [0,1]^{K_{\text{num}}}$ that induces feature-specific nonlinear trajectories conditioned on $\mathbf{x}_{\text{low}}$. We enforce monotonicity of $\boldsymbol{\gamma}_t(\mathbf{x}_{\text{low}})$ with boundary conditions $\boldsymbol{\gamma}_0 = 0$ and $\boldsymbol{\gamma}_1 = 1$. We adopt a neural-network–parameterized fifth-degree polynomial in $t$, to obtain an efficient parameterization with a closed-form time derivative $\dot{\boldsymbol{\gamma}}_t$ (Sahoo et al., 2024b, Appendix A.6.1).

Second, we utilize $\mathbf{z}$ to move $\mathbf{x}_0$ closer to the target $\mathbf{x}_1$ with *data-dependent couplings* (Albergo et al., 2024). We let the coarse information about $\mathbf{x}_1$ in $\mathbf{z}$ determine the mean $\boldsymbol{\mu}(\mathbf{z}) := (\mu_{z^{(i)}})_{i=1}^{K_{\text{num}}} \in \mathbb{R}^{K_{\text{num}}}$ and scale $\boldsymbol{\sigma}(\mathbf{z}) := (\sigma_{z^{(i)}})_{i=1}^{K_{\text{num}}} \in \mathbb{R}_+^{K_{\text{num}}}$ of the source distribution:

$$\mathbf{x}_0 = \boldsymbol{\mu}(\mathbf{z}) + \boldsymbol{\sigma}(\mathbf{z})\varepsilon, \text{ with } \varepsilon \sim \mathcal{N}(\mathbf{0}, \mathbf{I}). \quad (2)$$

Here, multiplication is understood as element-wise. As $\mathbf{x}_0$ depends on $\mathbf{x}_1$ only through $\mathbf{z}$, we can derive the induced coupling

$$p(\mathbf{x}_0, \mathbf{x}_1) = \sum_{\mathbf{z} \in \mathcal{Z}} p(\mathbf{x}_0|\mathbf{z})p(\mathbf{z}|\mathbf{x}_1)p(\mathbf{x}_1) \quad (3)$$
$$= \prod_i \sum_{z^{(i)} \in \mathcal{Z}^{(i)}} p(x_0^{(i)}|z^{(i)})p(z^{(i)}|x_1^{(i)})p(\mathbf{x}_1),$$

with Gaussian component $p(x_0^{(i)}|z^{(i)}) = \mathcal{N}(\mu_{z^{(i)}}, \sigma_{z^{(i)}}^2)$ parameterized based on $z^{(i)}$. Hence, we first draw $\mathbf{x}_1 \sim p(\mathbf{x}_1)$, retrieve $z^{(i)}$ for each $x_1^{(i)}$ feature-wise, and then sample $x_0^{(i)}$ from $p(x_0^{(i)}|z^{(i)})$. Intuitively, we use $z^{(i)}$ to construct a coupling that locates each $x_0^{(i)}$ in the proximity of its target $x_1^{(i)}$.

These innovations induce a *guided conditional probability path* $p_t(\mathbf{x}_t|\mathbf{x}_1, \mathbf{x}_{\text{low}})$ such that $\mathbf{x}_t \sim p_t(\mathbf{x}_t|\mathbf{x}_1, \mathbf{x}_{\text{low}})$ with

$$\mathbf{x}_t = \boldsymbol{\gamma}_t(\mathbf{x}_{\text{low}})\mathbf{x}_1 + (1 - \boldsymbol{\gamma}_t(\mathbf{x}_{\text{low}}))[\boldsymbol{\mu}(\mathbf{z}) + \boldsymbol{\sigma}(\mathbf{z})\varepsilon]. \quad (4)$$

The probability path is defined in an augmented space such that the samples take group-conditioned paths, with the groups defined by $\mathbf{x}_{\text{low}}$. Since we impose $\boldsymbol{\gamma}_1 = 1$ and $\boldsymbol{\gamma}_0 = 0$, we obtain $p_0(\mathbf{x}_t|\mathbf{x}_1, \mathbf{x}_{\text{low}}) = p(\mathbf{x}_0|\mathbf{z})$ and $p_1(\mathbf{x}_t|\mathbf{x}_1, \mathbf{x}_{\text{low}}) = \delta_{\mathbf{x}_1}(\mathbf{x}_t)$. Thus, $p_t(\mathbf{x}_t|\mathbf{x}_1, \mathbf{x}_{\text{low}})$ defines a valid conditional probability path.

**Guided conditional vector field.** Our knowledge of $p_t(\mathbf{x}_t|\mathbf{x}_1, \mathbf{x}_{\text{low}})$ allows us to apply Theorem 3 from Lipman et al. (2023) to derive the guided conditional vector field (see Appendix A.1.1):

$$\mathbf{u}_t(\mathbf{x}_t|\mathbf{x}_1, \mathbf{x}_{\text{low}}) = \frac{\dot{\boldsymbol{\gamma}}_t(\mathbf{x}_{\text{low}})(\mathbf{x}_1 - \mathbf{x}_t)}{1 - \boldsymbol{\gamma}_t(\mathbf{x}_{\text{low}})}. \quad (5)$$

By substituting Equation (4) in Equation (5) (see Appendix A.1.2), we obtain the target in the conditional flow

matching (CFM; Lipman et al., 2023) loss:

$$\mathcal{L}_{\text{CFM}} = \mathbb{E}_{t \sim [0,1], (\mathbf{x}_1, \mathbf{x}_{\text{low}}) \sim p_{\text{data}}^*, \varepsilon \sim \mathcal{N}(\mathbf{0}, \mathbf{I})} ||\mathcal{L}_t||_2^2, \quad (6)$$
$$\mathcal{L}_t = \mathbf{u}_t^{\boldsymbol{\theta}}(\mathbf{x}_t|\mathbf{x}_{\text{low}}) - \dot{\boldsymbol{\gamma}}_t(\mathbf{x}_{\text{low}})(\mathbf{x}_1 - [\boldsymbol{\mu}(\mathbf{z}) + \boldsymbol{\sigma}(\mathbf{z})\varepsilon])$$

with velocity field $\mathbf{u}_t^{\boldsymbol{\theta}}(\mathbf{x}_t|\mathbf{x}_{\text{low}}) = \dot{\boldsymbol{\gamma}}_t(\mathbf{x}_{\text{low}})f^{\boldsymbol{\theta}}(\mathbf{x}_t, \mathbf{x}_{\text{low}}, t)$ parameterized by a neural network $f^{\boldsymbol{\theta}}$. Note, for $\boldsymbol{\gamma}_t = t \cdot \mathbf{1}, \boldsymbol{\mu}(\mathbf{z}) = \mathbf{0}$ and $\boldsymbol{\sigma}(\mathbf{z}) = \mathbf{1}$, we recover the typical loss from a flow matching model with linear paths. Having trained $\mathbf{u}_t^{\boldsymbol{\theta}}$, we simulate $\text{d}\mathbf{x}_t = \mathbf{u}_t^{\boldsymbol{\theta}}(\mathbf{x}_t|\mathbf{x}_{\text{low}})\text{d}t$ starting from $\mathbf{x}_0 \sim p(\mathbf{x}_0|\mathbf{z})$ to sample from $p_1$. The cascaded pipeline ensures that $\mathbf{x}_{\text{low}}$ will be available during generation.

**Accommodation of discrete states.** If $\mathbf{x}_1$ contains discrete states, we mask their loss contributions, as they are deterministically inferred from $\mathbf{z} \sim p_{\text{low}}^{\boldsymbol{\theta}}$. This allows $p_{\text{high}}^{\boldsymbol{\theta}}$ to focus on feature dependencies and finer details.

### 4.3. Low-Resolution Representation

We have not yet specified how $\mathbf{z}$, $\boldsymbol{\mu}(\mathbf{z})$, and $\boldsymbol{\sigma}(\mathbf{z})$ are obtained. First, $z^{(i)}$ must be categorical and only summarizes information about $x_1^{(i)}$. Second, to minimize the noise introduced to the training process of the flow model, we learn feature-specific, deterministic encoders $p(z^{(i)}|x_1^{(i)}) = \delta_v(z^{(i)})$ with $v = \text{Enc}_i(x_1^{(i)})$ to output $z^{(i)}$ before training the generative model. Finally, $\mu_{z^{(i)}}$ and $\sigma_{z^{(i)}}^2$ of $p(x_0^{(i)}|z^{(i)})$ need to be dependent on $z^{(i)}$. Based on these requirements, we propose the distributional regression tree (DT; Schlosser et al., 2019). Additionally, we experiment with a Gaussian mixture model (GMM; Bishop, 2006). For details on the encoders, we refer to Appendix A.5.

Each model efficiently learns to approximate $p(x_1^{(i)})$ with Gaussian components $p_k(x_1^{(i)}) = \mathcal{N}(\mu_k, \sigma_k^2)$, $k \in \{1, \dots, K_i\}$. We set $z^{(i)} = \arg\max_k \log w_k p_k(x_1^{(i)})$ with weights $w_k$ for the GMM; and we let $z^{(i)} = \text{Tree}(x_1^{(i)})$ be the index of the terminal leaf node $x_1^{(i)}$ is allocated to for the DT. Since each observation of $x_1^{(i)}$ gets matched with a single Gaussian component, we can directly use $\mu_{z^{(i)}} = \mu_{k=z^{(i)}}$ and $\sigma_{z^{(i)}} = \sigma_{k=z^{(i)}}$ to parameterize $p(x_0^{(i)}|z^{(i)})$ in Equation (3). If $\sigma_{z^{(i)}}^2 \approx 0$, we treat $\mu_{z^{(i)}}$ as an inflated value and account for it explicitly as explained above. Missing values are removed before fitting the encoder but afterwards added as a separate category $c_{\text{miss}}$ to $z^{(i)}$.

Intuitively, for each data point $x_1^{(i)}$, we select $p(x_0^{(i)}|z^{(i)})$ to be the Gaussian component that the encoder suggests has most likely generated it. This moves the source distribution $p(\mathbf{x}_0|\mathbf{z})$ closer to the target $p(\mathbf{x}_1)$, which benefits both training and sampling by reducing the transport cost (see Figure 3). We provide a proof below. Compared to, e.g., minibatch Optimal Transport couplings (Tong et al., 2024), our method comes at no additional costs, aside from obtaining $\mathbf{z}$.

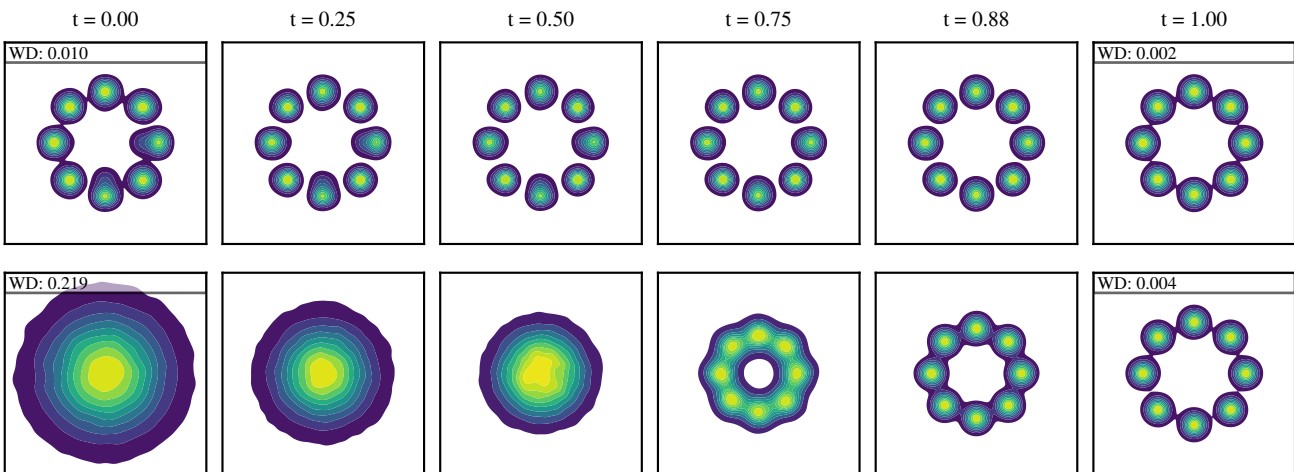

*Figure 3.* Densities $p_t$ generated from (top) a flow model with data-dependent coupling and non-linear paths, and (bottom) a classic flow model with linear paths and independent coupling. Both models condition on $\mathbf{z}$. For the top model, the source distribution is $p_0(\mathbf{x}_0) = \int p(\mathbf{x}_0, \mathbf{x}_1) \mathrm{d}\mathbf{x}_1$ with $p(\mathbf{x}_0, \mathbf{x}_1)$ as defined in Equation (3). WD represents the Wasserstein distance of $p_t$ to the true data distribution. The data-dependent coupling induces a source distribution that is much closer to the data distribution, and thus effectively reduces transport costs. Savings in model capacity and time are spent on more efficient learning of distributional details.

**Theorem 1** (Data-dependent coupling tightens transport cost bound). *Let* $\mathbf{z}$ *be derived from a DT encoder. Then, our data-dependent coupling (see Equation (3)) yields a tighter transport cost bound than an independent coupling.*

*Proof.* See Appendix A.1.3. □

### 4.4. Low-Resolution Model

Finally, the main requirements for the low-resolution model $p_{\text{low}}^{\boldsymbol{\theta}}$ to learn $p_{\text{low}}$ are the efficient and accurate generation of categorical data (and accommodating arbitrary cardinalities). A strength of our framework is that *any generative model for categorical data can be used.* For comparative purposes we choose the CDTD model (Mueller et al., 2025), which has been shown to be both efficient and effective at modeling high cardinality features.

## 5. Experiments

### 5.1. Experimental Setup

**Baselines.** We benchmark TabCascade against several state-of-the-art generative models for tabular data. These include CTGAN (Xu et al., 2019), TVAE (Xu et al., 2019), the tree-based ARF Watson et al. (2023), the diffusion-based TabDDPM (Kotelnikov et al., 2023) and TabSyn (Zhang et al., 2024b) as well as the two very recent TabDiff (Shi et al., 2025) and CDTD (Mueller et al., 2025) models.[1] For

[1]Following Mueller et al. (2025), we do not include ForestDiffusion (Jolicoeur-Martineau et al., 2024) or SMOTE (Chawla et al., 2002). For medium-sized to large datasets, these methods have been shown to suffer from a severe lack of efficiency. On adult, the default hyperparameters for ForestDiffusion require several hours of training time, which substantially exceeds the training

a fair comparison, we align all models as best as possible. Since none of the baseline models natively supports missing data generation, we augment each with a simple encoding-based mechanism for missing value simulation. Details on the implementation of the baselines and TabCascade are provided in Appendix A.3.1 and Appendix A.3.2, respectively.

**Evaluation metrics.** We evaluate all models on a broad set of standard metrics for synthetic tabular data (for details, see Appendix A.4). We consider Shape, Wasserstein distance (WD), Jensen-Shannon divergence (JSD) and Trend scores to illustrate the quality on uni- and bi-variate characteristics. We also compute the Shape (num) and Shape (cat) variants based on only numerical and only categorical features, respectively. Similarly, our Trend (mixed) metric only considers dependencies across feature types. A primary metric is the detection score, which quantifies the quality of the learned joint distribution. It is estimated based on the AUC of a strong gradient-boosting classifier trained to distinguish real from synthetic samples. Furthermore, we evaluate the performance of the synthetic relative to the real training data on downstream tasks, also known as machine learning efficiency (MLE). Additional results on fidelity, coverage and diversity are provided by the $\alpha$-Precision, $\beta$-Recall and DCR share metrics. Since our goal is to approximate the true distribution and provide a fair comparison to existing baselines, we are, similar to the baselines, not concerned with privacy considerations. However, for completeness, we do provide scores for a membership inference attack (MIA).

budget of all other diffusion-based models. Also, an early training stop is not an option: ForestDiffusion estimates separate models for each feature and each timestep. Ending training early therefore leaves the generative process incomplete.

*Table 1.* Average results across datasets and seeds. The best, row-wise result is indicated in **bold**, the second best is underlined. We report the standard deviation over the 12 datasets. Metrics with a test label have been computed relative to the test set.

| Metric | ARF | TVAE | CTGAN | TabDDPM | TabSyn | TabDiff | CDTD | Ours (DT) |
|---|---|---|---|---|---|---|---|---|
| Detection Score | $0.293_{\pm 0.191}$ | $0.205_{\pm 0.259}$ | $0.078_{\pm 0.075}$ | $0.478_{\pm 0.375}$ | $0.202_{\pm 0.173}$ | $0.430_{\pm 0.294}$ | $\underline{0.518}_{\pm 0.296}$ | $\mathbf{0.787}_{\pm 0.243}$ |
| Shape | $0.958_{\pm 0.019}$ | $0.896_{\pm 0.053}$ | $0.894_{\pm 0.039}$ | $0.938_{\pm 0.070}$ | $0.927_{\pm 0.039}$ | $0.954_{\pm 0.048}$ | $\underline{0.970}_{\pm 0.011}$ | $\mathbf{0.984}_{\pm 0.007}$ |
| Shape (test) | $0.952_{\pm 0.020}$ | $0.892_{\pm 0.052}$ | $0.892_{\pm 0.038}$ | $0.932_{\pm 0.067}$ | $0.924_{\pm 0.039}$ | $0.949_{\pm 0.047}$ | $\underline{0.964}_{\pm 0.012}$ | $\mathbf{0.975}_{\pm 0.012}$ |
| Shape (cat) | $\mathbf{0.993}_{\pm 0.005}$ | $0.903_{\pm 0.081}$ | $0.897_{\pm 0.055}$ | $0.936_{\pm 0.086}$ | $0.950_{\pm 0.031}$ | $0.972_{\pm 0.061}$ | $0.985_{\pm 0.017}$ | $\underline{0.986}_{\pm 0.012}$ |
| Shape (num) | $0.933_{\pm 0.028}$ | $0.899_{\pm 0.032}$ | $0.899_{\pm 0.025}$ | $0.943_{\pm 0.054}$ | $0.918_{\pm 0.045}$ | $0.952_{\pm 0.037}$ | $\underline{0.962}_{\pm 0.019}$ | $\mathbf{0.985}_{\pm 0.006}$ |
| WD (num) | $0.016_{\pm 0.013}$ | $0.023_{\pm 0.011}$ | $0.026_{\pm 0.015}$ | $0.015_{\pm 0.018}$ | $0.031_{\pm 0.031}$ | $0.016_{\pm 0.021}$ | $\underline{0.009}_{\pm 0.006}$ | $\mathbf{0.004}_{\pm 0.003}$ |
| WD (num, test) | $0.023_{\pm 0.022}$ | $0.027_{\pm 0.013}$ | $0.028_{\pm 0.015}$ | $0.069_{\pm 0.189}$ | $0.087_{\pm 0.186}$ | $0.021_{\pm 0.028}$ | $\underline{0.012}_{\pm 0.009}$ | $\mathbf{0.008}_{\pm 0.008}$ |
| JSD (cat) | $\mathbf{0.008}_{\pm 0.006}$ | $0.129_{\pm 0.102}$ | $0.113_{\pm 0.055}$ | $0.083_{\pm 0.103}$ | $0.063_{\pm 0.039}$ | $0.030_{\pm 0.061}$ | $0.020_{\pm 0.019}$ | $\underline{0.018}_{\pm 0.014}$ |
| JSD (cat, test) | $\mathbf{0.016}_{\pm 0.013}$ | $0.131_{\pm 0.100}$ | $0.114_{\pm 0.055}$ | $0.087_{\pm 0.101}$ | $0.065_{\pm 0.040}$ | $0.036_{\pm 0.060}$ | $0.026_{\pm 0.021}$ | $\underline{0.023}_{\pm 0.017}$ |
| Trend | $0.946_{\pm 0.039}$ | $0.852_{\pm 0.117}$ | $0.818_{\pm 0.098}$ | $0.900_{\pm 0.131}$ | $0.893_{\pm 0.071}$ | $0.924_{\pm 0.102}$ | $\underline{0.956}_{\pm 0.032}$ | $\mathbf{0.965}_{\pm 0.026}$ |
| Trend (test) | $0.919_{\pm 0.046}$ | $0.839_{\pm 0.114}$ | $0.818_{\pm 0.098}$ | $0.878_{\pm 0.126}$ | $0.876_{\pm 0.073}$ | $0.899_{\pm 0.097}$ | $\underline{0.932}_{\pm 0.037}$ | $\mathbf{0.940}_{\pm 0.034}$ |
| Trend (mixed) | $\underline{0.936}_{\pm 0.031}$ | $0.787_{\pm 0.113}$ | $0.723_{\pm 0.087}$ | $0.867_{\pm 0.137}$ | $0.867_{\pm 0.059}$ | $0.920_{\pm 0.085}$ | $0.928_{\pm 0.042}$ | $\mathbf{0.946}_{\pm 0.032}$ |
| MLE | $0.065_{\pm 0.049}$ | $0.079_{\pm 0.072}$ | $0.117_{\pm 0.069}$ | $0.312_{\pm 0.942}$ | $0.342_{\pm 0.933}$ | $0.045_{\pm 0.027}$ | $\underline{0.039}_{\pm 0.040}$ | $\mathbf{0.027}_{\pm 0.022}$ |
| $\alpha$-Precision | $0.961_{\pm 0.030}$ | $0.736_{\pm 0.274}$ | $0.858_{\pm 0.045}$ | $0.759_{\pm 0.282}$ | $0.868_{\pm 0.159}$ | $0.919_{\pm 0.100}$ | $\underline{0.971}_{\pm 0.039}$ | $\mathbf{0.975}_{\pm 0.023}$ |
| $\beta$-Recall | $0.348_{\pm 0.094}$ | $0.270_{\pm 0.208}$ | $0.214_{\pm 0.113}$ | $0.463_{\pm 0.262}$ | $0.255_{\pm 0.120}$ | $0.384_{\pm 0.186}$ | $\underline{0.580}_{\pm 0.128}$ | $\mathbf{0.591}_{\pm 0.109}$ |
| DCR Share | $0.807_{\pm 0.014}$ | $0.827_{\pm 0.053}$ | $\mathbf{0.783}_{\pm 0.017}$ | $0.861_{\pm 0.071}$ | $\underline{0.787}_{\pm 0.018}$ | $0.800_{\pm 0.041}$ | $0.883_{\pm 0.070}$ | $0.890_{\pm 0.081}$ |
| MIA Score | $0.974_{\pm 0.020}$ | $0.970_{\pm 0.027}$ | $\mathbf{0.982}_{\pm 0.016}$ | $0.953_{\pm 0.036}$ | $\underline{0.981}_{\pm 0.014}$ | $0.974_{\pm 0.019}$ | $0.958_{\pm 0.036}$ | $0.935_{\pm 0.042}$ |

As usual, any privacy guarantees would require the adoption of additional, context-specific techniques in practice. We provide modular code on all evaluation metrics to simplify future research on tabular data generation.

**Datasets.** We benchmark on a diverse set of 12 tabular datasets: Seven datasets from previous work, including `adult`, `beijing`, `default`, `diabetes`, `news`, `nmes` and `shoppers` (Kotelnikov et al., 2023; Zhang et al., 2024b; Mueller et al., 2025; Shi et al., 2025), and five datasets from the TabZilla tabular data benchmark (McElfresh et al., 2023), including `airlines`, `credit_g`, `electricity`, `kc1` and `phoneme`. The selected datasets include inflated values. The missing values are added (10%) via a simulated MNAR mechanism (Muzellec et al., 2020; Zhao et al., 2023; Zhang et al., 2024a). We provide details on the datasets and the missing value simulation in Appendix A.2.

**5.2. Results**

Table 1 summarizes the averaged results over all datasets, three training and ten sampling seeds. The training seeds also affect the missingness simulation and the train test split. TabDDPM produced NaNs for the `airlines`, `diabetes` and `news` datasets, and was assigned the worst score among all competing models. Detailed results are provided in Appendix A.8. The learned time schedules per dataset are given in Appendix A.6.2, and training and sampling times in Appendix A.10.

**State-of-the-art realism of the joint data distribution.** The detection score evaluates the realism of the joint distribution of the synthetic data, and therefore is our main metric of interest. On average, TabCascade with a DT encoder leads to 51.9% increase in the detection score compared to the best baseline (CDTD). We also provide qualitative comparisons of bivariate densities in Appendix A.7 which further illustrate that TabCascade fits the subtle details of distributions more accurately. Figure 2a illustrates the benefit of our cascaded pipeline compared the average of the competing diffusion-based models.

**Accurate feature-wise distributions.** Metrics reflecting the quality of the univariate densities, i.e., Shape, WD and JSD, indicate that TabCascade's ability to explicitly incorporate mixed-type feature distributions greatly improves the sample quality for numerical features over the baselines. The average WD decreases by more than 50% relative to the best baseline. For categorical features, it performs competitively to CDTD, mainly because of our choice of using CDTD for $p_{\text{low}}^{\boldsymbol{\theta}}$. TabCascade achieves this performance despite $p_{\text{low}}^{\boldsymbol{\theta}}$ being much smaller in parameter count compared to the baselines, as we split parameters between $p_{\text{low}}^{\boldsymbol{\theta}}$ and $p_{\text{high}}^{\boldsymbol{\theta}}$. This supports our initial motivation that categorical data distributions are easier to learn. In principle, further performance gains could be realized by choosing a different model as $p_{\text{low}}^{\boldsymbol{\theta}}$.

**Effective learning of inter-feature dependencies in a cascaded framework.** In principle, a cascaded pipeline could make it more challenging to capture dependencies across feature types compared to a joint model. However, our introduction of **z** completely alleviates this concern: On average, TabCascade performs *better* than the best baseline in terms of Trend and Trend (mixed), which evaluates the bivariate dependencies across feature types only.

**Enhanced predictive utility in downstream tasks.** The architecture of TabCascade enables a strong emphasis on distributional details, which can enhance data utility. Accordingly, we observe that, on average, TabCascade

*Table 2.* Ablation results averaged over all datasets and seeds. We report the standard deviation over the 12 datasets. The best, column-wise result is indicated in **bold**, the second best is underlined. Changing from CDTD to a flow matching (FM) high-resolution model implies *independent* coupling and *linear* paths. Grey represents the full TabCascade (DT).

| Metric | Detection Score | Shape | Shape (num) | WD (num) | Trend | Trend (mixed) | MLE | $\alpha$-Precision | $\beta$-Recall | MIA Score |
|---|---|---|---|---|---|---|---|---|---|---|
| CDTD | $0.518_{\pm0.296}$ | $0.970_{\pm0.011}$ | $0.962_{\pm0.019}$ | $0.009_{\pm0.006}$ | $0.956_{\pm0.032}$ | $0.928_{\pm0.042}$ | $0.039_{\pm0.040}$ | $0.971_{\pm0.039}$ | $0.580_{\pm0.128}$ | $\underline{0.958}_{\pm0.036}$ |
| + cascade | $0.694_{\pm0.297}$ | $0.974_{\pm0.024}$ | $0.965_{\pm0.033}$ | $0.011_{\pm0.015}$ | $0.962_{\pm0.037}$ | $0.936_{\pm0.037}$ | $0.039_{\pm0.042}$ | $0.959_{\pm0.082}$ | $0.547_{\pm0.169}$ | $0.952_{\pm0.038}$ |
| + latents **z** (DT) | $0.175_{\pm0.200}$ | $0.926_{\pm0.046}$ | $0.887_{\pm0.081}$ | $0.020_{\pm0.015}$ | $0.870_{\pm0.069}$ | $0.764_{\pm0.096}$ | $0.095_{\pm0.086}$ | $0.962_{\pm0.034}$ | $0.443_{\pm0.199}$ | $\mathbf{0.977}_{\pm0.020}$ |
| change to FM | $0.719_{\pm0.252}$ | $0.982_{\pm0.008}$ | $0.982_{\pm0.007}$ | $0.004_{\pm0.002}$ | $0.961_{\pm0.030}$ | $0.938_{\pm0.040}$ | $\underline{0.028}_{\pm0.021}$ | $0.974_{\pm0.027}$ | $0.587_{\pm0.108}$ | $0.937_{\pm0.044}$ |
| + data dep. coupling | $\underline{0.786}_{\pm0.249}$ | $\underline{0.983}_{\pm0.007}$ | $\underline{0.985}_{\pm0.006}$ | $\mathbf{0.004}_{\pm0.002}$ | $\underline{0.965}_{\pm0.025}$ | $\mathbf{0.947}_{\pm0.027}$ | $0.029_{\pm0.022}$ | $0.974_{\pm0.025}$ | $\underline{0.591}_{\pm0.110}$ | $0.934_{\pm0.041}$ |
| + non-linear paths | $\mathbf{0.787}_{\pm0.243}$ | $\mathbf{0.984}_{\pm0.007}$ | $\mathbf{0.985}_{\pm0.006}$ | $\underline{0.004}_{\pm0.003}$ | $\mathbf{0.965}_{\pm0.026}$ | $\underline{0.946}_{\pm0.032}$ | $\mathbf{0.027}_{\pm0.022}$ | $\underline{0.975}_{\pm0.023}$ | $\mathbf{0.591}_{\pm0.109}$ | $0.935_{\pm0.042}$ |
| switch DT to GMM | $0.479_{\pm0.251}$ | $0.967_{\pm0.012}$ | $0.958_{\pm0.017}$ | $0.009_{\pm0.005}$ | $0.952_{\pm0.030}$ | $0.921_{\pm0.031}$ | $0.037_{\pm0.026}$ | $\mathbf{0.976}_{\pm0.013}$ | $0.548_{\pm0.101}$ | $0.958_{\pm0.038}$ |

achieves a 30% lower MLE score relative to the best baseline, i.e., when the synthetic data is used as a plug-in replacement for the true data in a downstream task.

**Improved fidelity and coverage with moderate diversity trade-offs.** The greater focus on details naturally translates into greater sample fidelity, as highlighted by the $\alpha$-Precision and coverage, evaluated by the $\beta$-Recall score. However, moving samples to more precise areas in the data space comes with the downside of reduced diversity compared to a test set in terms of a greater DCR share.

**Privacy implications of high-fidelity synthesis.** High fidelity samples are naturally closer to the data distribution. This can have a negative impact on privacy. We emphasize that in practice samples should be carefully examined with respect to the relevant problem-specific privacy metric. Formal privacy guarantees require additional context-dependent mechanisms, like differential privacy. In Table 1, we show that privacy, when measured by MIA, remains high. The DCR share should not be interpreted as a privacy metric (Yao et al., 2025).

### 5.3. Ablation Studies

Below, we summarize the insights from multiple ablation studies. For many results we will refer to Appendix A.9.

**Impact of cascaded factorization and latent augmentation.** Table 2 compares the average performance of the vanilla CDTD (Mueller et al., 2025), to a model that adds the cascaded pipeline, i.e., specifies $p(\mathbf{x}_{cat})\,p(\mathbf{x}_{num}|\mathbf{x}_{cat})$, and a model that adds **z** to define $p(\mathbf{x}_{cat}, \mathbf{z})\,p(\mathbf{x}_{num}|\mathbf{x}_{cat}, \mathbf{z})$, including the relevant loss masking. Other hyperparameters were held constant. Results show that the CDTD model itself already benefits from the cascaded structure. However, adding the latents without the further improvements of TabCascade, leads to a substantial drop in sample quality. This may be caused by CDTD relying on learnable noise schedules that aim for the diffusion losses to develop linearly in time. Adding highly informative signal, like **z** makes this goal more difficult for the model, such that the learnable noise schedules actually become a hindrance.

**Benefits of data-dependent coupling and nonlinear probability paths.** To reap the benefits of introducing **z**, TabCascade adds data-dependent coupling and learnable, non-linear paths. As shown in Table 2, both improve the realism of the univariate and joint densities as well as the statistical dependencies among features over a vanilla flow matching (FM) model with linear paths and independent coupling. These changes greatly benefit the detection score in particular. The effect of adding non-linear paths is subtle. However, we emphasize that our specification is strictly more flexibly than fixed, linear paths. If it benefits $\mathcal{L}_{CFM}$, the learnable time schedule can become linear, see Appendix A.6.2 for illustrations.

**Impact of discretization strategy on high-resolution modeling.** In Table 2, the DT encoder consistently outperforms the GMM encoder. This is because the DT encoder induces a finer granularity into **z**, i.e., it estimates more Gaussian components. For instance, for the adult data, DT on average encodes 65.5 groups, whereas GMM only finds 12.5 on average. In addition, the reduced overlap in the Gaussian components estimated by the DT encoder (see Appendix A.5) may benefit the generative model by providing a more effective clustering of samples.

**Investigating potential overfit.** In Table 1, we also report the WD, JSD, Shape and Trend metrics computed using the test set. We can compare them to their estimated from the training sets to investigate potential overfitting. There are only mild differences between train and test set metrics. The differences we see for TabCascade are comparable with the results for the other models, indicating that TabCascade does not overfit to the data. If TabCascade would overfit, we would also expect a very low MIA score instead of the lower but still very high MIA scores we see in our results.

**Additional ablation I: Training data without missings.** In Table 21 in Appendix A.9, we confirm that TabCascade also outperforms the baselines on complete data, i.e., without any simulated missings. This highlights the value of our proposed cascaded pipeline for general tabular data generation tasks. We emphasize that generating missing values in $\mathbf{x}_{num}$ is an *option* and not a *necessity* of TabCascade. It

is evident that our cascaded approach has value beyond the generation of missing values.

**Additional ablation II: Effect of missingness rate.** We also investigate the effect of increasing the rate of simulated missings from $p = 0.10$ to $p = 0.25$ and $p = 0.50$. Table 22 in Appendix A.9 confirms the general pattern discussed above. The relative performance gain of using TabCascade over CDTD stays consistent as we increase $p$. Many metrics barely worsen, despite the significant increase in missings.

**Additional ablation III: Encoder complexity.** We provide a discussion on the effect of the varying encoder complexity in Appendix A.9.4. This includes an analysis of the proportion of masked inputs to $p_{\text{high}}^{\boldsymbol{\theta}}$. Changing the complexity allows us to navigate the fidelity-privacy trade-off, conditional on the low-resolution model choice.

**Additional ablation IV: ARF as low-resolution model.** We provide results for using ARF instead of CDTD as the low-resolution model $p_{\text{low}}^{\boldsymbol{\theta}}$ in Table 23 in Appendix A.9. We do not retrain $p_{\text{high}}^{\boldsymbol{\theta}}$. The results show that using ARF trades off a much lower detection score and $\beta$-Recall for improved univariate density metrics. Thus, depending on which metric is more important to the practitioner, a different choice of $p_{\text{low}}^{\boldsymbol{\theta}}$ can further improve the performance of TabCascade.

## 6. Conclusion

We introduced TabCascade, a cascaded flow matching model that generates high-resolution, numerical features based on their low-resolution latents and categorical features. The model builds on a novel conditional probability path guided by low-resolution information and combines it with feature-specific, learnable time schedules that enable non-linear paths. This framework allows the direct accommodation of mixed-type features and provably lowers the transport cost bound. Our extensive experiments demonstrate TabCascade's enhanced ability to generate realistic samples and learn the details of the data distribution. A multitude of ablation studies confirms the robustness of our findings and illustrates the value of the introduced model components.

Generalizing the cascaded framework to other data modalities, adopting it for data imputation, and integrating privacy guarantees are left for future work. To further improve sample quality, TabCascade could be combined with an autoregressive low-resolution model. Lastly, the number of parameters in the high-resolution model could be optimized depending on the number of numerical features and the proportion of masked entries.

## Impact Statement

This paper presents work whose goal is to advance the field of generative modeling of tabular data. Being able to generate faithful copies of true datasets comes with obvious risks, one of them being the manipulation of datasets to support otherwise untenable claims or steer the public opinion in certain directions. We advice to never blindly trust any tabular dataset but to confirm their origin, trustworthiness and integrity. This is in particular important when data is used for statistical inferences that inform decision-making processes. Any synthetic dataset should be labeled as such when it is made available to others to prevent unintended ill-usage.

## Acknowledgements

This work used the Dutch national e-infrastructure with the support of the SURF Cooperative using grant no. EINF-13805. We thank the reviewers for their constructive comments, and one anonymous reviewer in particular for helping us to fix a bug in our original post-processing routine that led to the detection model being mostly influenced by floating-point precision differences.

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

# A. Appendix

## A.1. Proofs and Derivations

### A.1.1. DERIVATION OF THE GUIDED CONDITIONAL VECTOR FIELD FOR THE HIGH-RESOLUTION MODEL

Theorem 3 in Lipman et al. (2023) proves that if the Gaussian conditional probability path is of the form $p_t(\mathbf{x}_t|\mathbf{x}_1) = \mathcal{N}(\boldsymbol{\mu}_t(\mathbf{x}_1), \sigma_t^2(\mathbf{x}_1)\mathbf{I})$ then the unique vector field that generates the flow $\Psi_t$ has the form:

$$\mathbf{u}_t(\mathbf{x}_t|\mathbf{x}_1) = \frac{\dot{\sigma}_t(\mathbf{x}_1)}{\sigma_t(\mathbf{x}_1)}(\mathbf{x}_t - \boldsymbol{\mu}_t(\mathbf{x}_1)) + \dot{\boldsymbol{\mu}}_t(\mathbf{x}_1). \tag{7}$$

In Equation (4), we implicitly define the guided conditional probability path as

$$\mathbf{x}_t = \boldsymbol{\gamma}_t(\mathbf{x}_{\text{low}})\mathbf{x}_1 + (1 - \boldsymbol{\gamma}_t(\mathbf{x}_{\text{low}}))[\boldsymbol{\mu}(\mathbf{z}) + \boldsymbol{\sigma}(\mathbf{z})\boldsymbol{\varepsilon}],$$

where multiplication is understood as element-wise. This induces the probability path

$$p_t(\mathbf{x}_t|\mathbf{x}_1, \mathbf{x}_{\text{low}}) = \mathcal{N}(\boldsymbol{\mu}_t(\mathbf{x}_1, \mathbf{x}_{\text{low}}), \text{diag}(\boldsymbol{\sigma}_t^2(\mathbf{x}_1, \mathbf{x}_{\text{low}}))), \tag{8}$$

with

$$\boldsymbol{\mu}_t(\mathbf{x}_1, \mathbf{x}_{\text{low}}) = \boldsymbol{\gamma}_t(\mathbf{x}_{\text{low}})\mathbf{x}_1 + (1 - \boldsymbol{\gamma}_t(\mathbf{x}_{\text{low}}))\boldsymbol{\mu}(\mathbf{z}), \tag{9}$$

and

$$\boldsymbol{\sigma}_t(\mathbf{x}_1, \mathbf{x}_{\text{low}}) = (1 - \boldsymbol{\gamma}_t(\mathbf{x}_{\text{low}}))\boldsymbol{\sigma}(\mathbf{z}), \tag{10}$$

since $\mathbf{x}_1$ and $\mathbf{x}_{\text{low}}$ are fixed. To proceed, note that we specified $K_{\text{num}}$ distinct Gaussian distributions. Therefore, we can simply apply Equation (7) to each element of $\mathbf{x}_t$ separately.

The time-derivatives are given by

$$\dot{\boldsymbol{\mu}}_t(\mathbf{x}_1, \mathbf{x}_{\text{low}}) = \dot{\boldsymbol{\gamma}}_t(\mathbf{x}_{\text{low}})(\mathbf{x}_1 - \boldsymbol{\mu}(\mathbf{z})) \text{ and } \dot{\boldsymbol{\sigma}}_t(\mathbf{x}_1, \mathbf{x}_{\text{low}}) = -\dot{\boldsymbol{\gamma}}_t(\mathbf{x}_{\text{low}})\boldsymbol{\sigma}(\mathbf{z}). \tag{11}$$

Plugging into Equation (7) and (for brevity) omitting the dependence of $\boldsymbol{\gamma}_t, \boldsymbol{\mu}$ and $\boldsymbol{\sigma}$ on $\mathbf{x}_{\text{low}}$ and $\mathbf{z}$, we derive the conditional vector field as

$$\begin{aligned}
\mathbf{u}_t(\mathbf{x}_t|\mathbf{x}_1, \mathbf{x}_{\text{low}}) &= \frac{-\dot{\boldsymbol{\gamma}}_t\boldsymbol{\sigma}}{(1 - \boldsymbol{\gamma}_t)\boldsymbol{\sigma}}(\mathbf{x}_t - [\boldsymbol{\gamma}_t\mathbf{x}_1 + (1 - \boldsymbol{\gamma}_t)\boldsymbol{\mu}]) + \dot{\boldsymbol{\gamma}}_t(\mathbf{x}_1 - \boldsymbol{\mu}) \\
&= \frac{-\dot{\boldsymbol{\gamma}}_t}{1 - \boldsymbol{\gamma}_t}(\mathbf{x}_t - \boldsymbol{\gamma}_t\mathbf{x}_1 - (1 - \boldsymbol{\gamma}_t)\boldsymbol{\mu} - (1 - \boldsymbol{\gamma}_t)\mathbf{x}_1 + (1 - \boldsymbol{\gamma}_t)\boldsymbol{\mu}) \\
&= \frac{\dot{\boldsymbol{\gamma}}_t(\mathbf{x}_1 - \mathbf{x}_t)}{1 - \boldsymbol{\gamma}_t}.
\end{aligned}$$

A.1.2. DERIVATION OF THE TRAINING TARGET FOR THE HIGH-RESOLUTION MODEL

To derive the training target, We plug Equation (4) into Equation (5) to get

$$
\begin{aligned}
\mathbf{u}_t(\mathbf{x}_t|\mathbf{x}_1, \mathbf{x}_{\text{low}}) &= \frac{\dot{\boldsymbol{\gamma}}_t(\mathbf{x}_{\text{low}})(\mathbf{x}_1 - \mathbf{x}_t)}{1 - \boldsymbol{\gamma}_t(\mathbf{x}_{\text{low}})} \\
&= \frac{\dot{\boldsymbol{\gamma}}_t(\mathbf{x}_{\text{low}})}{1 - \boldsymbol{\gamma}_t(\mathbf{x}_{\text{low}})} \big( (1 - \boldsymbol{\gamma}_t(\mathbf{x}_{\text{low}}))\mathbf{x}_1 - (1 - \boldsymbol{\gamma}_t(\mathbf{x}_{\text{low}}))[\boldsymbol{\mu}(\mathbf{z}) + \boldsymbol{\sigma}(\mathbf{z})\boldsymbol{\varepsilon}] \big) \\
&= \dot{\boldsymbol{\gamma}}_t(\mathbf{x}_{\text{low}}) \big( \mathbf{x}_1 - [\boldsymbol{\mu}(\mathbf{z}) + \boldsymbol{\sigma}(\mathbf{z})\boldsymbol{\varepsilon}] \big),
\end{aligned}
$$

which is the scaled difference between ground-truth sample $\mathbf{x}_1$ and source sample $\mathbf{x}_0$ from our data-dependent source distribution.

A.1.3. PROOF: DATA-DEPENDENT COUPLING TIGHTENS TRANSPORT COST BOUND

Proposition 3.1 by Albergo et al. (2024) shows that for a probability flow defined as

$$
\Psi_t(\mathbf{x}_0) = \alpha_t \mathbf{x}_1 + \beta_t \mathbf{x}_0 \in \mathbb{R}^{K_{\text{num}}},
$$

such that $\Psi_0(\mathbf{x}_0) = \mathbf{x}_0 \sim p_0$ and $\Psi_1(\mathbf{x}_0) = \mathbf{x}_1 \sim p_1$, the transport costs are upper-bounded by

$$
\mathbb{E}_{\mathbf{x}_0 \sim p_0}\big[||\Psi_1(\mathbf{x}_0) - \mathbf{x}_0||^2\big] \leq \int_0^1 \mathbb{E}[||\dot{\Psi}_t||^2]\mathrm{d}t < \infty. \tag{12}
$$

Minimizing the left-hand side implies finding the optimal transport plan as defined by Benamou & Brenier (2000), corresponding to the minimum Wasserstein-2 distance between $p_0$ and $p_1$. Below, we show that our proposed data-dependent coupling leads to a provably tighter transport cost bound when using a distributional tree (DT) as the encoder.

Our high-resolution model defines $\Psi_t(\mathbf{x}_0) = \boldsymbol{\gamma}_t \mathbf{x}_1 + (1 - \boldsymbol{\gamma}_t)\mathbf{x}_0$ such that $\dot{\Psi}_t = \dot{\boldsymbol{\gamma}}_t(\mathbf{x}_1 - \mathbf{x}_0)$.

We need to show that

$$
\int_{\mathbb{R}^{2d}} ||\dot{\Psi}_t||^2 p^*(\mathbf{x}_0, \mathbf{x}_1)\mathrm{d}\mathbf{x}_0\mathrm{d}\mathbf{x}_1 \leq \int_{\mathbb{R}^{2d}} ||\dot{\Psi}_t||^2 p(\mathbf{x}_0)p(\mathbf{x}_1)\mathrm{d}\mathbf{x}_0\mathrm{d}\mathbf{x}_1,
$$

where $p^*(\mathbf{x}_0, \mathbf{x}_1)$ is our data-dependent coupling from Equation (3) and $\mathbf{z}$ is derived by the DT encoder. We assume that $\dot{\boldsymbol{\gamma}}_t$ is the same, regardless of the used coupling.

First, for the independent coupling the expectation is taken over $\mathbf{x}_0 \sim p(\mathbf{x}_0) = \mathcal{N}(\mathbf{0}, \mathbf{I})$ and $\mathbf{x}_1 \sim p_1$ such that

$$
\begin{aligned}
\mathbb{E}[||\dot{\Psi}_t||^2] &= \mathbb{E}[||\dot{\boldsymbol{\gamma}}_t(\mathbf{x}_1 - \mathbf{x}_0)||^2] \\
&= \dot{\boldsymbol{\gamma}}_t^2\big[\mathbb{E}[||\mathbf{x}_1||^2 + ||\mathbf{x}_0||^2 - 2\mathbf{x}_1^\mathsf{T}\mathbf{x}_0]\big] \\
&= \dot{\boldsymbol{\gamma}}_t^2\big[\mathbb{E}[||\mathbf{x}_1||^2 + K_{\text{num}}]\big],
\end{aligned}
$$

where we used that $\mathrm{Var}[X] = \mathbb{E}[X^2] - \mathbb{E}[X]^2$ and $\mathrm{Cov}[X, Y] = \mathbb{E}[XY] - \mathbb{E}[X]\mathbb{E}[Y]$. We can deconstruct the expression into a sum over the $K_{\text{num}}$ features $x_1^{(i)}$:

$$
\mathbb{E}[||\dot{\Psi}_t||^2] = \dot{\boldsymbol{\gamma}}_t^2 \sum_i^{K_{\text{num}}} \Big[\mathbb{E}[(x_1^{(i)})^2]\Big] + \dot{\boldsymbol{\gamma}}_t^2 \sum_i^{K_{\text{num}}} \Big[\mathbb{E}[1]\Big]. \tag{13}
$$

For our data-dependent coupling, we have $p(\mathbf{x}_0, \mathbf{x}_1) = \sum_{\mathbf{z} \in \mathcal{Z}} p(\mathbf{x}_0|\mathbf{z})p(\mathbf{z}|\mathbf{x}_1)p(\mathbf{x}_1)$ from Equation (3) such that (from Equation (2)):

$$
\mathbf{x}_0 = \boldsymbol{\mu}(\mathbf{z}) + \boldsymbol{\sigma}(\mathbf{z})\boldsymbol{\varepsilon} \text{ with } \boldsymbol{\varepsilon} \sim \mathcal{N}(\mathbf{0}, \mathbf{I}).
$$

Since $\mathbf{z} = f(\mathbf{x}_1)$ is a deterministic function of $\mathbf{x}_1$, we only take the expectation over $\mathbf{x}_1$ and $\boldsymbol{\varepsilon}$ to derive

$$
\begin{aligned}
\mathbb{E}[||\dot{\Psi}_t||^2] &= \mathbb{E}[||\dot{\boldsymbol{\gamma}}_t(\mathbf{x}_1 - \mathbf{x}_0)||^2] \\
&= \dot{\boldsymbol{\gamma}}_t^2 \mathbb{E}[||(\mathbf{x}_1 - \boldsymbol{\mu}(f(\mathbf{x}_1)) - \boldsymbol{\sigma}(f(\mathbf{x}_1))\boldsymbol{\varepsilon}||^2]
\end{aligned}
$$

We let $z^{(i)} = f(x_1)^{(i)}$ and deconstruct the above expression as a sum over $K_{\text{num}}$ features $x_1^{(i)}$ as

$$\mathbb{E}[||\dot{\Psi}_t||^2] = \dot{\gamma}_t^2 \mathbb{E}\left[\sum_i^{K_{\text{num}}} \left(x_1^{(i)} - \mu_{z^{(i)}} - \sigma_{z^{(i)}}\varepsilon^{(i)}\right)^2\right]$$

$$= \dot{\gamma}_t^2 \mathbb{E}\sum_i^{K_{\text{num}}}\left[\left(x_1^{(i)} - \mu_{z^{(i)}}\right)^2 + \left(\sigma_{z^{(i)}}\varepsilon^{(i)}\right)^2 - 2\left(x_1^{(i)} - \mu_{z^{(i)}}\right)\sigma_{z^{(i)}}\varepsilon^{(i)}\right]$$

$$= \dot{\gamma}_t^2 \sum_i^{K_{\text{num}}}\left[\mathbb{E}\left(x_1^{(i)} - \mu_{z^{(i)}}\right)^2 + \mathbb{E}\left(\sigma_{z^{(i)}}^2(\varepsilon^{(i)})^2\right)\right],$$

since $x_1^{(i)} \perp \varepsilon^{(i)}$ which implies

$$\mathbb{E}\left(x_1^{(i)} - \mu_{z^{(i)}}\right)\sigma_{z^{(i)}}\varepsilon^{(i)} = \mathbb{E}\left[x_1^{(i)}\sigma_{z^{(i)}}\varepsilon^{(i)}\right] - \mathbb{E}\left[\mu_{z^{(i)}}\sigma_{z^{(i)}}\varepsilon^{(i)}\right]$$

$$= \mathbb{E}\left[x_1^{(i)}\sigma_{z^{(i)}}\right]\mathbb{E}\left[\varepsilon^{(i)}\right] - \mathbb{E}\left[\mu_{z^{(i)}}\sigma_{z^{(i)}}\right]\mathbb{E}\left[\varepsilon^{(i)}\right]$$

$$= 0,$$

as $\mathbb{E}[\varepsilon^{(i)}] = 0$. Using $\text{Var}[\varepsilon^{(i)}] = \mathbb{E}[(\varepsilon^{(i)})^2] - \mathbb{E}[\varepsilon^{(i)}]^2 = 1$, we can further derive

$$\mathbb{E}[||\dot{\Psi}_t||^2] = \dot{\gamma}_t^2 \sum_i^{K_{\text{num}}}\left[\mathbb{E}\left[\left(x_1^{(i)} - \mu_{z^{(i)}}\right)^2\right]\right] + \dot{\gamma}_t^2 \sum_i^{K_{\text{num}}}\left[\mathbb{E}\left[\sigma_{z^{(i)}}^2\right]\right]. \tag{14}$$

If we now compare Equation (13) and Equation (14), we recognize that to show that $\mathbb{E}[||\dot{\Psi}_t||^2]$ is smaller under our data-dependent coupling, it suffices to show feature-wise that

$$\mathbb{E}\left[\left(x_1^{(i)} - \mu_{z^{(i)}}\right)^2\right] \leq \mathbb{E}[(x_1^{(i)})^2] = 1, \tag{15}$$

since we standardize $x_1^{(i)}$ to zero mean, unit variance, and that

$$\mathbb{E}\left[\sigma_{z^{(i)}}^2\right] \leq \mathbb{E}[1] = 1. \tag{16}$$

Note that if we are using the DT encoder, $z^{(i)} = f(x_1)^{(i)}$ simply indicates in which of the $K_i$ terminal leafs the observation falls. The $k$th terminal leaf reflects an interval $[\tau_{k-1}^{(i)}, \tau_k^{(i)})$ on the real line. Based on all observations falling into the $k$th interval, DT learns a Gaussian distribution with parameters $\mu_k$ and $\sigma_k$. This allows us to rewrite Equation (15) as

$$\mathbb{E}\left[\left(x_1^{(i)} - \mu_{z^{(i)}}\right)^2\right] = \sum_{k=1}^{K_i} \Pr(\tau_{k-1}^{(i)} < x_1^{(i)} \leq \tau_k^{(i)}) \underbrace{\mathbb{E}_{x_1^{(i)}|x_1^{(i)} \in [\tau_{k-1}^{(i)}, \tau_k^{(i)})}\left[\left(x_1^{(i)} - \mu_k\right)^2\right]}_{\text{MSE in } k\text{th interval}}.$$

For each interval, the DT encoder learns the optimal $\mu_k$ by maximizing the likelihood, i.e., minimizing the mean squared error *within the kth interval*, which is equivalent to the expectation on the right-hand side. We assign the *optimal* $\mu_k$, i.e., $\mu_{z^{(i)}} = \mu_{k=z^{(i)}}$ such that the MSE is necessarily lower than choosing $\mu_k = 0$ in the case of an independent coupling. This proves that Equation (15) holds.

For proving the second condition, given in Equation (16), we only need to show $\sigma_{z^{(i)}}^2 \leq 1$ for all $x_1^{(i)}$. That is, the variance of the terminal leaf in which $x_1^{(i)}$ falls should be at most one for all possible $x_1^{(i)}$. This directly follows from the fact that we separate observations into *smaller* groups based on the intervals determined by the DT encoder. Note that $[\tau_{k-1}^{(i)}, \tau_k^{(i)}) \subseteq \text{supp}(x_1^{(i)})$ for all $k$, which implies $\sigma_k^2 \leq 1$ for all $k$.

Since both sufficient conditions in Equation (15) and Equation (16) are proven to hold, we conclude that

$$\dot{\gamma}_t^2\mathbb{E}[||(\mathbf{x}_1 - \boldsymbol{\mu}(f(\mathbf{x}_1)) - \boldsymbol{\sigma}(f(\mathbf{x}_1))\boldsymbol{\varepsilon})||^2] \leq \dot{\gamma}_t^2\left[\mathbb{E}[||\mathbf{x}_1||^2 + K_{\text{num}}\right], \tag{17}$$

i.e., our data-dependent coupling based on the DT encoder is able to achieve a lower transport cost bound than the independent coupling.

## A.2. Benchmark Datasets

Our 12 selected benchmark datasets are highly diverse, this particularly includes the number of rows and columns and the cardinality of categorical features (see Table 3). We selected the seven datasets `adult`, `beijing`, `default`, `diabetes`, `news` and `nmes` based on their popularity and usage in previous work (Kotelnikov et al., 2023; Mueller et al., 2025; Shi et al., 2025; Tiwald et al., 2025; Zhang et al., 2024b). The other datasets, i.e., `airlines`, `credit_g`, `electricity`, `kc1` and `phoneme`, were selected from the TabZilla benchmark suite for tabular data (McElfresh et al., 2023). These datasets have been shown to be associated with particularly difficult classification or regression tasks. We assume that part of this difficulty translates into more challenging generation tasks as well. Thus, the added datasets serve as an increased challenge for the generative models compared to the popular benchmark datasets. We selected datasets from TabZilla that a) are truly tabular, e.g., not just tabulated image data, b) include heterogenous features and c) do not include too many missings. All datasets are publicly accessible and licensed under creative commons. We randomly split each dataset into 70/10/20 training, validation and test sets. Numerical features in $\mathbf{x}_{\text{num}}$ are quantile transformed and standardized, following the usual practice for tabular data generation.

All synthetic data is post-processed. This includes a rounding operation to feature-specific precision. Since for training we use the float32 data format, we ensure that at the time of evaluation the maximum precision of both the synthetic and training data is six decimal digits. This is crucial for computing the detection score, as the model may otherwise pick up on differences in precision to distinguish between fake and real samples.

*Table 3.* Overview of the selected experimental datasets. We count the target towards the respective features. The minimum and maximum number of categories are taken over all categorical features.

| Dataset | License | Prediction task | Total no. observations | No. of features categorical | continuous | No. of categories min | max |
|---------|---------|-----------------|------------------------|------------------------------|------------|------------------------|-----|
| adult (Becker & Kohavi, 1996) | CC BY 4.0 | binary class. | 48 842 | 9 | 6 | 2 | 42 |
| airlines | Public | binary class. | 539 383 | 5 | 3 | 2 | 293 |
| beijing (Chen, 2015) | CC BY 4.0 | regression | 41 757 | 1 | 10 | 4 | 4 |
| credit_g | CC BY 4.0 | binary class. | 1 000 | 14 | 7 | 2 | 11 |
| default (Yeh, 2009) | CC BY 4.0 | binary class. | 30 000 | 10 | 14 | 2 | 11 |
| diabetes (Clore et al., 2014) | CC BY 4.0 | binary class. | 101 766 | 29 | 8 | 2 | 523 |
| electricity | CC BY 4.0 | binary class. | 45 312 | 2 | 7 | 2 | 8 |
| kc1 (Niu & Mahmoud, 2012) | Public | binary class. | 2 109 | 1 | 12 | 2 | 2 |
| news (Fernandes et al., 2015) | CC BY 4.0 | regression | 39 644 | 14 | 46 | 2 | 2 |
| nmes (Deb & Trivedi, 1997) | Public | regression | 4 406 | 9 | 10 | 2 | 4 |
| phoneme | Public | binary class. | 5 404 | 1 | 5 | 2 | 2 |
| shoppers (Sakar et al., 2019) | CC BY 4.0 | binary class. | 12 330 | 8 | 10 | 2 | 20 |

**Missing value simulation.**    First, we remove any rows with missing values in the target, to ensure a valid estimation of the MLE metric, or in any of the numerical features. This gives us full control over the missingness proportion and mechanism. To simulate missingness, we adopt the approach from prior imputation studies (see e.g., Muzellec et al., 2020; Zhao et al., 2023; Zhang et al., 2024a). We choose to simulate missing values for numerical features under a missing not at random (MNAR) mechanism, as it combines a missing at random (MAR), i.e., $p(\mathbf{m}|\mathbf{x}^{(\text{observed})}, \mathbf{x}^{(\text{missing})}) = p(\mathbf{m}|\mathbf{x}^{(\text{observed})})$, with a missing completely at random (MCAR), i.e., $p(\mathbf{m}|\mathbf{x}^{(\text{observed})}, \mathbf{x}^{(\text{missing})}) = p(\mathbf{m})$, mechanism (see Little & Rubin, 1987). Following prior work (Muzellec et al., 2020; Zhao et al., 2023; Zhang et al., 2024a), we simulate missing values using a two-step procedure. First, under a MAR mechanism, we randomly select 30% of the numerical and categorical features as inputs to a randomly initialized logistic model, to determine the missingness probabilities for the remaining numerical features. The model's coefficients are scaled to preserve variance, and the bias term is adjusted via line search to achieve a 10% missing rate. Second, we apply an MCAR mechanism by setting 10% of the logistic model's input features (including selected categorical ones) to missing. Thus, the missingness introduced by the MAR mechanism may be explained by values which now have been masked by the MCAR mechanism, making them latent to the model. Throughout, we ensure that we do not introduce any missings to the target, to ensure that we can determine the MLE metric. Introducing non-trivial missings increases the complexity of the joint distribution, both in terms of dimensions and dependencies, and makes the job for the generative models more difficult. Missing values in categorical features are simply encoded as a separate category.

## A.3. Implementation Details

### A.3.1. BASELINE IMPLEMENTATIONS

We benchmark TabCascade against recent state-of-the-art generative models, many of which are diffusion-based. To ensure that the benchmarks are fair, we align the models as much as possible. For diffusion-based models, we use the same MLP-based architecture with the same bottleneck dimension. The MLP contains a projection layer onto the bottleneck dimension (256-dimensional), five fully connected layers, and an output layer. The only differences stem from variations in the required inputs or outputs, which make certain minor model-specific changes to the MLP necessary, e.g., CDTD requires predicted logits for categorical features. For all models, we use the same time encoder based on positional embeddings with a subsequent two-layer MLP. For non-diffusion-based models, we try to align the layer dimensions. In any case, similar to Mueller et al. (2025) we scale each model to a total of approx. 3 million parameters on the adult dataset (when simulating missing values according to the MNAR mechanism) and train it for 30 000 steps with a batch size of 4096. For diffusion-based models, we limit the maximum training time to 30 minutes to increase model comparability. We use the same data pre-processing pipeline for all models and add model-specific pre-processing steps where necessary. For diffusion-based models, we mostly align the sampling steps to 200. One exception is TabDDPM, which builds on DDPM and therefore requires more sampling steps (default = 1000). A second exception is TabDiff, for which we adopt the authors' suggestion of 50 sampling steps. Otherwise, TabDiff sampling will take an order of magnitude more time than other models, in particular for larger datasets. When available, we follow the default hyperparameters provided by the authors or the package / code documentation. We run all experiments using PyTorch 2.7.1 and TensorFloat32 using a MIG instance on an A100 GPU. All code and configuration files are made available to ensure reproducibility.

Below, we briefly elaborate on each baseline model and its implementation:

**ARF** (Watson et al., 2023) – a generative model that is based on a random forest for density estimation. The implementation is available at https://github.com/bips-hb/arfpy and licensed under the MIT license. We use package version 0.1.1. For training, we utilize 16 CPU cores and 20 trees as suggested in the paper.

**CTGAN** (Xu et al., 2019) – one of the most popular GAN-based models for tabular data. The implementation is available as part of the Synthetic Data Vault (Patki et al., 2016) at https://github.com/sdv-dev/CTGAN and licensed under the Business Source License 1.1. We use package version 0.11.0. The architecture dimensions are adjusted to be comparable to MLP used for the diffusion-based models. The model requires that the batch size is divisible by 10. Therefore, we adjust the default batch size of 4096 downwards accordingly.

**TVAE** (Xu et al., 2019) – a VAE-based model for tabular data. The implementation is available as part of the Synthetic Data Vault (Patki et al., 2016) at https://github.com/sdv-dev/CTGAN and licensed under the Business Source License 1.1. We use package version 0.11.0. The architecture dimensions are adjusted to be comparable to MLP used for the diffusion-based models.

**TabDDPM** (Kotelnikov et al., 2023) – a diffusion-based generative model for tabular data that combines multinomial diffusion (Hoogeboom et al., 2021) and DDPM (Sohl-Dickstein et al., 2015; Ho et al., 2020). We base our code on the official implementation available at https://github.com/yandex-research/tab-ddpm under the MIT license. However, we adjust the model to allow for unconditional generation in case of classification tasks.

**TabSyn** (Zhang et al., 2024b) – a latent diffusion model that first learns a transformer-based VAE to map mixed-type data to a continuous latent space. The diffusion model is then trained on that latent space. Note that despite TabSyn utilizing a separately trained encoder, this does *not* result in a lower-dimensional latent space and therefore, does not speed up sampling. We use the official code available at https://github.com/amazon-science/tabsyn under the Apache 2.0 license. We leave the transformer-based VAE unchanged and scale only the MLP.

**TabDiff** (Shi et al., 2025) – a continuous time diffusion model that combines score matching (Song et al., 2021; Karras et al., 2022) with masked diffusion (Sahoo et al., 2024a) and learnable, feature-specific noise schedules. Originally, it relies on transformer-based encoder and decoder parts, which we remove from the model to improve comparability. However, we keep the other parts, including the tokenizer. We scale the bottleneck dimension down to 256 and adjust the hidden layers accordingly, to align the architecture more with the other diffusion-based models. Otherwise, we use the official implementation available at https://github.com/MinkaiXu/TabDiff under the MIT license.

**CDTD** (Mueller et al., 2025) – a continuous time diffusion that combines score matching (Song et al., 2021; Karras et al., 2022) with score interpolation (Dieleman et al., 2022) and leanable noise schedules. Based on the performance results in the original paper, we use the *by type* noise schedule, that is, we learn an adaptive noise schedule per feature type. We use the official implementation available at `https://github.com/muellermarkus/cdtd_simple` under the MIT license. To align architectures and improve comparability, we adjust the MLP dimensions.

None of the selected benchmark models accommodate the generation of missing values in numerical features out of the box. Therefore, to achieve a fair comparison, we endow each model with the simple means to generate missing values. To avoid manipulating a model's internals and therewith potentially disrupting the training dynamics, we confine ourselves to changing the data encoding. For each numerical feature that contains missing values, we introduce an additional binary missingness mask. We simply treat this mask as an additional categorical feature to be generated, and we mean-impute the missing values. After sampling, we overwrite the generated numerical values with `NaN` based on the generated missingness mask.

### A.3.2. TABCASCADE IMPLEMENTATION

Since we make use of two separate models instead of a single model, we use the same MLP architecture as for the baselines for both, but scale various layers and components down to achieve approx. 3 million total parameters on the `adult` dataset. These are divided into approx. 2 million parameters for the low-resolution model $p_{\text{low}}^{\boldsymbol{\theta}}$ and approx. 1 million parameters for the high-resolution model $p_{\text{high}}^{\boldsymbol{\theta}}$. We add the conditioning information about $\mathbf{x}_{\text{low}}$ as an additive embedding to the bottleneck layer. We train $p_{\text{low}}^{\boldsymbol{\theta}}$ and $p_{\text{high}}^{\boldsymbol{\theta}}$ simultaneously using teacher forcing. That is, we train $p_{\text{high}}^{\boldsymbol{\theta}}$ using the real data instances, instead of the ones generated by $p_{\text{low}}^{\boldsymbol{\theta}}$. This enables an end-to-end training of two separate models with a reduced time penalty. The training and generation processes are described in detail in Algorithm 1 and Algorithm 2 below. Compared to the sampling process of CDTD, we cache the normalized embeddings used in $p_{\text{low}}^{\boldsymbol{\theta}}$ at the start of generation to improve sampling efficiency.

The encoders are trained on the pre-processed numerical features. For the DT encoder we set a max depth of 8 which on the `adult` dataset translates to an average of 65.5 distinct groups for each feature that are captured by $\mathbf{z}$. For the GMM encoder, we set the maximum number of components to 30 to keep the training time below 1 minute on the `adult` dataset. Empirical evidence shows that this does not tend to limit the estimated number of components, which typically lies below 30, e.g., 12.5 on average on the `adult` dataset.

---

**Algorithm 1** Training

---

**# Pre-training**
Learn feature-wise encoder $z^{(i)} = \text{Enc}_i(x_{\text{num}}^{(i)})$

**# Training**
Sample $\mathbf{x}_{\text{num}}, \mathbf{x}_{\text{cat}} \sim p_{\text{data}}$
Retrieve $z^{(i)} = \text{Enc}_i(x_{\text{num}}^{(i)}) \forall i$ and construct $\mathbf{x}_{\text{low}} = (\mathbf{x}_{\text{cat}}, \mathbf{z}) = (x_{\text{low}}^{(j)})_{j=1}^{K_{\text{low}}}$
Construct mask for inflated and missing values in $\mathbf{x}_{\text{num}}$

**# Low-resolution Model**
Train CDTD model (Mueller et al., 2025)

**# High-resolution Model**
Sample $t \sim \mathcal{U}(0,1)$ and $\boldsymbol{\varepsilon} \sim \mathcal{N}(\mathbf{0}, \mathbf{I})$
Compute $\mathbf{x}_0$ using Equation (2), set $\mathbf{x}_1 = \mathbf{x}_{\text{num}}$
Compute $\mathbf{x}_t = \boldsymbol{\gamma}_t(\mathbf{x}_{\text{low}})\mathbf{x}_1 + (1 - \mathbf{x}_{\text{low}})\mathbf{x}_0$
Form predictions $\mathbf{u}_t^{\boldsymbol{\theta}}(\mathbf{x}_t|\mathbf{x}_{\text{low}}) = \dot{\boldsymbol{\gamma}}_t(\mathbf{x}_{\text{low}}) f^{\boldsymbol{\theta}}(\mathbf{x}_t, \mathbf{x}_{\text{low}}, t)$
Compute MSE between $\mathbf{u}_t^{\boldsymbol{\theta}}(\mathbf{x}_t|\mathbf{x}_{\text{low}})$ and the target as in Equation (6) (mask losses for missing and inflated values)
Backpropagate.

---

---

**Algorithm 2** Generation

---

    **# Low-resolution Model** (for more details, see Mueller et al. (2025))

    Sample $\mathbf{x}_0^{(j)} \sim \mathcal{N}(\mathbf{0}, \mathbf{I}) \, \forall j$ categorical features

    **for** $t$ in $t_{\text{grid}}$ with step size $h$ **do**

        Predict $\Pr\left(x_{\text{low}}^{(j)} = c \middle| (\mathbf{x}_t^{(j)})_{j=1}^{K_{\text{cat}}}, t\right) \forall c \in \{0, 1, \ldots, C_j\} \, \forall j$

        Compute $\boldsymbol{\mu}_t^{(j)} = \sum_{c=1}^{C_j} \Pr\left(x_{\text{low}}^{(j)} = c \middle| (\mathbf{x}_t^{(j)})_{j=1}^{K_{\text{low}}}, t\right) \cdot \mathbf{x}_1^{(j)}(c) \, \forall j$, where $\mathbf{x}_1^{(j)}(c)$ is the embedding of category $c$

        Compute $\mathbf{u}_t^{(j)}(\mathbf{x}_t|\mathbf{x}_1) = \frac{\boldsymbol{\mu}_t^{(j)} - \mathbf{x}_t^{(j)}}{\sigma^2(t)}$

        Take update step $\mathbf{x}_t^{(j)} = \mathbf{x}_t^{(j)} + h \cdot \mathbf{u}_t^{(j)}(\mathbf{x}_t|\mathbf{x}_1) \, \forall j$

    **end for**

    Assign classes based on $\arg\max_c \Pr\left(x_{\text{low}}^{(j)} = c \middle| (\mathbf{x}_1^{(j)})_{j=1}^{K_{\text{low}}}, t = 1 - h\right) \forall c \in \{0, 1, \ldots, C_j\} \, \forall j$

    **# High-resolution Model**

    Retrieve $\boldsymbol{\mu}(\mathbf{z}), \boldsymbol{\sigma}(\mathbf{z})$ and sample $\mathbf{x}_0$ using Equation (2)

    Solve ODE $\mathbf{x}_{\text{num}} = \mathbf{x}_0 + \int_{t=0}^{t=1} \dot{\boldsymbol{\gamma}}(\mathbf{x}_{\text{low}}) f^{\boldsymbol{\theta}}(\mathbf{x}_t, \mathbf{x}_{\text{low}}, t) \mathrm{d}t$

    **# Post-process Samples**

    Overwrite $\mathbf{x}_{\text{num}}$ with inflated or missing values using the information in $\mathbf{z}$

    Return $\mathbf{x}_{\text{cat}}, \mathbf{x}_{\text{num}}$

---

## A.4. Evaluation Metrics

**Univariate densities (Shape, WD, JSD).** To evaluate the quality of the column-wise, univariate densities, we mainly use the popular Shape metric, which is part of the SDMetrics library (version 0.22.0) of the Synthetic Data Vault (Patki et al., 2016). This metric is constructed as follows: For numerical features, we use the Kolmogorov-Smirnov statistic $K_{\text{stat}} \in [0, 1]$ and compute the score as $1 - K_{\text{stat}}$ feature-wise. Note that $K_{\text{stat}}$ cannot be computed from observations with missing values, which are therefore removed beforehand. For categorical features, we compute the Total Variation Distance (TVD) based on the empirical frequencies of each category value, expressed as proportions $R_c$ and $S_c$ in the real and synthetic datasets, respectively. The TVD between real and synthetic datasets is then given as

$$\delta(R, S) = \frac{1}{2} \sum_{c \in \mathcal{C}} |R_c - S_c|.$$

Again, we let the score be $1 - \delta(R, S)$ to ensure that an increasing score (up to 1) indicates improved sample quality. The average score over all features gives the Shape score reported in our results. We report similar scores for numerical and categorical features only, and indicate them by Shape (num) and Shape (cat), respectively.

To get a more nuanced impression of the univariate densities, we additionally report the Wasserstein distance (WD) for numerical features and the Jensen-Shannon divergence (JSD) for categorical features. Qualitatively, we expect them to convey the same information as the Shape metric.

**Bivariate densities (Trend).** To get a better idea of the accuracy of feature interactions in the synthetic data, we evaluate the Trend score, which is another metric provided by the SDMetrics library (version 0.22.0) of the Synthetic Data Vault (Patki et al., 2016). This metric focuses on the accuracy of pair-wise correlations. Hence, the aim is to compute a score between every pair of features. For two numerical features, we can simply compute the Pearson correlation coefficient. We denote the score as

$$d_{i,j}^{\text{num}} = 1 - 0.5 \cdot |S_{i,j} - R_{i,j}|,$$

where $S_{i,j}$ and $R_{i,j}$ represent the Pearson correlation between features $i$ and $j$ computed on the synthetic and real data, respectively.

For two categorical features, we derive the score from the normalized contingency tables, i.e., from the proportion of samples in each possible combination of categories. To determine the difference between real and synthetic data, we can use the

Total Variation Distance (TVD) such that

$$d_{i,j}^{\text{cat}} = 1 - 0.5 \sum_{c_i \in \mathcal{C}_i} \sum_{c_j \in \mathcal{C}_j} |S_{c_i, c_j} - R_{c_i, c_j}|,$$

where $\mathcal{C}_i$ and $\mathcal{C}_j$ are the set of categories of features $i$ and $j$ and $S_{i,j}, R_{i,j}$ are the cells from the normalized contigency table corresponding to these categories.

To be able to compute a comparable score when comparing features of different types, i.e., a numerical and a categorical feature, we first discretize the numerical feature into ten bins and then compute the TVD as explained above. For all scores, a higher score indicates better sample quality. The overall Trend score is the average over all pair-wise scores. Since this metric cannot accommodate missing values in numerical features, we remove observations with missings in the relevant features beforehand. Lastly, to provide further insight into correlations across data types, we also report a Trend (mixed) metric.

**Joint density (Detection score).** While the other metrics so far focus on the sample quality in terms of univariate densities or pair-wise distributions, we are particularly interested in the overall quality of the full joint distribution. Following the typical approach in the literature (Bischoff et al., 2024; Mueller et al., 2025; Shi et al., 2025), we train a detection model to differentiate between fake and real samples, which make up the training data in equal proportions. This approach is also called a classifier two-sample test (C2ST) (Bischoff et al., 2024).

To ensure that the detection model is sensitive to small changes in the distribution, we choose LightGBM (Ke et al., 2017). Gradient-boosting models have shown remarkable performance on tabular datasets (Borisov et al., 2022). LightGBM has been particularly designed for improved efficiency, which is important for the evaluation of the detection score on larger datasets. Another advantage is that it naturally accommodates missings in numerical features. This allows the detection score to indirectly capture how well the generative model learned the missingness mechanism. To train LightGBM, we sample a synthetic dataset of the same size as the training set used for the generative model. The objective is to classify whether a given sample is real or synthetic. We use 5-fold cross-validation to estimate the out-of-sample performance, with a max depth of 5 and 500 boosting iterations. To get the final detection score, we first record the highest average AUC obtained over validation sets across boosting iterations, denoted by $\bar{A}$. The detection score is then computed as

$$\text{Detection Score} = 1 - (\max(0.5, \bar{A}) \cdot 2 - 1),$$

such that a score of one indicates that the model cannot distinguish between fake and real samples at all. On the other extreme, a score of zero indicates that the model can perfectly classify the samples into fake and real. This procedure mimics the detection metric in the SDMetrics library of the Synthetic Data Vault (Patki et al., 2016) but uses a much more powerful detection model.

**Downstream-task performance (Machine learning efficiency).** Machine learning efficiency (MLE; sometimes also called efficacy or utility) measures the usefulness of the synthetic data for the downstream prediction task, either binary classification or regression, associated with a given dataset. This represents a train-synthetic-test-real strategy: We train a predictor on the synthetic data and test the predictor's out-of-sample performance on the real test data. Similarly, we get the test set performance by training the predictor on the real training data. For regression tasks, we evaluate the RMSE and for classification tasks the AUC. Since our goal is to generate a realistic and faithful copy of the true data, we expect both models to perform similarly on the downstream task, regardless of which data has been used for training. Thus, only the relative comparison of the model performances matters, which we report using their absolute difference

$$\text{MLE Score} = |M_S - M_R|, \text{ with } M \in \{\text{AUC}, \text{RMSE}\}.$$

As the predictor, we again pick LightGBM (Ke et al., 2017) with a max depth of 5 and 500 boosting iterations because of its efficiency and strong predictive performance on tabular data. It also automatically accommodates missings in numerical features. Note that the generative model's ability to generate missing values is evaluated in two different ways: (1) LightGBM may rely directly on missing values to infer the target and (2) the generative model may place missing values incorrectly and thereby eradicates information that would be needed (and is available in the true training data) for the prediction task. Hence, there is a twofold negative impact of a generative model that is not able to accurately learn the missingness mechanism on the downstream task performance.

**Diversity (Distance to closest record share).** Our goal is to approximate the true generative process and provide a fair comparison to existing baselines. As such, we are, similar to previous work, not concerned with any privacy considerations. To obtain privacy guarantees, context-specific choices, for instance, with regards to the budget for differential privacy, must be made. Such in-processing privacy mechanisms as well as pre-processing and post-processing techniques are typically model agnostic but depend heavily on the dataset as well other considerations, such as legal and ethical questions. Hence, we investigate the distance to closest record (DCR) share only as a metric of diversity rather than privacy. Most importantly, it can inform about models which simply copy training samples, without actually learning the distribution.

To ensure all features are on the same scale, we min-max-scale numerical features and one-hot encode categorical features. We allow for missing values in numerical features by using mean imputation and adding the missingness indicator to the one-hot encoded categorical features. For each synthetic sample we then find the nearest neighbor in the training set in terms of their $L_2$ distance (Zhao et al., 2021). Since the DCR is only meaningful when compared to some reference, we report the DCR share (Zhang et al., 2024b; Shi et al., 2025). Let $d_{\text{train}}^{(i)}$ and $d_{\text{test}}^{(i)}$ be the $L_2$ distance of the $i$-th synthetic sample to the closest training and test sample, respectively. Then we set

$$S^{(i)} = \begin{cases} 1 & \text{if } d_{\text{train}}^{(i)} < d_{\text{test}}^{(i)}, \\ 0 & \text{if } d_{\text{train}}^{(i)} > d_{\text{test}}^{(i)}, \\ 0.5 & \text{if } d_{\text{train}}^{(i)} = d_{\text{test}}^{(i)}, \end{cases}$$

such that synthetic samples being closer to the training samples than the test samples increase the score. The DCR share is then computed as an average over the scores $S^{(i)}$ obtained for all synthetic samples. In the absence of ties, the optimal DCR share is $N_{\text{train}}/(N_{\text{train}} + N_{\text{test}}) \approx 0.778$, where $N_{\text{train}}$ and $N_{\text{test}}$ are the train and test set sizes, respectively.

**Fidelity and coverage ($\alpha$-Precision and $\beta$-Recall).** Precision and Recall metrics for generative model evaluation have been proposed by Sajjadi et al. (2018) and refined for tabular data by Alaa et al. (2022). $\alpha$-Precision measures the probability that synthetic samples resides in the $\alpha$-support of the true distribution and therefore measures sample fidelity. $\beta$-Recall, on the other hand, measures the sample diversity or coverage. That is, what fraction of real samples reside in the $\beta$-support of the generative distribution. For both metrics, higher values indicate better sample quality. For estimation, we rely on the official implementation in the synthcity package (Qian et al., 2023) available at `https://github.com/vanderschaarlab/synthcity`. However, we need to make some minor adjustments, in the same way as for the DCR computation, to accommodate missing values in numerical features.

**Privacy (Membership inference attack).** For completeness, we also the provide scores of a membership inference attack (MIA; Shokri et al., 2017). We follow the implementation in the SynthEval package (Lautrup et al., 2024) available at `https://github.com/schneiderkamplab/syntheval/`.

Let $\mathcal{D}_{\text{train}}, \mathcal{D}_{\text{test}}$ and $\mathcal{D}_{\text{gen}}$ be the training set, test set, and generated data, respectively. First, we split $\mathcal{D}_{\text{test}}$ into $\mathcal{D}_{\text{test}}^{(\text{train})}$ (75%) and $\mathcal{D}_{\text{test}}^{(\text{test})}$ (25%). We then train a LightGBM classifier (Ke et al., 2017) on a training set made up of $\mathcal{D}_{\text{test}}^{(\text{train})}$ and an equally-sized subsample of $\mathcal{D}_{\text{gen}}$. The classifier is trained to predict which samples originated from the generative model. To retrieve score, we combine $\mathcal{D}_{\text{test}}^{(\text{test})}$ with an equally-sized subsample of $\mathcal{D}_{\text{train}}$ and use the predictions to compute the AUC score. We then derive the MIA score as

$$\text{MIA Score} = 1 - (\max(0.5, \text{AUC}) \cdot 2 - 1),$$

such that a score of one indicates that an attack is not better than random guessing. The final score we report is an average over five repetitions of the above steps, to account for the uncertainty in the subsampling.

## A.5. Encoder Details

To encode each $x_{\text{num}}^{(i)}$ into its categorical low-resolution representation $z^{(i)}$, we propose two different encoders: (1) a Dirichlet Process Variational Gaussian Mixture Model and (2) a distributional regression tree. Below, we briefly elaborate on their respective implementations and explain our reasoning behind as well as the differences between these choices.

### A.5.1. GAUSSIAN MIXTURE MODEL

An obvious choice for an encoder is a Gaussian Mixture Model (GMM), because it can approximate any density arbitrarily closely. However, its classical variant requires pre-specification of the number of components $K$. This is not desirable, since it

would require setting a potentially different $K$ for each feature. Instead, we rely on the Dirichlet Process Variational Gaussian Mixture Model (Bishop, 2006) as provided by the sklearn package. The combination with a Dirichlet Process leads to a mixture of a theoretically infinite number of components. For practical purposes, this allows us to avoid specifying the number of components per feature and instead infer them directly from the data. We specify a weight concentration prior of 0.001, following settings in Synthetic Data Vault (Patki et al., 2016) package RDT (see `https://github.com/sdv-dev/RDT`). A low prior encourages the model to put most weight on few components, leading to fewer estimated components after training.

During training, the Variational GMM maximizes a variational lower bound to the maximum likelihood objective and does soft clustering of the data points. To assign an observation $x_{\text{num}}^{(i)}$ to a discrete category $z^{(i)}$ after training and achieve a hard clustering, we let

$$z^{(i)} = \arg\max_k w_k \log p_k(x_{\text{num}}^{(i)}) = \arg\max_k \log w_k \mathcal{N}(x_{\text{num}}^{(i)}; \mu_k, \sigma_k^2),$$

where the $w_k$ are the mixture weights. A drawback of the GMM is that its components may substantially overlap (see Figure 4). For instance, it is possible that a small variance Gaussian lies in the middle of a high variance Gaussian if this benefits the overall fit. This can make the cluster derived from hard clustering disconnected on the real line. After assigning data points to clusters, this can also cause the mean of the Gaussian component to deviate from the actual mean within the cluster. To address these downsides, we investigate the use of a distributional regression tree instead.

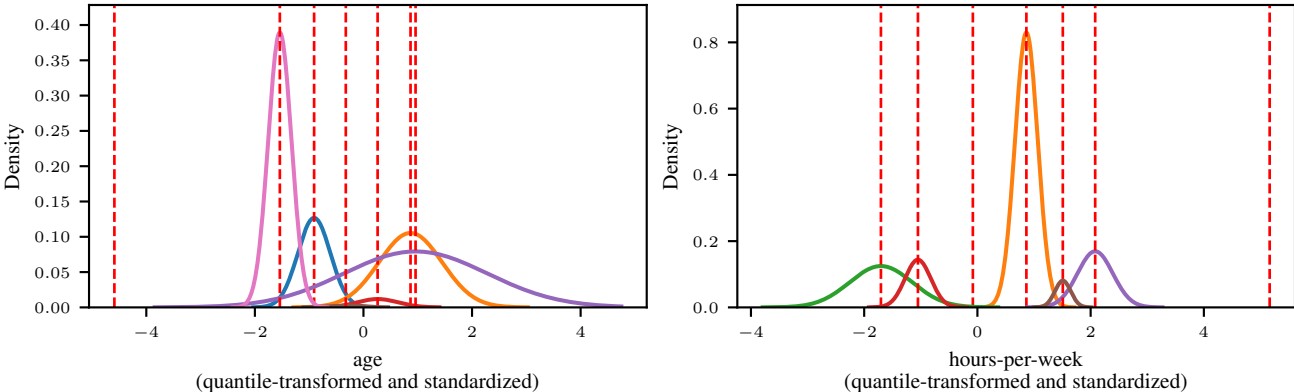

*Figure 4.* Gaussian components found by the GMM encoder (max components = 7, to align with the number of components found by DT) for two features in the `adult` dataset. The red vertical lines indicate the means of the Gaussian components.

### A.5.2. DISTRIBUTIONAL REGRESSION TREE

Trees split the data into more homogeneous subgroups via binary splits. This can capture abrupt shifts and non-linear functions. Distributional regression trees (DT; Schlosser et al., 2019) utilize the non-parametric nature of trees and combine it with parametric distributions. The goal is to find homogeneous groups with respect to a parametric distribution such that the model captures abrupt changes in any distributional parameters, such as the mean and variance of a Gaussian distribution.

Training a DT can be interpreted as maximizing a weighted likelihood over $n$ observations:

$$\hat{\boldsymbol{\theta}}(x_{\text{num}}^{(i)}) = \max_{\boldsymbol{\theta} \in \Theta} \sum_{k=1}^{K} w_k(x_{\text{num}}^{(i)}) \cdot \ell(\boldsymbol{\theta}_k; x_{\text{num}}^{(i)}), \tag{18}$$

where $\boldsymbol{\theta}_k = (\mu_k, \sigma_k)$ are the parameters of the $k$th Gaussian component. Note that unlike the GMM, the tree-based approach directly leads to a hard clustering since $w_k(x_{\text{num}}^{(i)}) \in \{0, 1\}$ simply indicates the allocated terminal leaf for that data point. For each $x_{\text{num}}^{(i)}$, the fitting algorithm goes through the following steps:

- estimate $\hat{\boldsymbol{\theta}}$ via maximum likelihood,
- test for associations or instabilities of the score $\frac{\partial \ell}{\partial \boldsymbol{\theta}}(\hat{\boldsymbol{\theta}}; x_{\text{num}}^{(i)})$,
- choose split of $\text{supp}(x_{\text{num}}^{(i)})$ that yields the highest improvement in the log likelihood,
- repeat until convergence.

A DT exhibits various benefits compared to the GMM encoder. It searches for a partitioning of $\text{supp}(x_{\text{num}}^{(i)})$ such that values falling into a given segment are more homogeneous with respect to the moments of the Gaussian distribution. Hence, it directly optimizes a hard clustering of data points and defines a Gaussian component only within the resulting clusters. This substantially reduces the possible overlap of the Gaussian components compared to GMM, a feature which allows us to prove Theorem 1. For empirical evidence, compare Figure 5 to Figure 4. This is also an attractive property when determining a suitable Gaussian-based source distribution for flow matching, as sampling from a Gaussian component guarantees that samples are close in data space.

The level of granularity captured by $z^{(i)}$ is governed by the complexity of the encoder. DT allows us to specify a maximum tree depth but otherwise learns the optimal number of components from the data. Additionally, it is much faster to train than a GMM. We investigate the effect of increasing max depth in additional ablation experiments in Appendix A.9.4.

Since no Python implementation of DT is available and the disttree R package is rather outdated, we fork the package and combine it with rpy2 to make it callable in Python. An install script is provided as part of our code repository.

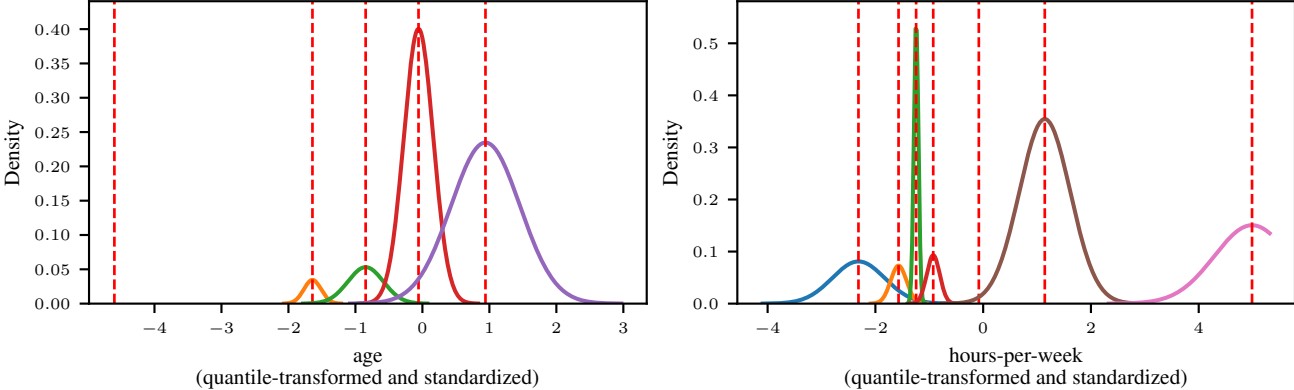

*Figure 5.* Gaussian components found by the DT encoder (max depth = 3) for two features in the `adult` dataset. The red vertical lines indicate the means of the Gaussian components.

### A.5.3. PRACTICAL CONSIDERATIONS

In practice, $\sigma_k^2$ is never exactly zero due to numerical precision. Therefore, if $\sigma_k^2 < \epsilon$, we check empirically whether $\text{Var}[x_{\text{num}}^{(i)}|z^{(i)} = k] = 0$. If this is the case, we treat $\mu_k$ as representing an inflated value. Furthermore, many features may be integers rather than truly continuous. To preserve the ordinal structure, integers are typically modeled as "continuous". For integers with few unique values, a complex encoder may produce a $z^{(i)}$ that recovers all unique values. This is *not* a failure case; it simply means that the low-resolution model already has access to *all* information about that feature, such that the high-resolution model does not need to generate it at all. We can interpret this as a data-informed process of deciding whether to treat an integer-valued feature as discrete versus continuous.

### A.6. Time Schedule Details

### A.6.1. POLYNOMIAL PARAMETERIZATION OF TIME SCHEDULE

We parameterize the feature-specific time schedules using the polynomial form proposed by Sahoo et al. (2024b). Let $f_\phi : \mathbb{R}^m \times [0,1] \to \mathbb{R}^d$, where $d$ is the number of features and $\mathbf{c} \in \mathbb{R}^m$ be a vector with conditioning information. We define $f_\phi$ as

$$f_\phi(\mathbf{c}, t) = \frac{\mathbf{a}_\phi^2(\mathbf{c})}{5}t^5 + \frac{\mathbf{a}_\phi(\mathbf{c})\mathbf{b}_\phi(\mathbf{c})}{2}t^4 + \frac{\mathbf{b}_\phi^2(\mathbf{c}) + 2\mathbf{a}_\phi(\mathbf{c})\mathbf{d}_\phi(\mathbf{c})}{3}t^3 + \mathbf{b}_\phi(\mathbf{c})\mathbf{d}_\phi(\mathbf{c})t^2 + \mathbf{d}_\phi(\mathbf{c})t, \tag{19}$$

where multiplication and division operations are defined element-wise. The parameters $\mathbf{a}_\psi(\mathbf{c}), \mathbf{b}_\psi(\mathbf{c})$ and $\mathbf{d}_\psi(\mathbf{c})$ are outputs of a neural network with parameters $\psi$ that maps $\mathbb{R}^m \to \mathbb{R}^d \to \mathbb{R}^d$ to construct a common embedding which is the input to separate linear layers that map to $\mathbf{a}_\psi(\mathbf{c}), \mathbf{b}_\psi(\mathbf{c})$ and $\mathbf{d}_\psi(\mathbf{c})$, respectively. We restrict $\mathbf{d}_\phi(\mathbf{c}) \geq \epsilon$ and use SiLU activation

functions. We normalize the function output to get

$$\boldsymbol{\gamma}_t(\mathbf{c}) = \frac{f_\phi(\mathbf{c}, t)}{f_\phi(\mathbf{c}, 1)}, \tag{20}$$

such that $\boldsymbol{\gamma}_t(\mathbf{c})$ is monotonically increasing for $t \in [0, 1]$ and has end points $\boldsymbol{\gamma}_0(\mathbf{c}) = 0$ and $\boldsymbol{\gamma}_1(\mathbf{c}) = 1$. Note that its time-derivative $\dot{\boldsymbol{\gamma}}_t(\mathbf{c})$ is available in closed form.

### A.6.2. LEARNED TIME SCHEDULES

Below, we display the feature-specific time schedules $\boldsymbol{\gamma}_t(\mathbf{x}_{\text{low}})$ for each dataset learned by the TabCascade model with DT encoder (one line per feature). Since the time schedule is conditioned on $\mathbf{x}_{\text{low}}$ we picture $\mathbb{E}_{\mathbf{x}_{\text{low}}}[\boldsymbol{\gamma}_t(\mathbf{x}_{\text{low}})]$ (left) and $\text{Var}_{\mathbf{x}_{\text{low}}}[\boldsymbol{\gamma}_t(\mathbf{x}_{\text{low}})]$ (right). While on average a linear time schedule seems beneficial, the model does capture some heterogeneity across features.

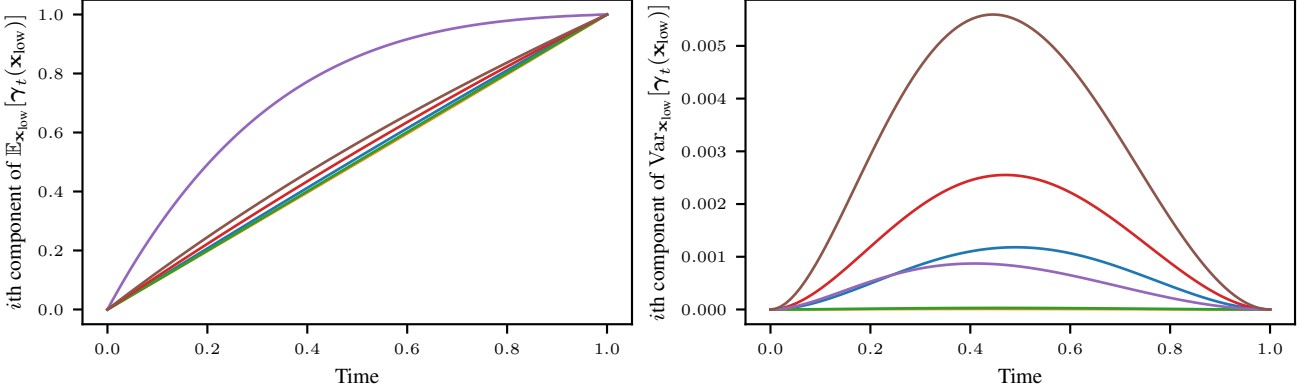

*Figure 6.* Learned time schedule for the `adult` dataset.

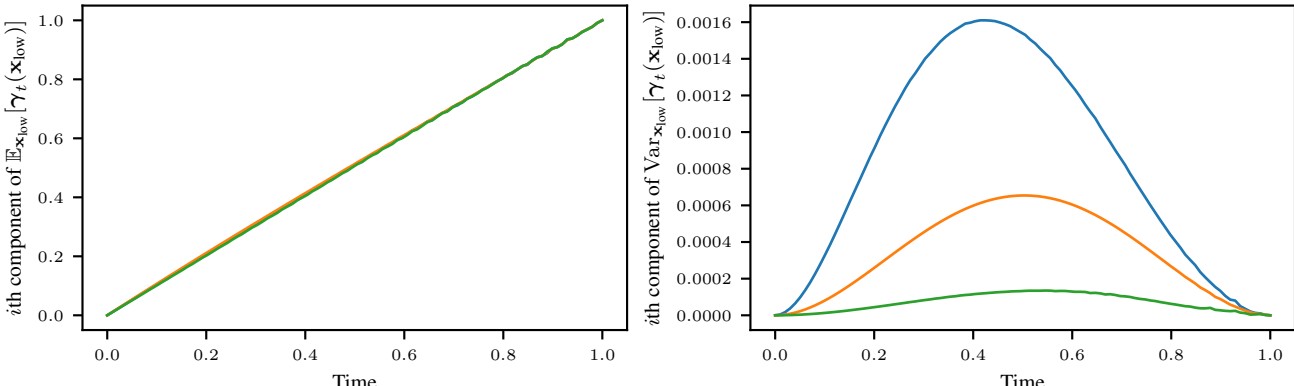

*Figure 7.* Learned time schedule for the `airlines` dataset.

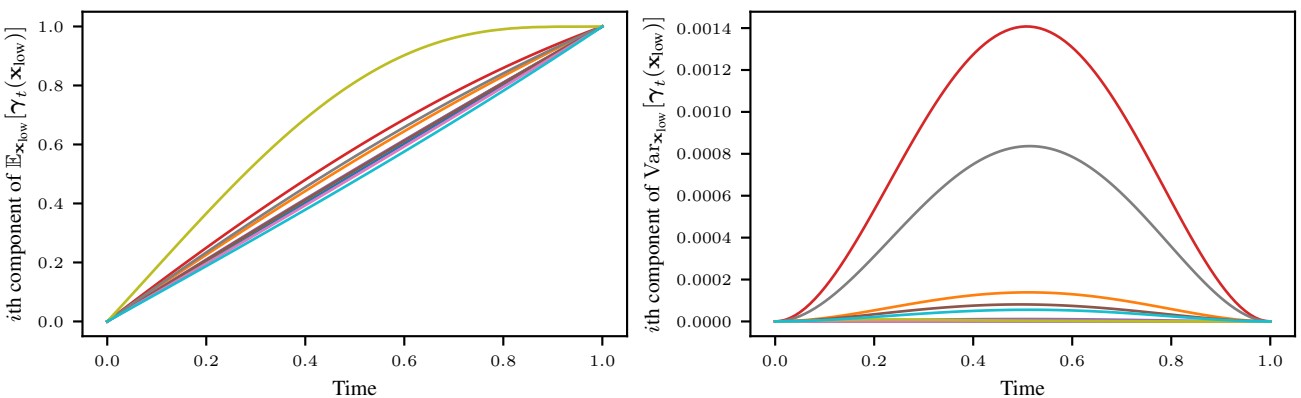

*Figure 8.* Learned time schedule for the `beijing` dataset.

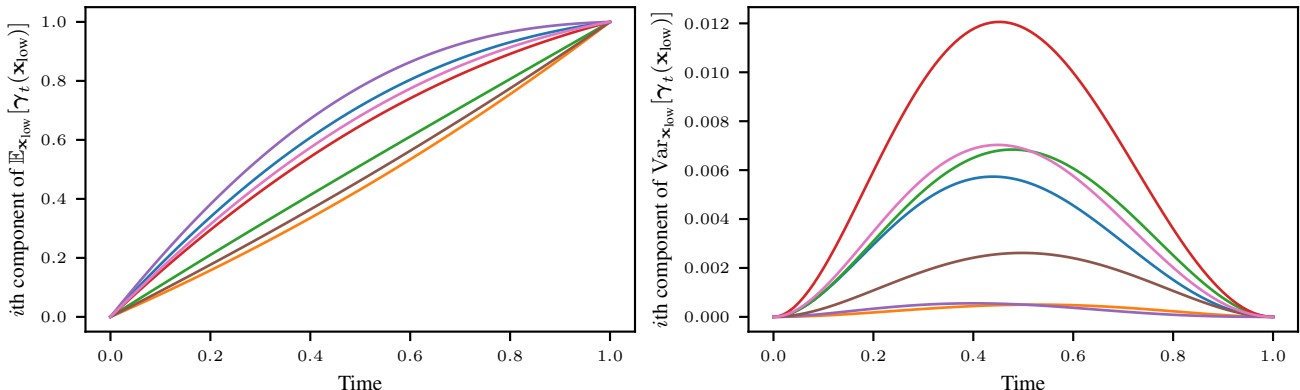

*Figure 9.* Learned time schedule for the `credit_g` dataset.

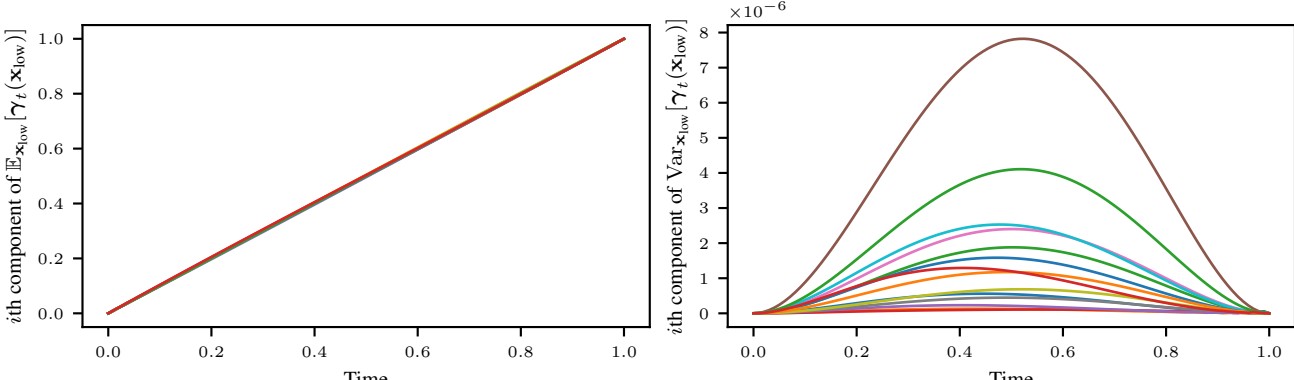

*Figure 10.* Learned time schedule for the `default` dataset.

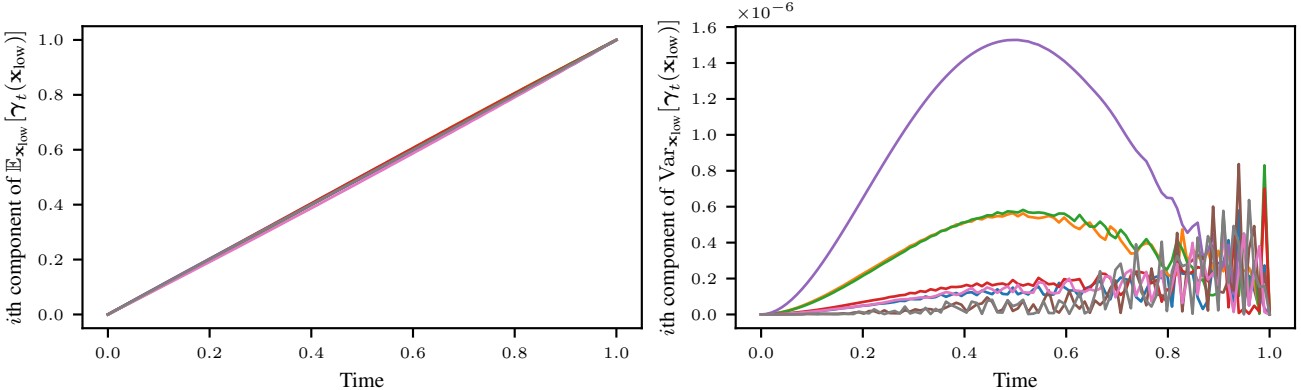

*Figure 11.* Learned time schedule for the `diabetes` dataset.

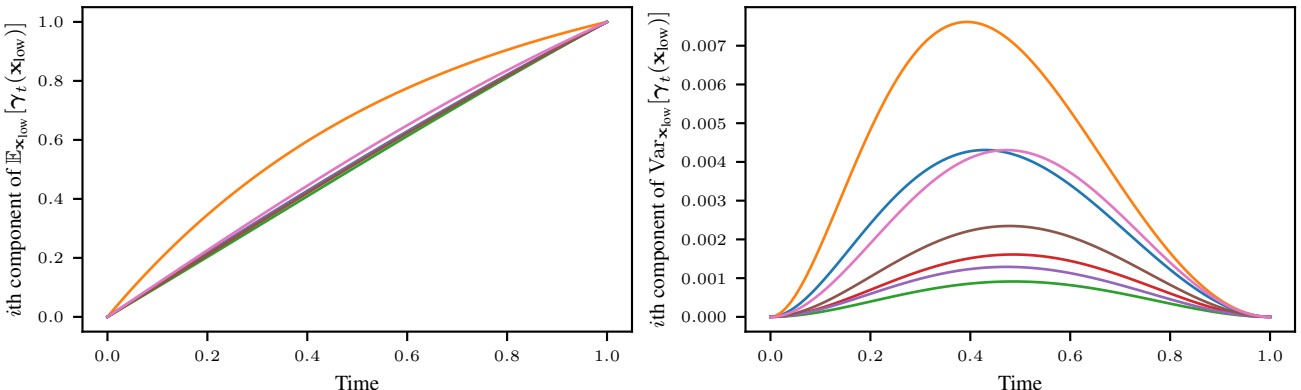

*Figure 12.* Learned time schedule for the `electricity` dataset.

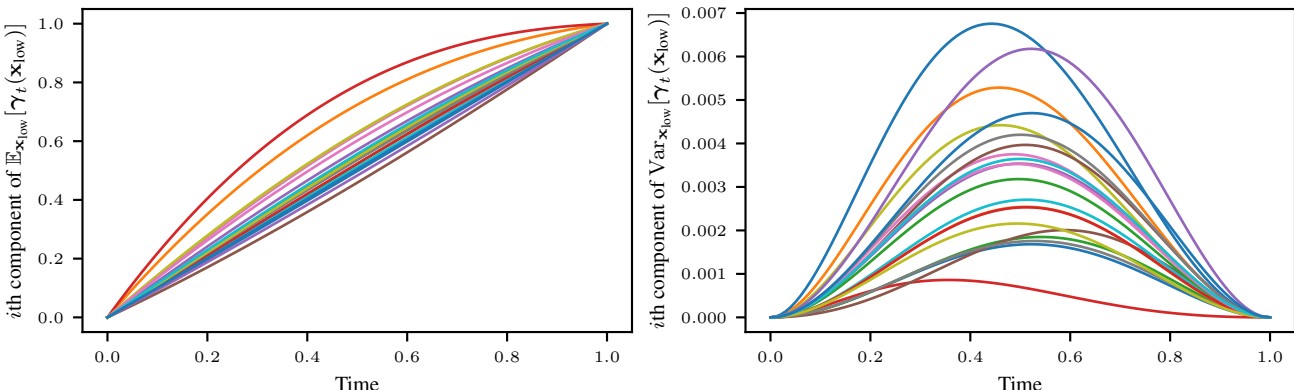

*Figure 13.* Learned time schedule for the `kc1` dataset.

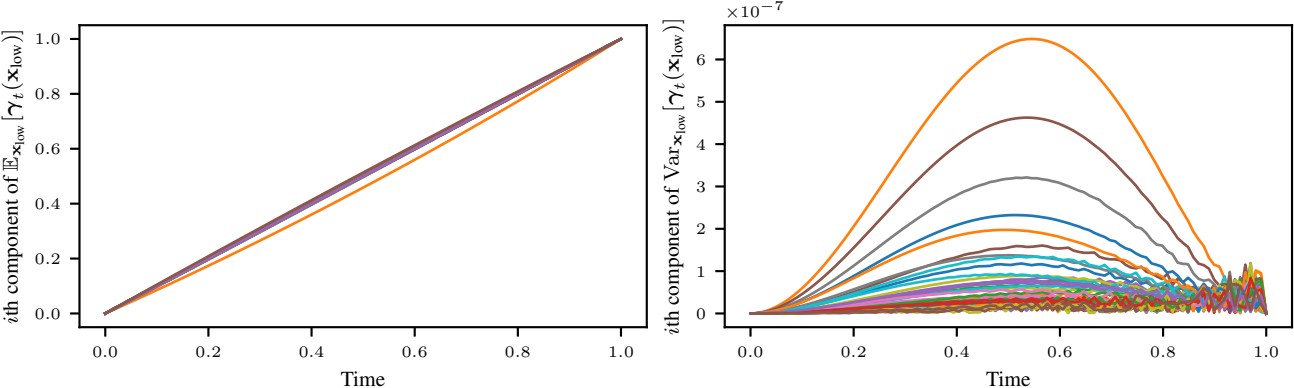

*Figure 14.* Learned time schedule for the `news` dataset.

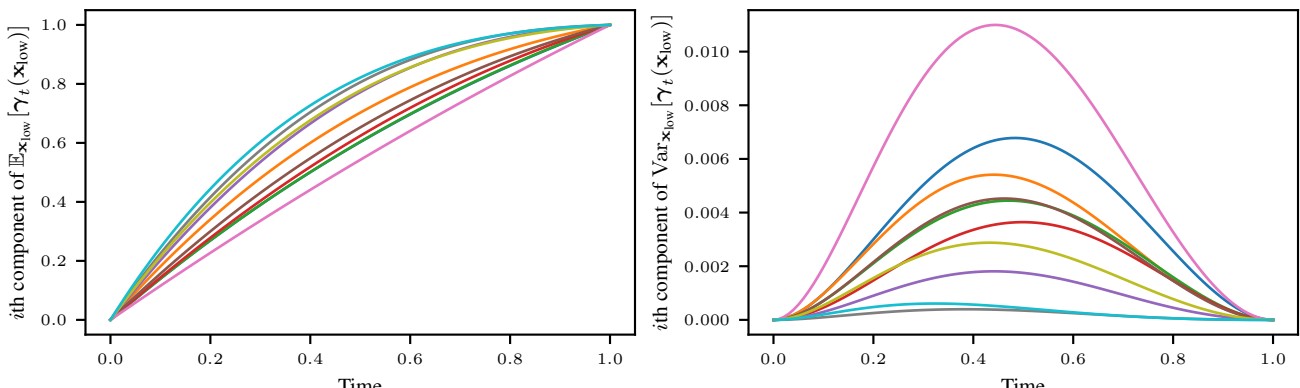

*Figure 15.* Learned time schedule for the `nmes` dataset.

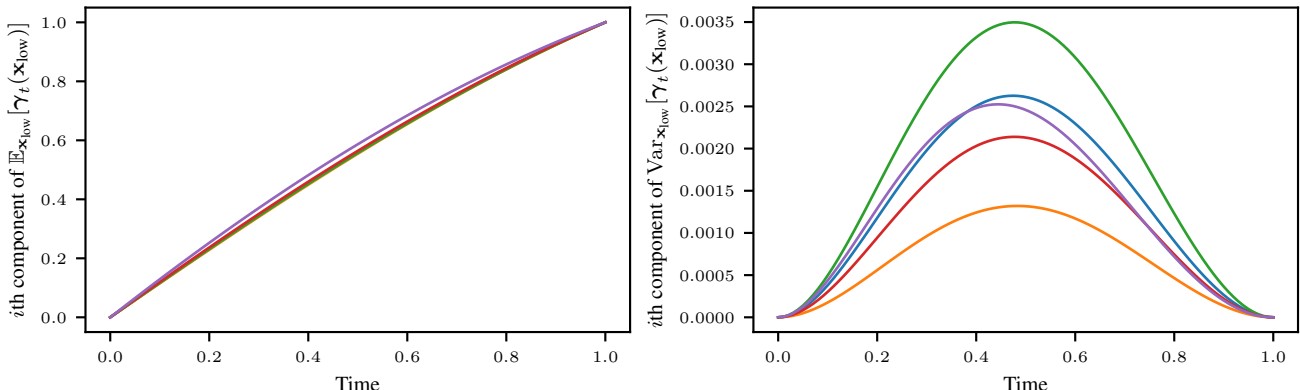

*Figure 16.* Learned time schedule for the `phoneme` dataset.

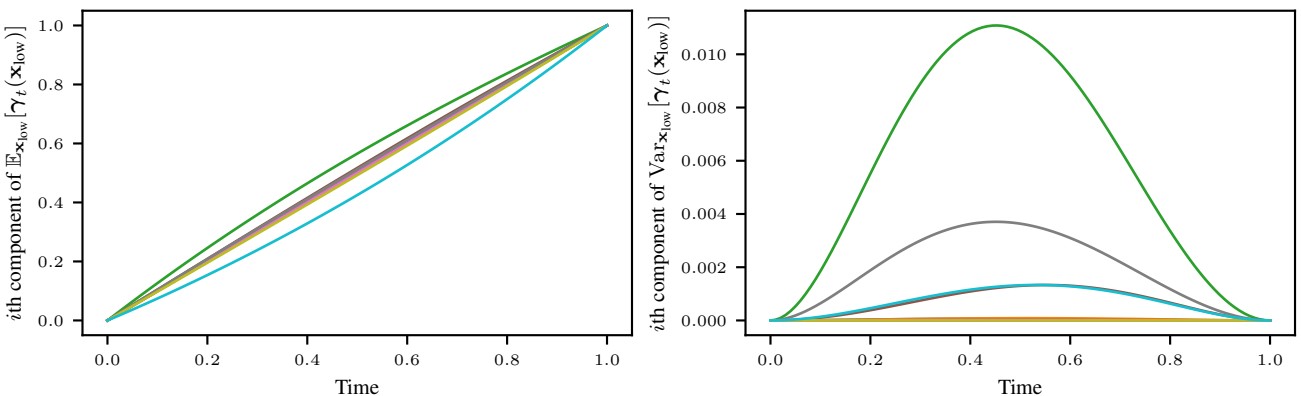

*Figure 17.* Learned time schedule for the `shoppers` dataset.

## A.7. Qualitative Comparisons

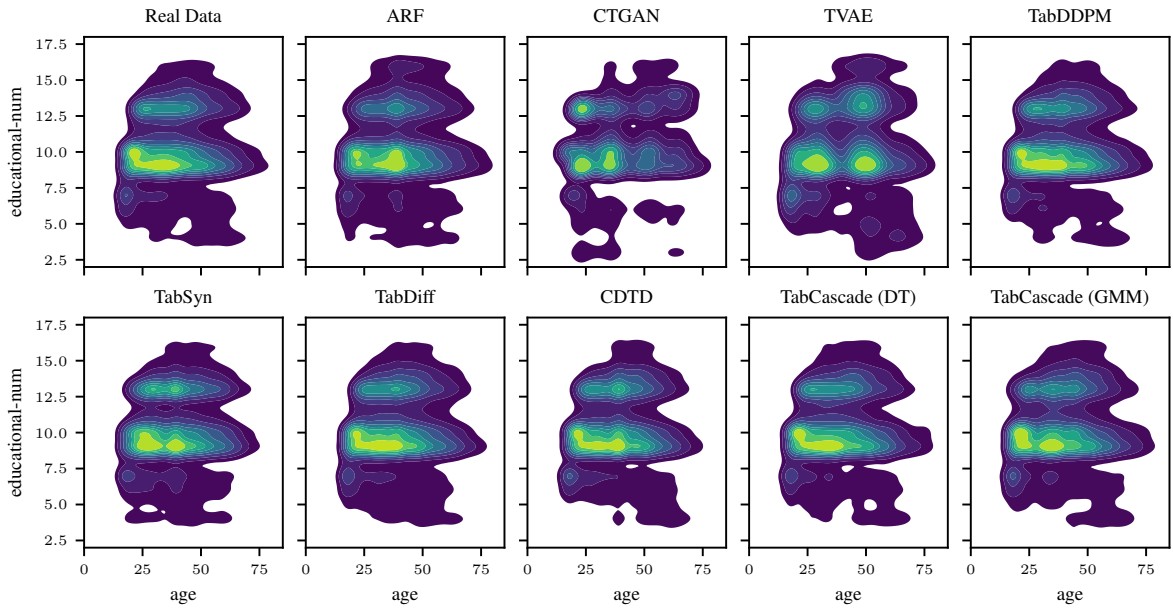

*Figure 18.* Example of bivariate density from the `adult` dataset.

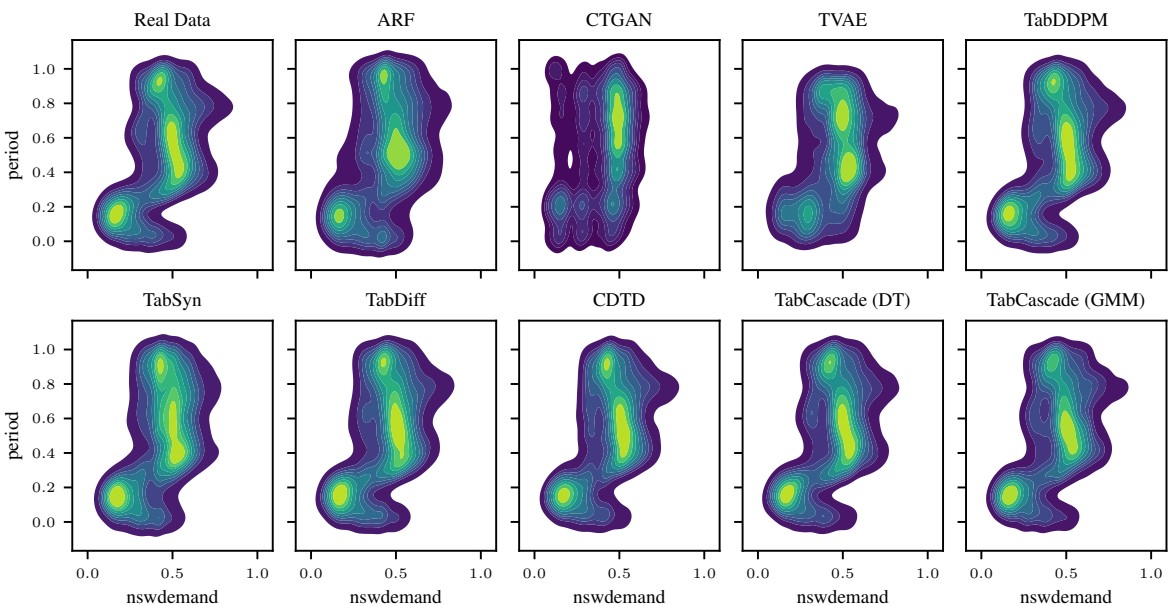

*Figure 19.* Example of bivariate density from the `electricity` dataset.

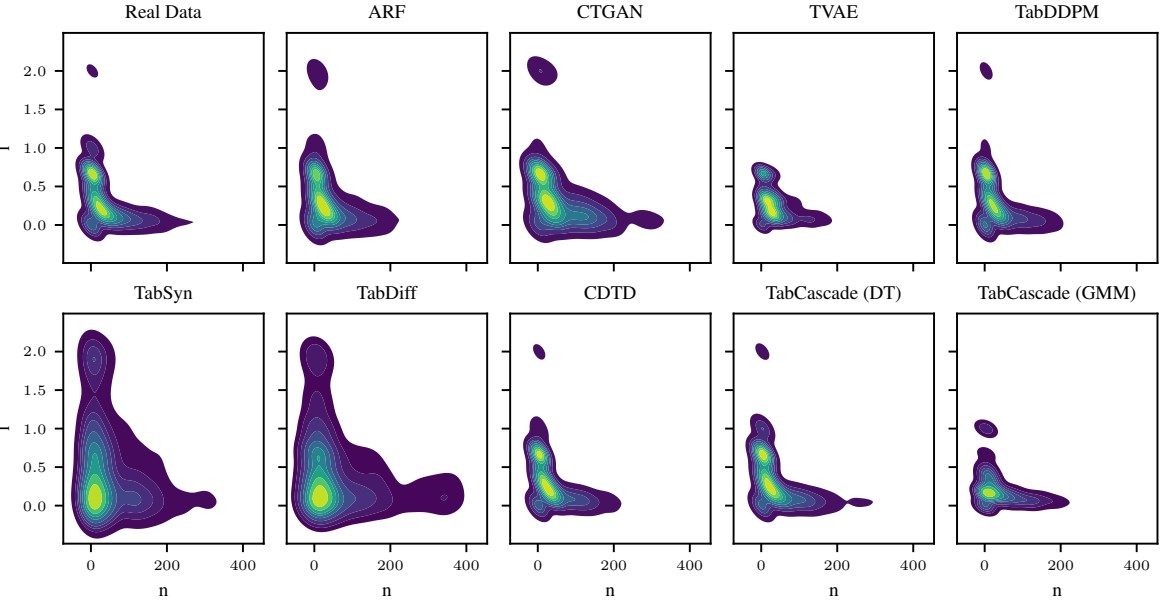

*Figure 20.* Example of bivariate density from the `kc1` dataset.

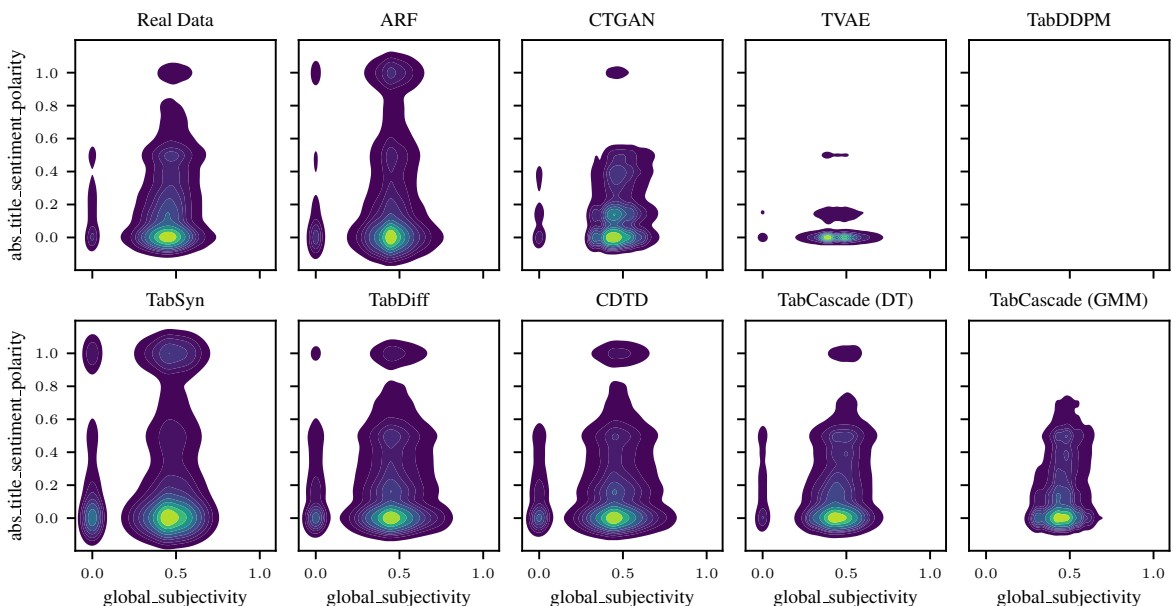

*Figure 21.* Example of bivariate density from the `news` dataset. TabDDPM produces NaNs for this dataset.

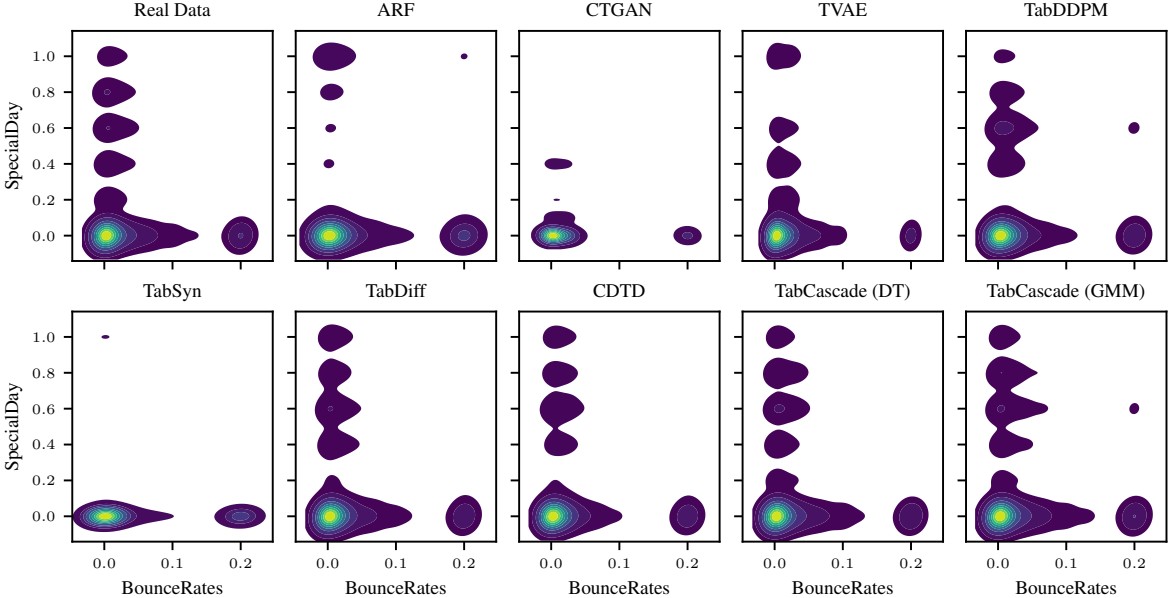

*Figure 22.* Example of bivariate density from the `shoppers` dataset.

## A.8. Detailed Main Results

*Table 4.* Comparison of **Detection scores**. **Bold** indicates the best and underline the second best result. We report the average and standard deviation across three training runs and 10 different generated samples each.

| | ARF | TVAE | CTGAN | TabDDPM | TabSyn | TabDiff | CDTD | Ours (DT) |
|---|---|---|---|---|---|---|---|---|
| adult | $0.350_{\pm 0.011}$ | $0.120_{\pm 0.015}$ | $0.077_{\pm 0.026}$ | $0.725_{\pm 0.013}$ | $0.424_{\pm 0.022}$ | $\underline{0.747}_{\pm 0.014}$ | $0.622_{\pm 0.009}$ | $\mathbf{0.891}_{\pm 0.016}$ |
| airlines | $\mathbf{0.669}_{\pm 0.008}$ | $0.009_{\pm 0.000}$ | $0.012_{\pm 0.003}$ | - | $0.448_{\pm 0.021}$ | $0.060_{\pm 0.003}$ | $0.464_{\pm 0.020}$ | $\underline{0.595}_{\pm 0.143}$ |
| beijing | $0.306_{\pm 0.014}$ | $0.068_{\pm 0.067}$ | $0.091_{\pm 0.017}$ | $0.610_{\pm 0.045}$ | $0.424_{\pm 0.069}$ | $\underline{0.623}_{\pm 0.060}$ | $0.514_{\pm 0.043}$ | $\mathbf{0.765}_{\pm 0.059}$ |
| credit_g | $0.460_{\pm 0.024}$ | $0.773_{\pm 0.048}$ | $0.263_{\pm 0.036}$ | $\mathbf{1.000}_{\pm 0.000}$ | $0.129_{\pm 0.032}$ | $0.488_{\pm 0.020}$ | $0.990_{\pm 0.016}$ | $\underline{1.000}_{\pm 0.003}$ |
| default | $0.052_{\pm 0.004}$ | $0.038_{\pm 0.006}$ | $0.022_{\pm 0.006}$ | $0.225_{\pm 0.004}$ | $0.027_{\pm 0.004}$ | $\underline{0.227}_{\pm 0.023}$ | $0.190_{\pm 0.008}$ | $\mathbf{0.579}_{\pm 0.009}$ |
| diabetes | $0.288_{\pm 0.009}$ | $0.005_{\pm 0.004}$ | $0.090_{\pm 0.041}$ | - | $0.090_{\pm 0.004}$ | $\underline{0.430}_{\pm 0.005}$ | $0.310_{\pm 0.052}$ | $\mathbf{0.654}_{\pm 0.030}$ |
| electricity | $0.255_{\pm 0.007}$ | $0.130_{\pm 0.072}$ | $0.045_{\pm 0.012}$ | $0.551_{\pm 0.004}$ | $0.351_{\pm 0.035}$ | $\underline{0.629}_{\pm 0.008}$ | $0.491_{\pm 0.007}$ | $\mathbf{0.847}_{\pm 0.011}$ |
| kc1 | $0.085_{\pm 0.012}$ | $0.058_{\pm 0.011}$ | $0.020_{\pm 0.004}$ | $0.419_{\pm 0.029}$ | $0.003_{\pm 0.001}$ | $0.004_{\pm 0.001}$ | $\underline{0.440}_{\pm 0.030}$ | $\mathbf{0.974}_{\pm 0.027}$ |
| news | $0.000_{\pm 0.000}$ | $0.001_{\pm 0.000}$ | $0.000_{\pm 0.000}$ | - | $0.000_{\pm 0.000}$ | $\underline{0.005}_{\pm 0.001}$ | $0.002_{\pm 0.001}$ | $\mathbf{0.202}_{\pm 0.018}$ |
| nmes | $0.391_{\pm 0.010}$ | $0.641_{\pm 0.045}$ | $0.171_{\pm 0.017}$ | $\underline{0.873}_{\pm 0.015}$ | $0.249_{\pm 0.052}$ | $0.667_{\pm 0.024}$ | $0.864_{\pm 0.024}$ | $\mathbf{0.982}_{\pm 0.015}$ |
| phoneme | $0.446_{\pm 0.020}$ | $0.328_{\pm 0.039}$ | $0.077_{\pm 0.009}$ | $\underline{0.998}_{\pm 0.004}$ | $0.202_{\pm 0.042}$ | $0.845_{\pm 0.022}$ | $0.934_{\pm 0.027}$ | $\mathbf{1.000}_{\pm 0.000}$ |
| shoppers | $0.210_{\pm 0.007}$ | $0.293_{\pm 0.010}$ | $0.065_{\pm 0.011}$ | $0.327_{\pm 0.008}$ | $0.075_{\pm 0.037}$ | $\underline{0.431}_{\pm 0.025}$ | $0.400_{\pm 0.013}$ | $\mathbf{0.961}_{\pm 0.028}$ |

*Table 5.* Comparison of **Shape scores**. **Bold** indicates the best and underline the second best result. We report the average and standard deviation across three training runs and 10 different generated samples each.

| | ARF | TVAE | CTGAN | TabDDPM | TabSyn | TabDiff | CDTD | Ours (DT) |
|---|---|---|---|---|---|---|---|---|
| adult | $0.985_{\pm 0.000}$ | $0.893_{\pm 0.008}$ | $0.902_{\pm 0.012}$ | $0.983_{\pm 0.001}$ | $0.972_{\pm 0.003}$ | $\mathbf{0.991}_{\pm 0.001}$ | $0.984_{\pm 0.000}$ | $\underline{0.989}_{\pm 0.001}$ |
| airlines | $\mathbf{0.986}_{\pm 0.000}$ | $0.754_{\pm 0.014}$ | $0.795_{\pm 0.010}$ | - | $0.946_{\pm 0.005}$ | $0.835_{\pm 0.004}$ | $0.949_{\pm 0.003}$ | $\underline{0.966}_{\pm 0.018}$ |
| beijing | $0.957_{\pm 0.001}$ | $0.895_{\pm 0.030}$ | $0.916_{\pm 0.003}$ | $0.981_{\pm 0.002}$ | $0.969_{\pm 0.004}$ | $\underline{0.984}_{\pm 0.003}$ | $0.973_{\pm 0.002}$ | $\mathbf{0.989}_{\pm 0.002}$ |
| credit_g | $0.954_{\pm 0.002}$ | $0.943_{\pm 0.007}$ | $0.875_{\pm 0.007}$ | $0.974_{\pm 0.002}$ | $0.888_{\pm 0.016}$ | $0.945_{\pm 0.003}$ | $\underline{0.975}_{\pm 0.003}$ | $\mathbf{0.977}_{\pm 0.002}$ |
| default | $0.948_{\pm 0.001}$ | $0.905_{\pm 0.007}$ | $0.908_{\pm 0.012}$ | $0.968_{\pm 0.001}$ | $0.938_{\pm 0.005}$ | $\underline{0.975}_{\pm 0.003}$ | $0.963_{\pm 0.002}$ | $\mathbf{0.985}_{\pm 0.002}$ |
| diabetes | $\underline{0.978}_{\pm 0.000}$ | $0.869_{\pm 0.012}$ | $0.925_{\pm 0.012}$ | - | $0.917_{\pm 0.005}$ | $0.969_{\pm 0.001}$ | $0.968_{\pm 0.004}$ | $\mathbf{0.986}_{\pm 0.002}$ |
| electricity | $0.964_{\pm 0.002}$ | $0.902_{\pm 0.021}$ | $0.885_{\pm 0.008}$ | $0.986_{\pm 0.001}$ | $0.976_{\pm 0.003}$ | $\underline{0.989}_{\pm 0.001}$ | $0.981_{\pm 0.001}$ | $\mathbf{0.991}_{\pm 0.001}$ |
| kc1 | $0.937_{\pm 0.002}$ | $0.871_{\pm 0.008}$ | $0.913_{\pm 0.011}$ | $0.960_{\pm 0.005}$ | $0.849_{\pm 0.013}$ | $0.880_{\pm 0.006}$ | $\underline{0.967}_{\pm 0.004}$ | $\mathbf{0.978}_{\pm 0.005}$ |
| news | $0.929_{\pm 0.001}$ | $0.883_{\pm 0.018}$ | $0.944_{\pm 0.001}$ | - | $0.883_{\pm 0.011}$ | $\underline{0.952}_{\pm 0.001}$ | $0.951_{\pm 0.002}$ | $\mathbf{0.980}_{\pm 0.001}$ |
| nmes | $0.939_{\pm 0.002}$ | $0.958_{\pm 0.007}$ | $0.899_{\pm 0.010}$ | $0.971_{\pm 0.001}$ | $0.924_{\pm 0.014}$ | $0.978_{\pm 0.001}$ | $\underline{0.980}_{\pm 0.001}$ | $\mathbf{0.990}_{\pm 0.001}$ |
| phoneme | $0.964_{\pm 0.003}$ | $0.938_{\pm 0.009}$ | $0.858_{\pm 0.022}$ | $\underline{0.984}_{\pm 0.002}$ | $0.947_{\pm 0.007}$ | $0.978_{\pm 0.002}$ | $0.982_{\pm 0.002}$ | $\mathbf{0.985}_{\pm 0.002}$ |
| shoppers | $0.950_{\pm 0.001}$ | $0.934_{\pm 0.010}$ | $0.909_{\pm 0.003}$ | $0.947_{\pm 0.003}$ | $0.912_{\pm 0.013}$ | $\underline{0.978}_{\pm 0.001}$ | $0.969_{\pm 0.002}$ | $\mathbf{0.987}_{\pm 0.001}$ |

*Table 6.* Comparison of **Shape (test) scores** computed based on the test data. **Bold** indicates the best and underline the second best result. We report the average and standard deviation across three training runs and 10 different generated samples each.

| | ARF | TVAE | CTGAN | TabDDPM | TabSyn | TabDiff | CDTD | Ours (DT) |
|---|---|---|---|---|---|---|---|---|
| adult | $0.982_{\pm 0.001}$ | $0.894_{\pm 0.007}$ | $0.902_{\pm 0.012}$ | $0.980_{\pm 0.002}$ | $0.971_{\pm 0.004}$ | $\mathbf{0.987}_{\pm 0.001}$ | $0.981_{\pm 0.001}$ | $\underline{0.986}_{\pm 0.001}$ |
| airlines | $\mathbf{0.984}_{\pm 0.001}$ | $0.754_{\pm 0.014}$ | $0.794_{\pm 0.010}$ | - | $0.945_{\pm 0.005}$ | $0.835_{\pm 0.004}$ | $0.947_{\pm 0.002}$ | $\underline{0.965}_{\pm 0.018}$ |
| beijing | $0.956_{\pm 0.002}$ | $0.895_{\pm 0.030}$ | $0.916_{\pm 0.002}$ | $0.980_{\pm 0.002}$ | $0.970_{\pm 0.005}$ | $\underline{0.983}_{\pm 0.003}$ | $0.972_{\pm 0.003}$ | $\mathbf{0.986}_{\pm 0.002}$ |
| credit_g | $0.932_{\pm 0.003}$ | $0.928_{\pm 0.004}$ | $0.871_{\pm 0.007}$ | $0.946_{\pm 0.005}$ | $0.879_{\pm 0.018}$ | $0.923_{\pm 0.005}$ | $\underline{0.948}_{\pm 0.004}$ | $\mathbf{0.951}_{\pm 0.005}$ |
| default | $0.944_{\pm 0.002}$ | $0.905_{\pm 0.007}$ | $0.906_{\pm 0.012}$ | $0.967_{\pm 0.002}$ | $0.935_{\pm 0.005}$ | $\underline{0.972}_{\pm 0.003}$ | $0.962_{\pm 0.002}$ | $\mathbf{0.982}_{\pm 0.001}$ |
| diabetes | $\underline{0.975}_{\pm 0.000}$ | $0.868_{\pm 0.012}$ | $0.924_{\pm 0.012}$ | - | $0.917_{\pm 0.004}$ | $0.968_{\pm 0.000}$ | $0.967_{\pm 0.004}$ | $\mathbf{0.984}_{\pm 0.002}$ |
| electricity | $0.962_{\pm 0.002}$ | $0.902_{\pm 0.023}$ | $0.883_{\pm 0.009}$ | $0.982_{\pm 0.001}$ | $0.973_{\pm 0.003}$ | $\underline{0.985}_{\pm 0.001}$ | $0.977_{\pm 0.002}$ | $\mathbf{0.987}_{\pm 0.001}$ |
| kc1 | $0.928_{\pm 0.004}$ | $0.857_{\pm 0.011}$ | $0.900_{\pm 0.010}$ | $0.950_{\pm 0.009}$ | $0.853_{\pm 0.013}$ | $0.882_{\pm 0.004}$ | $\underline{0.953}_{\pm 0.007}$ | $\mathbf{0.956}_{\pm 0.010}$ |
| news | $0.927_{\pm 0.000}$ | $0.883_{\pm 0.017}$ | $0.942_{\pm 0.002}$ | - | $0.882_{\pm 0.010}$ | $\underline{0.951}_{\pm 0.001}$ | $0.950_{\pm 0.002}$ | $\mathbf{0.978}_{\pm 0.001}$ |
| nmes | $0.934_{\pm 0.006}$ | $0.954_{\pm 0.009}$ | $0.902_{\pm 0.006}$ | $0.966_{\pm 0.002}$ | $0.921_{\pm 0.010}$ | $0.970_{\pm 0.003}$ | $\underline{0.971}_{\pm 0.002}$ | $\mathbf{0.980}_{\pm 0.002}$ |
| phoneme | $0.952_{\pm 0.006}$ | $0.932_{\pm 0.005}$ | $0.855_{\pm 0.012}$ | $0.968_{\pm 0.006}$ | $0.938_{\pm 0.008}$ | $0.965_{\pm 0.005}$ | $\underline{0.969}_{\pm 0.005}$ | $\mathbf{0.971}_{\pm 0.005}$ |
| shoppers | $0.945_{\pm 0.002}$ | $0.934_{\pm 0.011}$ | $0.907_{\pm 0.003}$ | $0.945_{\pm 0.003}$ | $0.909_{\pm 0.012}$ | $\underline{0.972}_{\pm 0.002}$ | $0.965_{\pm 0.004}$ | $\mathbf{0.980}_{\pm 0.001}$ |

*Table 7.* Comparison of **Shape (cat) scores**, which evaluate categorical univariate densities only. **Bold** indicates the best and underline the second best result. We report the average and standard deviation across three training runs and 10 different generated samples each.

| | ARF | TVAE | CTGAN | TabDDPM | TabSyn | TabDiff | CDTD | Ours (DT) |
|---|---|---|---|---|---|---|---|---|
| adult | $\mathbf{0.996}_{\pm 0.000}$ | $0.896_{\pm 0.004}$ | $0.893_{\pm 0.008}$ | $0.981_{\pm 0.002}$ | $0.975_{\pm 0.008}$ | $\underline{0.995}_{\pm 0.001}$ | $0.988_{\pm 0.001}$ | $0.989_{\pm 0.001}$ |
| airlines | $\mathbf{0.992}_{\pm 0.000}$ | $0.693_{\pm 0.008}$ | $0.756_{\pm 0.024}$ | - | $0.946_{\pm 0.004}$ | $0.780_{\pm 0.007}$ | $0.932_{\pm 0.004}$ | $\underline{0.950}_{\pm 0.029}$ |
| beijing | $\mathbf{0.996}_{\pm 0.002}$ | $0.839_{\pm 0.022}$ | $0.912_{\pm 0.022}$ | $0.988_{\pm 0.002}$ | $0.990_{\pm 0.006}$ | $\underline{0.995}_{\pm 0.002}$ | $0.994_{\pm 0.002}$ | $0.995_{\pm 0.002}$ |
| credit_g | $0.979_{\pm 0.002}$ | $0.949_{\pm 0.008}$ | $0.872_{\pm 0.009}$ | $0.973_{\pm 0.003}$ | $0.906_{\pm 0.019}$ | $0.973_{\pm 0.003}$ | $\mathbf{0.980}_{\pm 0.003}$ | $\underline{0.979}_{\pm 0.002}$ |
| default | $\mathbf{0.996}_{\pm 0.001}$ | $0.883_{\pm 0.025}$ | $0.899_{\pm 0.017}$ | $0.978_{\pm 0.002}$ | $0.949_{\pm 0.005}$ | $\underline{0.992}_{\pm 0.003}$ | $0.987_{\pm 0.003}$ | $0.987_{\pm 0.003}$ |
| diabetes | $\mathbf{0.996}_{\pm 0.000}$ | $0.875_{\pm 0.012}$ | $0.929_{\pm 0.010}$ | - | $0.916_{\pm 0.004}$ | $0.969_{\pm 0.001}$ | $0.982_{\pm 0.002}$ | $\underline{0.986}_{\pm 0.002}$ |
| electricity | $\mathbf{0.996}_{\pm 0.001}$ | $0.917_{\pm 0.038}$ | $0.873_{\pm 0.026}$ | $0.994_{\pm 0.002}$ | $0.993_{\pm 0.002}$ | $\underline{0.995}_{\pm 0.001}$ | $0.995_{\pm 0.002}$ | $0.994_{\pm 0.001}$ |
| kc1 | $\mathbf{0.993}_{\pm 0.004}$ | $0.980_{\pm 0.013}$ | $0.943_{\pm 0.049}$ | $0.958_{\pm 0.011}$ | $0.926_{\pm 0.048}$ | $0.991_{\pm 0.006}$ | $\underline{0.992}_{\pm 0.007}$ | $0.991_{\pm 0.005}$ |
| news | $\mathbf{0.998}_{\pm 0.000}$ | $0.888_{\pm 0.009}$ | $0.988_{\pm 0.002}$ | - | $0.941_{\pm 0.020}$ | $\underline{0.997}_{\pm 0.001}$ | $0.990_{\pm 0.001}$ | $0.993_{\pm 0.000}$ |
| nmes | $\underline{0.993}_{\pm 0.002}$ | $0.970_{\pm 0.009}$ | $0.900_{\pm 0.020}$ | $0.969_{\pm 0.002}$ | $0.945_{\pm 0.025}$ | $\mathbf{0.993}_{\pm 0.002}$ | $0.993_{\pm 0.002}$ | $0.993_{\pm 0.001}$ |
| phoneme | $0.992_{\pm 0.006}$ | $0.991_{\pm 0.005}$ | $0.898_{\pm 0.011}$ | $0.994_{\pm 0.004}$ | $0.994_{\pm 0.004}$ | $\mathbf{0.995}_{\pm 0.003}$ | $\underline{0.994}_{\pm 0.005}$ | $0.993_{\pm 0.004}$ |
| shoppers | $\mathbf{0.992}_{\pm 0.001}$ | $0.952_{\pm 0.008}$ | $0.902_{\pm 0.014}$ | $0.939_{\pm 0.007}$ | $0.916_{\pm 0.038}$ | $\underline{0.991}_{\pm 0.001}$ | $0.989_{\pm 0.001}$ | $0.984_{\pm 0.002}$ |

*Table 8.* Comparison of **Shape (num) scores**, which evaluates numerical univariate densities only. **Bold** indicates the best and underline the second best result. We report the average and standard deviation across three training runs and 10 different generated samples each.

| | ARF | TVAE | CTGAN | TabDDPM | TabSyn | TabDiff | CDTD | Ours (DT) |
|---|---|---|---|---|---|---|---|---|
| adult | $0.969_{\pm 0.001}$ | $0.890_{\pm 0.016}$ | $0.915_{\pm 0.024}$ | $\underline{0.985}_{\pm 0.001}$ | $0.968_{\pm 0.006}$ | $0.984_{\pm 0.001}$ | $0.978_{\pm 0.001}$ | $\mathbf{0.989}_{\pm 0.002}$ |
| airlines | $0.977_{\pm 0.001}$ | $0.856_{\pm 0.047}$ | $0.859_{\pm 0.014}$ | - | $0.945_{\pm 0.013}$ | $0.927_{\pm 0.003}$ | $\underline{0.977}_{\pm 0.002}$ | $\mathbf{0.993}_{\pm 0.001}$ |
| beijing | $0.953_{\pm 0.001}$ | $0.901_{\pm 0.032}$ | $0.916_{\pm 0.004}$ | $0.980_{\pm 0.003}$ | $0.967_{\pm 0.005}$ | $\underline{0.983}_{\pm 0.003}$ | $0.971_{\pm 0.002}$ | $\mathbf{0.988}_{\pm 0.003}$ |
| credit_g | $0.906_{\pm 0.005}$ | $0.932_{\pm 0.008}$ | $0.879_{\pm 0.017}$ | $\mathbf{0.975}_{\pm 0.004}$ | $0.850_{\pm 0.012}$ | $0.889_{\pm 0.007}$ | $0.965_{\pm 0.005}$ | $\underline{0.973}_{\pm 0.004}$ |
| default | $0.914_{\pm 0.002}$ | $0.921_{\pm 0.008}$ | $0.914_{\pm 0.009}$ | $0.960_{\pm 0.001}$ | $0.929_{\pm 0.005}$ | $\underline{0.962}_{\pm 0.002}$ | $0.946_{\pm 0.002}$ | $\mathbf{0.984}_{\pm 0.001}$ |
| diabetes | $0.913_{\pm 0.001}$ | $0.847_{\pm 0.011}$ | $0.911_{\pm 0.018}$ | - | $0.921_{\pm 0.011}$ | $\underline{0.971}_{\pm 0.002}$ | $0.918_{\pm 0.016}$ | $\mathbf{0.986}_{\pm 0.001}$ |
| electricity | $0.954_{\pm 0.002}$ | $0.898_{\pm 0.020}$ | $0.888_{\pm 0.002}$ | $0.984_{\pm 0.001}$ | $0.972_{\pm 0.004}$ | $\underline{0.987}_{\pm 0.001}$ | $0.977_{\pm 0.001}$ | $\mathbf{0.989}_{\pm 0.001}$ |
| kc1 | $0.934_{\pm 0.002}$ | $0.866_{\pm 0.009}$ | $0.911_{\pm 0.010}$ | $0.961_{\pm 0.005}$ | $0.846_{\pm 0.012}$ | $0.875_{\pm 0.006}$ | $\underline{0.965}_{\pm 0.004}$ | $\mathbf{0.978}_{\pm 0.006}$ |
| news | $0.908_{\pm 0.001}$ | $0.881_{\pm 0.020}$ | $0.931_{\pm 0.002}$ | - | $0.865_{\pm 0.013}$ | $0.939_{\pm 0.001}$ | $\underline{0.940}_{\pm 0.002}$ | $\mathbf{0.976}_{\pm 0.001}$ |
| nmes | $0.891_{\pm 0.003}$ | $0.947_{\pm 0.005}$ | $0.899_{\pm 0.005}$ | $\underline{0.973}_{\pm 0.001}$ | $0.905_{\pm 0.008}$ | $0.964_{\pm 0.002}$ | $0.969_{\pm 0.002}$ | $\mathbf{0.987}_{\pm 0.002}$ |
| phoneme | $0.959_{\pm 0.003}$ | $0.928_{\pm 0.011}$ | $0.850_{\pm 0.024}$ | $\underline{0.982}_{\pm 0.002}$ | $0.937_{\pm 0.008}$ | $0.975_{\pm 0.002}$ | $0.979_{\pm 0.002}$ | $\mathbf{0.984}_{\pm 0.002}$ |
| shoppers | $0.917_{\pm 0.002}$ | $0.920_{\pm 0.012}$ | $0.914_{\pm 0.008}$ | $0.952_{\pm 0.003}$ | $0.908_{\pm 0.015}$ | $\underline{0.967}_{\pm 0.002}$ | $0.954_{\pm 0.004}$ | $\mathbf{0.988}_{\pm 0.001}$ |

*Table 9.* Comparison of **Wasserstein (WD) distances**, which we use to evaluate numerical univariate densities only. **Bold** indicates the best and underline the second best result. We report the average and standard deviation across three training runs and 10 different generated samples each.

| | ARF | TVAE | CTGAN | TabDDPM | TabSyn | TabDiff | CDTD | Ours (DT) |
|---|---|---|---|---|---|---|---|---|
| adult | $0.007_{\pm 0.000}$ | $0.020_{\pm 0.003}$ | $0.019_{\pm 0.001}$ | $\underline{0.002}_{\pm 0.000}$ | $0.007_{\pm 0.002}$ | $\mathbf{0.002}_{\pm 0.000}$ | $0.005_{\pm 0.000}$ | $0.002_{\pm 0.000}$ |
| airlines | $\underline{0.003}_{\pm 0.000}$ | $0.048_{\pm 0.015}$ | $0.041_{\pm 0.005}$ | - | $0.013_{\pm 0.005}$ | $0.021_{\pm 0.004}$ | $0.003_{\pm 0.000}$ | $\mathbf{0.002}_{\pm 0.000}$ |
| beijing | $0.017_{\pm 0.000}$ | $0.034_{\pm 0.010}$ | $0.028_{\pm 0.002}$ | $0.005_{\pm 0.001}$ | $0.009_{\pm 0.001}$ | $\underline{0.005}_{\pm 0.001}$ | $0.008_{\pm 0.000}$ | $\mathbf{0.003}_{\pm 0.001}$ |
| credit_g | $0.051_{\pm 0.003}$ | $0.027_{\pm 0.003}$ | $0.059_{\pm 0.013}$ | $\mathbf{0.011}_{\pm 0.002}$ | $0.099_{\pm 0.014}$ | $0.059_{\pm 0.004}$ | $0.015_{\pm 0.003}$ | $\underline{0.011}_{\pm 0.002}$ |
| default | $0.007_{\pm 0.000}$ | $0.009_{\pm 0.002}$ | $0.007_{\pm 0.002}$ | $0.003_{\pm 0.000}$ | $0.007_{\pm 0.001}$ | $\underline{0.002}_{\pm 0.000}$ | $0.005_{\pm 0.000}$ | $\mathbf{0.002}_{\pm 0.000}$ |
| diabetes | $0.021_{\pm 0.001}$ | $0.026_{\pm 0.003}$ | $0.019_{\pm 0.004}$ | - | $0.025_{\pm 0.010}$ | $\underline{0.007}_{\pm 0.000}$ | $0.025_{\pm 0.006}$ | $\mathbf{0.004}_{\pm 0.000}$ |
| electricity | $0.007_{\pm 0.000}$ | $0.028_{\pm 0.007}$ | $0.027_{\pm 0.002}$ | $0.003_{\pm 0.000}$ | $0.006_{\pm 0.000}$ | $\mathbf{0.002}_{\pm 0.000}$ | $0.004_{\pm 0.000}$ | $\underline{0.002}_{\pm 0.000}$ |
| kc1 | $0.013_{\pm 0.001}$ | $0.019_{\pm 0.002}$ | $0.015_{\pm 0.001}$ | $0.007_{\pm 0.001}$ | $0.083_{\pm 0.010}$ | $0.061_{\pm 0.005}$ | $\underline{0.005}_{\pm 0.001}$ | $\mathbf{0.005}_{\pm 0.001}$ |
| news | $0.023_{\pm 0.000}$ | $0.026_{\pm 0.007}$ | $\underline{0.013}_{\pm 0.001}$ | - | $0.054_{\pm 0.003}$ | $0.017_{\pm 0.001}$ | $0.014_{\pm 0.001}$ | $\mathbf{0.005}_{\pm 0.000}$ |
| nmes | $0.021_{\pm 0.001}$ | $0.009_{\pm 0.001}$ | $0.019_{\pm 0.001}$ | $\underline{0.007}_{\pm 0.000}$ | $0.022_{\pm 0.004}$ | $0.007_{\pm 0.000}$ | $0.007_{\pm 0.001}$ | $\mathbf{0.004}_{\pm 0.001}$ |
| phoneme | $0.006_{\pm 0.001}$ | $0.019_{\pm 0.004}$ | $0.046_{\pm 0.009}$ | $\underline{0.004}_{\pm 0.000}$ | $0.015_{\pm 0.004}$ | $0.005_{\pm 0.000}$ | $0.004_{\pm 0.001}$ | $\mathbf{0.004}_{\pm 0.001}$ |
| shoppers | $0.015_{\pm 0.000}$ | $0.014_{\pm 0.003}$ | $0.016_{\pm 0.002}$ | $0.008_{\pm 0.001}$ | $0.032_{\pm 0.023}$ | $\underline{0.004}_{\pm 0.001}$ | $0.008_{\pm 0.001}$ | $\mathbf{0.002}_{\pm 0.000}$ |

*Table 10.* Comparison of **Wasserstein (WD (test)) distances**, computed using the test set. **Bold** indicates the best and underline the second best result. We report the average and standard deviation across three training runs and 10 different generated samples each.

|  | ARF | TVAE | CTGAN | TabDDPM | TabSyn | TabDiff | CDTD | Ours (DT) |
|---|---|---|---|---|---|---|---|---|
| adult | $0.007_{\pm0.001}$ | $0.021_{\pm0.004}$ | $0.020_{\pm0.002}$ | $0.003_{\pm0.000}$ | $0.006_{\pm0.002}$ | $0.003$$_{\pm0.000}$ | $0.006_{\pm0.000}$ | **$0.003$**$_{\pm0.001}$ |
| airlines | $0.003$$_{\pm0.000}$ | $0.048_{\pm0.015}$ | $0.041_{\pm0.004}$ | - | $0.014_{\pm0.005}$ | $0.021_{\pm0.004}$ | $0.003_{\pm0.000}$ | **$0.002$**$_{\pm0.000}$ |
| beijing | $0.018_{\pm0.001}$ | $0.035_{\pm0.010}$ | $0.029_{\pm0.001}$ | $0.006_{\pm0.001}$ | $0.009_{\pm0.001}$ | $0.006$$_{\pm0.001}$ | $0.009_{\pm0.000}$ | **$0.004$**$_{\pm0.000}$ |
| credit_g | $0.062_{\pm0.005}$ | $0.041_{\pm0.006}$ | $0.059_{\pm0.015}$ | $0.031$$_{\pm0.004}$ | $0.107_{\pm0.019}$ | $0.071_{\pm0.005}$ | $0.032_{\pm0.004}$ | **$0.029$**$_{\pm0.004}$ |
| default | $0.009_{\pm0.001}$ | $0.010_{\pm0.003}$ | $0.009_{\pm0.002}$ | $0.004_{\pm0.000}$ | $0.009_{\pm0.001}$ | $0.003$$_{\pm0.000}$ | $0.006_{\pm0.000}$ | **$0.003$**$_{\pm0.000}$ |
| diabetes | $0.022_{\pm0.000}$ | $0.027_{\pm0.003}$ | $0.019_{\pm0.004}$ | - | $0.026_{\pm0.009}$ | $0.007$$_{\pm0.001}$ | $0.026_{\pm0.006}$ | **$0.004$**$_{\pm0.000}$ |
| electricity | $0.008_{\pm0.000}$ | $0.029_{\pm0.007}$ | $0.028_{\pm0.003}$ | $0.004_{\pm0.000}$ | $0.007_{\pm0.001}$ | **$0.003$**$_{\pm0.000}$ | $0.005_{\pm0.000}$ | $0.004$$_{\pm0.000}$ |
| kc1 | $0.021_{\pm0.003}$ | $0.025_{\pm0.005}$ | $0.023_{\pm0.006}$ | $0.011_{\pm0.002}$ | $0.116_{\pm0.023}$ | $0.086_{\pm0.011}$ | **$0.011$**$_{\pm0.002}$ | $0.011$$_{\pm0.003}$ |
| news | $0.072_{\pm0.038}$ | $0.045_{\pm0.014}$ | **$0.014$**$_{\pm0.001}$ | - | $0.666_{\pm0.654}$ | $0.030_{\pm0.008}$ | $0.024_{\pm0.006}$ | $0.015$$_{\pm0.007}$ |
| nmes | $0.031_{\pm0.003}$ | $0.012_{\pm0.002}$ | $0.025_{\pm0.002}$ | $0.010$$_{\pm0.001}$ | $0.033_{\pm0.007}$ | $0.011_{\pm0.001}$ | $0.010_{\pm0.001}$ | **$0.009$**$_{\pm0.002}$ |
| phoneme | $0.011_{\pm0.002}$ | $0.021_{\pm0.004}$ | $0.048_{\pm0.006}$ | $0.009_{\pm0.003}$ | $0.017_{\pm0.004}$ | $0.010_{\pm0.002}$ | **$0.009$**$_{\pm0.002}$ | $0.009$$_{\pm0.003}$ |
| shoppers | $0.018_{\pm0.001}$ | $0.013_{\pm0.002}$ | $0.016_{\pm0.001}$ | $0.009_{\pm0.001}$ | $0.036_{\pm0.025}$ | $0.006$$_{\pm0.001}$ | $0.009_{\pm0.001}$ | **$0.005$**$_{\pm0.001}$ |

*Table 11.* Comparison of **Jensen-Shannon divergences (JSD)**, which we use to evaluate categorical univariate densities only. **Bold** indicates the best and underline the second best result. We report the average and standard deviation across three training runs and 10 different generated samples each.

|  | ARF | TVAE | CTGAN | TabDDPM | TabSyn | TabDiff | CDTD | Ours (DT) |
|---|---|---|---|---|---|---|---|---|
| adult | **$0.006$**$_{\pm0.001}$ | $0.149_{\pm0.008}$ | $0.128_{\pm0.005}$ | $0.031_{\pm0.002}$ | $0.042_{\pm0.011}$ | $0.007$$_{\pm0.001}$ | $0.020_{\pm0.001}$ | $0.018_{\pm0.001}$ |
| airlines | **$0.011$**$_{\pm0.000}$ | $0.341_{\pm0.013}$ | $0.251_{\pm0.017}$ | - | $0.063_{\pm0.004}$ | $0.220_{\pm0.008}$ | $0.072_{\pm0.002}$ | $0.054$$_{\pm0.031}$ |
| beijing | **$0.004$**$_{\pm0.002}$ | $0.260_{\pm0.021}$ | $0.102_{\pm0.016}$ | $0.012_{\pm0.002}$ | $0.010_{\pm0.005}$ | $0.005$$_{\pm0.002}$ | $0.007_{\pm0.002}$ | $0.006_{\pm0.002}$ |
| credit_g | $0.025_{\pm0.003}$ | $0.057_{\pm0.008}$ | $0.134_{\pm0.009}$ | $0.032_{\pm0.004}$ | $0.110_{\pm0.022}$ | $0.032_{\pm0.003}$ | **$0.024$**$_{\pm0.003}$ | $0.025$$_{\pm0.002}$ |
| default | **$0.007$**$_{\pm0.001}$ | $0.134_{\pm0.024}$ | $0.127_{\pm0.016}$ | $0.035_{\pm0.001}$ | $0.076_{\pm0.007}$ | $0.010$$_{\pm0.002}$ | $0.024_{\pm0.003}$ | $0.023_{\pm0.003}$ |
| diabetes | **$0.007$**$_{\pm0.000}$ | $0.189_{\pm0.018}$ | $0.091_{\pm0.008}$ | - | $0.110_{\pm0.003}$ | $0.040_{\pm0.000}$ | $0.031_{\pm0.002}$ | $0.024$$_{\pm0.001}$ |
| electricity | **$0.004$**$_{\pm0.001}$ | $0.110_{\pm0.038}$ | $0.125_{\pm0.025}$ | $0.005_{\pm0.002}$ | $0.007_{\pm0.002}$ | $0.005_{\pm0.001}$ | $0.005_{\pm0.001}$ | $0.005_{\pm0.001}$ |
| kc1 | **$0.008$**$_{\pm0.005}$ | $0.024_{\pm0.014}$ | $0.061_{\pm0.050}$ | $0.052_{\pm0.014}$ | $0.087_{\pm0.055}$ | $0.011_{\pm0.007}$ | $0.009$$_{\pm0.009}$ | $0.010_{\pm0.006}$ |
| news | **$0.002$**$_{\pm0.000}$ | $0.182_{\pm0.018}$ | $0.017_{\pm0.002}$ | - | $0.072_{\pm0.019}$ | $0.004$$_{\pm0.001}$ | $0.015_{\pm0.002}$ | $0.011_{\pm0.001}$ |
| nmes | **$0.008$**$_{\pm0.002}$ | $0.034_{\pm0.009}$ | $0.108_{\pm0.017}$ | $0.034_{\pm0.002}$ | $0.063_{\pm0.027}$ | $0.008_{\pm0.002}$ | $0.008$$_{\pm0.002}$ | $0.008_{\pm0.002}$ |
| phoneme | $0.007_{\pm0.006}$ | $0.008_{\pm0.005}$ | $0.094_{\pm0.009}$ | $0.006_{\pm0.004}$ | $0.006_{\pm0.004}$ | **$0.005$**$_{\pm0.003}$ | $0.006$$_{\pm0.005}$ | $0.006_{\pm0.004}$ |
| shoppers | **$0.011$**$_{\pm0.001}$ | $0.057_{\pm0.005}$ | $0.113_{\pm0.013}$ | $0.075_{\pm0.006}$ | $0.106_{\pm0.035}$ | $0.012$$_{\pm0.001}$ | $0.015_{\pm0.001}$ | $0.021_{\pm0.002}$ |

*Table 12.* Comparison of **Jensen-Shannon divergences (JSD (test))**, computed using the test set. **Bold** indicates the best and underline the second best result. We report the average and standard deviation across three training runs and 10 different generated samples each.

|  | ARF | TVAE | CTGAN | TabDDPM | TabSyn | TabDiff | CDTD | Ours (DT) |
|---|---|---|---|---|---|---|---|---|
| adult | **$0.014$**$_{\pm0.001}$ | $0.149_{\pm0.007}$ | $0.128_{\pm0.005}$ | $0.033_{\pm0.003}$ | $0.045_{\pm0.010}$ | $0.014$$_{\pm0.001}$ | $0.023_{\pm0.001}$ | $0.022_{\pm0.002}$ |
| airlines | **$0.016$**$_{\pm0.001}$ | $0.341_{\pm0.013}$ | $0.251_{\pm0.017}$ | - | $0.064_{\pm0.004}$ | $0.220_{\pm0.008}$ | $0.075_{\pm0.002}$ | $0.057$$_{\pm0.030}$ |
| beijing | $0.009_{\pm0.003}$ | $0.261_{\pm0.022}$ | $0.101_{\pm0.018}$ | $0.013_{\pm0.005}$ | **$0.009$**$_{\pm0.002}$ | $0.009$$_{\pm0.003}$ | $0.011_{\pm0.004}$ | $0.010_{\pm0.003}$ |
| credit_g | $0.055_{\pm0.005}$ | $0.077_{\pm0.004}$ | $0.145_{\pm0.014}$ | $0.062_{\pm0.009}$ | $0.121_{\pm0.017}$ | $0.061_{\pm0.005}$ | $0.055$$_{\pm0.006}$ | **$0.054$**$_{\pm0.006}$ |
| default | **$0.015$**$_{\pm0.001}$ | $0.134_{\pm0.022}$ | $0.126_{\pm0.016}$ | $0.037_{\pm0.001}$ | $0.077_{\pm0.005}$ | $0.016$$_{\pm0.003}$ | $0.028_{\pm0.003}$ | $0.026_{\pm0.002}$ |
| diabetes | **$0.014$**$_{\pm0.000}$ | $0.190_{\pm0.018}$ | $0.092_{\pm0.009}$ | - | $0.110_{\pm0.003}$ | $0.042_{\pm0.000}$ | $0.034_{\pm0.002}$ | $0.027$$_{\pm0.001}$ |
| electricity | **$0.009$**$_{\pm0.001}$ | $0.110_{\pm0.038}$ | $0.125_{\pm0.023}$ | $0.010_{\pm0.002}$ | $0.011_{\pm0.003}$ | $0.009$$_{\pm0.002}$ | $0.010_{\pm0.002}$ | $0.010_{\pm0.001}$ |
| kc1 | **$0.008$**$_{\pm0.005}$ | $0.024_{\pm0.014}$ | $0.061_{\pm0.051}$ | $0.052_{\pm0.014}$ | $0.087_{\pm0.055}$ | $0.010_{\pm0.007}$ | $0.009$$_{\pm0.009}$ | $0.010_{\pm0.006}$ |
| news | **$0.006$**$_{\pm0.001}$ | $0.183_{\pm0.018}$ | $0.019_{\pm0.002}$ | - | $0.074_{\pm0.020}$ | $0.006$$_{\pm0.001}$ | $0.017_{\pm0.001}$ | $0.013_{\pm0.001}$ |
| nmes | $0.016_{\pm0.002}$ | $0.040_{\pm0.008}$ | $0.104_{\pm0.014}$ | $0.036_{\pm0.003}$ | $0.064_{\pm0.026}$ | $0.016$$_{\pm0.002}$ | $0.016_{\pm0.002}$ | **$0.016$**$_{\pm0.002}$ |
| phoneme | $0.007_{\pm0.006}$ | $0.008_{\pm0.005}$ | $0.094_{\pm0.009}$ | $0.006_{\pm0.004}$ | $0.006_{\pm0.004}$ | **$0.005$**$_{\pm0.003}$ | $0.006$$_{\pm0.005}$ | $0.006_{\pm0.004}$ |
| shoppers | **$0.024$**$_{\pm0.001}$ | $0.061_{\pm0.007}$ | $0.117_{\pm0.013}$ | $0.077_{\pm0.006}$ | $0.109_{\pm0.033}$ | $0.024$$_{\pm0.002}$ | $0.025_{\pm0.002}$ | $0.028_{\pm0.003}$ |

*Table 13.* Comparison of **Trend scores**. **Bold** indicates the best and underline the second best result. We report the average and standard deviation across three training runs and 10 different generated samples each.

|  | ARF | TVAE | CTGAN | TabDDPM | TabSyn | TabDiff | CDTD | Ours (DT) |
|---|---|---|---|---|---|---|---|---|
| adult | $0.969_{\pm0.001}$ | $0.782_{\pm0.012}$ | $0.765_{\pm0.017}$ | $0.971_{\pm0.002}$ | $0.943_{\pm0.006}$ | $\mathbf{0.982}_{\pm0.001}$ | $0.971_{\pm0.002}$ | $\underline{0.976}_{\pm0.003}$ |
| airlines | $\mathbf{0.938}_{\pm0.001}$ | $0.532_{\pm0.013}$ | $0.549_{\pm0.015}$ | - | $0.875_{\pm0.005}$ | $0.670_{\pm0.005}$ | $0.878_{\pm0.004}$ | $\underline{0.903}_{\pm0.033}$ |
| beijing | $0.976_{\pm0.001}$ | $0.925_{\pm0.016}$ | $0.942_{\pm0.006}$ | $0.989_{\pm0.001}$ | $0.983_{\pm0.003}$ | $\underline{0.990}_{\pm0.002}$ | $0.986_{\pm0.002}$ | $\mathbf{0.991}_{\pm0.001}$ |
| credit_g | $0.903_{\pm0.003}$ | $0.889_{\pm0.015}$ | $0.797_{\pm0.012}$ | $0.938_{\pm0.010}$ | $0.811_{\pm0.018}$ | $0.897_{\pm0.006}$ | $\underline{0.941}_{\pm0.007}$ | $\mathbf{0.942}_{\pm0.009}$ |
| default | $0.952_{\pm0.003}$ | $0.835_{\pm0.008}$ | $0.816_{\pm0.006}$ | $0.953_{\pm0.009}$ | $0.903_{\pm0.010}$ | $\mathbf{0.968}_{\pm0.008}$ | $0.936_{\pm0.019}$ | $\underline{0.964}_{\pm0.006}$ |
| diabetes | $\mathbf{0.962}_{\pm0.000}$ | $0.761_{\pm0.024}$ | $0.818_{\pm0.016}$ | - | $0.848_{\pm0.019}$ | $\underline{0.940}_{\pm0.002}$ | $0.917_{\pm0.008}$ | $0.936_{\pm0.003}$ |
| electricity | $0.970_{\pm0.004}$ | $0.902_{\pm0.033}$ | $0.807_{\pm0.009}$ | $\underline{0.986}_{\pm0.002}$ | $0.977_{\pm0.002}$ | $\mathbf{0.987}_{\pm0.002}$ | $0.983_{\pm0.002}$ | $0.985_{\pm0.002}$ |
| kc1 | $0.838_{\pm0.007}$ | $0.894_{\pm0.007}$ | $0.903_{\pm0.006}$ | $0.940_{\pm0.023}$ | $0.736_{\pm0.008}$ | $0.765_{\pm0.008}$ | $\underline{0.979}_{\pm0.002}$ | $\mathbf{0.984}_{\pm0.002}$ |
| news | $0.954_{\pm0.000}$ | $0.883_{\pm0.013}$ | $0.889_{\pm0.011}$ | - | $0.908_{\pm0.005}$ | $\underline{0.961}_{\pm0.003}$ | $0.961_{\pm0.004}$ | $\mathbf{0.976}_{\pm0.001}$ |
| nmes | $0.952_{\pm0.001}$ | $0.937_{\pm0.014}$ | $0.823_{\pm0.025}$ | $0.948_{\pm0.009}$ | $0.900_{\pm0.024}$ | $0.968_{\pm0.002}$ | $\underline{0.970}_{\pm0.004}$ | $\mathbf{0.979}_{\pm0.003}$ |
| phoneme | $\underline{0.976}_{\pm0.002}$ | $0.946_{\pm0.007}$ | $0.850_{\pm0.012}$ | $0.969_{\pm0.015}$ | $0.952_{\pm0.011}$ | $\mathbf{0.982}_{\pm0.003}$ | $0.974_{\pm0.012}$ | $0.974_{\pm0.012}$ |
| shoppers | $0.956_{\pm0.001}$ | $0.935_{\pm0.007}$ | $0.856_{\pm0.010}$ | $0.932_{\pm0.005}$ | $0.879_{\pm0.019}$ | $\underline{0.973}_{\pm0.001}$ | $0.973_{\pm0.002}$ | $\mathbf{0.977}_{\pm0.002}$ |

*Table 14.* Comparison of **Trend (test) scores** computed using the test set. **Bold** indicates the best and underline the second best result. We report the average and standard deviation across three training runs and 10 different generated samples each.

|  | ARF | TVAE | CTGAN | TabDDPM | TabSyn | TabDiff | CDTD | Ours (DT) |
|---|---|---|---|---|---|---|---|---|
| adult | $0.959_{\pm0.004}$ | $0.780_{\pm0.010}$ | $0.768_{\pm0.014}$ | $0.961_{\pm0.003}$ | $0.938_{\pm0.008}$ | $\mathbf{0.971}_{\pm0.003}$ | $0.962_{\pm0.003}$ | $\underline{0.968}_{\pm0.002}$ |
| airlines | $\mathbf{0.927}_{\pm0.011}$ | $0.527_{\pm0.013}$ | $0.550_{\pm0.017}$ | - | $0.870_{\pm0.008}$ | $0.666_{\pm0.002}$ | $0.872_{\pm0.010}$ | $\underline{0.896}_{\pm0.038}$ |
| beijing | $0.972_{\pm0.002}$ | $0.923_{\pm0.016}$ | $0.941_{\pm0.004}$ | $0.984_{\pm0.002}$ | $0.979_{\pm0.003}$ | $\underline{0.985}_{\pm0.002}$ | $0.982_{\pm0.003}$ | $\mathbf{0.986}_{\pm0.002}$ |
| credit_g | $0.851_{\pm0.006}$ | $0.847_{\pm0.008}$ | $0.780_{\pm0.020}$ | $0.875_{\pm0.008}$ | $0.781_{\pm0.018}$ | $0.850_{\pm0.011}$ | $\mathbf{0.882}_{\pm0.008}$ | $\underline{0.881}_{\pm0.009}$ |
| default | $0.870_{\pm0.016}$ | $0.816_{\pm0.032}$ | $0.823_{\pm0.012}$ | $0.887_{\pm0.021}$ | $0.843_{\pm0.015}$ | $\underline{0.888}_{\pm0.018}$ | $0.887_{\pm0.027}$ | $\mathbf{0.896}_{\pm0.020}$ |
| diabetes | $\mathbf{0.953}_{\pm0.001}$ | $0.758_{\pm0.025}$ | $0.820_{\pm0.018}$ | - | $0.845_{\pm0.017}$ | $\underline{0.935}_{\pm0.001}$ | $0.910_{\pm0.008}$ | $0.930_{\pm0.001}$ |
| electricity | $0.958_{\pm0.007}$ | $0.905_{\pm0.025}$ | $0.804_{\pm0.011}$ | $0.973_{\pm0.007}$ | $0.965_{\pm0.009}$ | $\underline{0.973}_{\pm0.008}$ | $0.971_{\pm0.008}$ | $\mathbf{0.973}_{\pm0.010}$ |
| kc1 | $0.829_{\pm0.013}$ | $0.886_{\pm0.011}$ | $0.898_{\pm0.005}$ | $0.929_{\pm0.024}$ | $0.729_{\pm0.003}$ | $0.758_{\pm0.002}$ | $\underline{0.963}_{\pm0.006}$ | $\mathbf{0.967}_{\pm0.005}$ |
| news | $0.918_{\pm0.003}$ | $0.860_{\pm0.016}$ | $0.895_{\pm0.004}$ | - | $0.883_{\pm0.008}$ | $0.931_{\pm0.002}$ | $\underline{0.931}_{\pm0.004}$ | $\mathbf{0.943}_{\pm0.003}$ |
| nmes | $0.917_{\pm0.008}$ | $0.914_{\pm0.010}$ | $0.826_{\pm0.025}$ | $0.918_{\pm0.008}$ | $0.881_{\pm0.013}$ | $0.930_{\pm0.009}$ | $\underline{0.934}_{\pm0.008}$ | $\mathbf{0.940}_{\pm0.007}$ |
| phoneme | $0.939_{\pm0.013}$ | $0.928_{\pm0.011}$ | $0.851_{\pm0.013}$ | $\underline{0.939}_{\pm0.011}$ | $0.928_{\pm0.011}$ | $\mathbf{0.942}_{\pm0.011}$ | $0.936_{\pm0.013}$ | $0.935_{\pm0.010}$ |
| shoppers | $0.941_{\pm0.002}$ | $0.926_{\pm0.005}$ | $0.855_{\pm0.008}$ | $0.923_{\pm0.004}$ | $0.874_{\pm0.017}$ | $\underline{0.955}_{\pm0.002}$ | $0.955_{\pm0.002}$ | $\mathbf{0.958}_{\pm0.002}$ |

*Table 15.* Comparison of **Trend (mixed) scores**, which evaluate only the dependencies across feature types. **Bold** indicates the best and underline the second best result. We report the average and standard deviation across three training runs and 10 different generated samples each.

|  | ARF | TVAE | CTGAN | TabDDPM | TabSyn | TabDiff | CDTD | Ours (DT) |
|---|---|---|---|---|---|---|---|---|
| adult | $0.959_{\pm0.001}$ | $0.705_{\pm0.021}$ | $0.685_{\pm0.024}$ | $0.966_{\pm0.002}$ | $0.932_{\pm0.005}$ | $\mathbf{0.977}_{\pm0.001}$ | $0.960_{\pm0.004}$ | $\underline{0.968}_{\pm0.006}$ |
| airlines | $\mathbf{0.938}_{\pm0.001}$ | $0.527_{\pm0.028}$ | $0.545_{\pm0.023}$ | - | $0.877_{\pm0.006}$ | $0.683_{\pm0.006}$ | $0.881_{\pm0.006}$ | $\underline{0.908}_{\pm0.026}$ |
| beijing | $0.918_{\pm0.005}$ | $0.723_{\pm0.055}$ | $0.792_{\pm0.016}$ | $0.957_{\pm0.006}$ | $0.937_{\pm0.010}$ | $\underline{0.961}_{\pm0.008}$ | $0.944_{\pm0.008}$ | $\mathbf{0.969}_{\pm0.006}$ |
| credit_g | $0.851_{\pm0.006}$ | $0.848_{\pm0.026}$ | $0.760_{\pm0.018}$ | $\underline{0.920}_{\pm0.020}$ | $0.750_{\pm0.013}$ | $0.833_{\pm0.014}$ | $0.917_{\pm0.013}$ | $\mathbf{0.921}_{\pm0.018}$ |
| default | $\mathbf{0.964}_{\pm0.006}$ | $0.759_{\pm0.019}$ | $0.715_{\pm0.019}$ | $0.928_{\pm0.018}$ | $0.867_{\pm0.019}$ | $\underline{0.957}_{\pm0.015}$ | $0.892_{\pm0.038}$ | $0.948_{\pm0.013}$ |
| diabetes | $\underline{0.928}_{\pm0.001}$ | $0.716_{\pm0.032}$ | $0.708_{\pm0.038}$ | - | $0.830_{\pm0.047}$ | $\mathbf{0.937}_{\pm0.004}$ | $0.829_{\pm0.019}$ | $0.872_{\pm0.003}$ |
| electricity | $0.959_{\pm0.002}$ | $0.812_{\pm0.074}$ | $0.624_{\pm0.017}$ | $\underline{0.978}_{\pm0.002}$ | $0.963_{\pm0.004}$ | $\mathbf{0.981}_{\pm0.001}$ | $0.971_{\pm0.006}$ | $0.976_{\pm0.005}$ |
| kc1 | $0.956_{\pm0.003}$ | $0.890_{\pm0.014}$ | $0.877_{\pm0.027}$ | $0.926_{\pm0.009}$ | $0.820_{\pm0.038}$ | $0.902_{\pm0.006}$ | $\mathbf{0.963}_{\pm0.006}$ | $\underline{0.960}_{\pm0.010}$ |
| news | $0.926_{\pm0.001}$ | $0.754_{\pm0.023}$ | $0.738_{\pm0.029}$ | - | $0.821_{\pm0.013}$ | $\underline{0.927}_{\pm0.007}$ | $0.922_{\pm0.009}$ | $\mathbf{0.952}_{\pm0.003}$ |
| nmes | $0.931_{\pm0.003}$ | $0.907_{\pm0.020}$ | $0.751_{\pm0.041}$ | $0.927_{\pm0.016}$ | $0.862_{\pm0.031}$ | $0.954_{\pm0.003}$ | $\underline{0.955}_{\pm0.008}$ | $\mathbf{0.970}_{\pm0.006}$ |
| phoneme | $\underline{0.948}_{\pm0.005}$ | $0.880_{\pm0.021}$ | $0.679_{\pm0.034}$ | $0.918_{\pm0.044}$ | $0.891_{\pm0.027}$ | $\mathbf{0.957}_{\pm0.008}$ | $0.935_{\pm0.036}$ | $0.934_{\pm0.036}$ |
| shoppers | $0.959_{\pm0.002}$ | $0.923_{\pm0.011}$ | $0.805_{\pm0.018}$ | $0.912_{\pm0.006}$ | $0.852_{\pm0.026}$ | $\underline{0.967}_{\pm0.001}$ | $0.964_{\pm0.002}$ | $\mathbf{0.971}_{\pm0.003}$ |

*Table 16.* Comparison of **MLE**. Per dataset, **bold** indicates the best and underline the second best result. We report the average and standard deviation across three training runs and 10 different generated samples each.

| | ARF | TVAE | CTGAN | TabDDPM | TabSyn | TabDiff | CDTD | Ours (DT) |
|---|---|---|---|---|---|---|---|---|
| adult | $0.019_{\pm0.003}$ | $0.077_{\pm0.018}$ | $0.094_{\pm0.016}$ | $0.018_{\pm0.005}$ | $0.029_{\pm0.003}$ | $\underline{0.015}_{\pm0.002}$ | $0.016_{\pm0.003}$ | $\mathbf{0.007}_{\pm0.001}$ |
| airlines | $\mathbf{0.013}_{\pm0.004}$ | $0.098_{\pm0.017}$ | $0.135_{\pm0.014}$ | - | $\underline{0.032}_{\pm0.004}$ | $0.083_{\pm0.007}$ | $0.068_{\pm0.008}$ | $0.080_{\pm0.047}$ |
| beijing | $0.101_{\pm0.006}$ | $0.286_{\pm0.060}$ | $0.253_{\pm0.019}$ | $0.044_{\pm0.003}$ | $0.093_{\pm0.014}$ | $0.053_{\pm0.004}$ | $\underline{0.034}_{\pm0.005}$ | $\mathbf{0.032}_{\pm0.004}$ |
| credit_g | $0.125_{\pm0.056}$ | $0.063_{\pm0.037}$ | $0.137_{\pm0.049}$ | $0.023_{\pm0.017}$ | $0.098_{\pm0.065}$ | $0.062_{\pm0.021}$ | $\underline{0.023}_{\pm0.017}$ | $\mathbf{0.017}_{\pm0.013}$ |
| default | $0.014_{\pm0.003}$ | $0.017_{\pm0.007}$ | $0.039_{\pm0.007}$ | $\mathbf{0.007}_{\pm0.005}$ | $0.034_{\pm0.019}$ | $\underline{0.009}_{\pm0.004}$ | $0.009_{\pm0.005}$ | $0.009_{\pm0.004}$ |
| diabetes | $\underline{0.031}_{\pm0.014}$ | $0.063_{\pm0.015}$ | $0.081_{\pm0.030}$ | - | $0.093_{\pm0.017}$ | $\mathbf{0.023}_{\pm0.018}$ | $0.053_{\pm0.016}$ | $0.036_{\pm0.015}$ |
| electricity | $0.054_{\pm0.004}$ | $0.126_{\pm0.024}$ | $0.196_{\pm0.016}$ | $0.037_{\pm0.002}$ | $0.065_{\pm0.002}$ | $0.038_{\pm0.001}$ | $\underline{0.033}_{\pm0.002}$ | $\mathbf{0.021}_{\pm0.001}$ |
| kc1 | $0.048_{\pm0.028}$ | $0.040_{\pm0.022}$ | $0.042_{\pm0.021}$ | $0.031_{\pm0.019}$ | $0.101_{\pm0.075}$ | $0.075_{\pm0.041}$ | $\underline{0.024}_{\pm0.015}$ | $\mathbf{0.016}_{\pm0.011}$ |
| news | $0.115_{\pm0.044}$ | $0.059_{\pm0.066}$ | $\mathbf{0.013}_{\pm0.004}$ | - | $3.302_{\pm2.642}$ | $0.083_{\pm0.026}$ | $0.153_{\pm0.035}$ | $\underline{0.054}_{\pm0.045}$ |
| nmes | $0.160_{\pm0.043}$ | $0.037_{\pm0.023}$ | $0.124_{\pm0.070}$ | $\mathbf{0.023}_{\pm0.018}$ | $0.121_{\pm0.096}$ | $0.052_{\pm0.034}$ | $0.034_{\pm0.018}$ | $\underline{0.027}_{\pm0.024}$ |
| phoneme | $0.047_{\pm0.006}$ | $0.061_{\pm0.011}$ | $0.170_{\pm0.038}$ | $\underline{0.011}_{\pm0.005}$ | $0.088_{\pm0.020}$ | $0.024_{\pm0.006}$ | $0.011_{\pm0.004}$ | $\mathbf{0.010}_{\pm0.005}$ |
| shoppers | $0.052_{\pm0.012}$ | $0.026_{\pm0.010}$ | $0.117_{\pm0.015}$ | $0.014_{\pm0.006}$ | $0.045_{\pm0.012}$ | $0.021_{\pm0.006}$ | $\underline{0.011}_{\pm0.005}$ | $\mathbf{0.009}_{\pm0.005}$ |

*Table 17.* Comparison of $\alpha$-**Precision scores**. Per dataset, **bold** indicates the best and underline the second best result. We report the average and standard deviation across three training runs and 10 different generated samples each.

| | ARF | TVAE | CTGAN | TabDDPM | TabSyn | TabDiff | CDTD | Ours (DT) |
|---|---|---|---|---|---|---|---|---|
| adult | $0.991_{\pm0.003}$ | $0.766_{\pm0.021}$ | $0.804_{\pm0.077}$ | $0.928_{\pm0.012}$ | $0.970_{\pm0.023}$ | $\mathbf{0.995}_{\pm0.001}$ | $\underline{0.993}_{\pm0.002}$ | $0.981_{\pm0.003}$ |
| airlines | $\mathbf{0.996}_{\pm0.001}$ | $0.623_{\pm0.043}$ | $0.903_{\pm0.038}$ | - | $\underline{0.991}_{\pm0.002}$ | $0.733_{\pm0.025}$ | $0.963_{\pm0.003}$ | $0.974_{\pm0.011}$ |
| beijing | $0.933_{\pm0.003}$ | $0.698_{\pm0.165}$ | $0.806_{\pm0.009}$ | $0.964_{\pm0.005}$ | $\underline{0.982}_{\pm0.013}$ | $0.973_{\pm0.004}$ | $\mathbf{0.992}_{\pm0.003}$ | $0.980_{\pm0.007}$ |
| credit_g | $0.948_{\pm0.013}$ | $0.903_{\pm0.030}$ | $0.812_{\pm0.118}$ | $0.910_{\pm0.020}$ | $0.812_{\pm0.145}$ | $0.923_{\pm0.023}$ | $\mathbf{0.978}_{\pm0.008}$ | $\underline{0.968}_{\pm0.014}$ |
| default | $0.957_{\pm0.004}$ | $0.772_{\pm0.078}$ | $0.825_{\pm0.006}$ | $0.907_{\pm0.007}$ | $0.941_{\pm0.040}$ | $0.975_{\pm0.007}$ | $\underline{0.978}_{\pm0.005}$ | $\mathbf{0.986}_{\pm0.003}$ |
| diabetes | $0.976_{\pm0.002}$ | $0.261_{\pm0.070}$ | $0.878_{\pm0.055}$ | - | $0.926_{\pm0.047}$ | $0.826_{\pm0.010}$ | $\underline{0.979}_{\pm0.013}$ | $\mathbf{0.993}_{\pm0.004}$ |
| electricity | $\mathbf{0.995}_{\pm0.001}$ | $0.910_{\pm0.078}$ | $0.833_{\pm0.025}$ | $0.978_{\pm0.004}$ | $\underline{0.992}_{\pm0.002}$ | $0.979_{\pm0.004}$ | $0.988_{\pm0.002}$ | $0.990_{\pm0.004}$ |
| kc1 | $0.931_{\pm0.013}$ | $0.913_{\pm0.048}$ | $0.879_{\pm0.074}$ | $0.802_{\pm0.018}$ | $0.496_{\pm0.032}$ | $0.721_{\pm0.031}$ | $\mathbf{0.979}_{\pm0.008}$ | $\underline{0.960}_{\pm0.014}$ |
| news | $0.898_{\pm0.005}$ | $0.139_{\pm0.061}$ | $\underline{0.930}_{\pm0.008}$ | - | $0.611_{\pm0.172}$ | $\mathbf{0.972}_{\pm0.008}$ | $0.851_{\pm0.012}$ | $0.907_{\pm0.008}$ |
| nmes | $0.950_{\pm0.008}$ | $0.959_{\pm0.021}$ | $0.832_{\pm0.064}$ | $0.846_{\pm0.008}$ | $0.916_{\pm0.016}$ | $0.983_{\pm0.005}$ | $\underline{0.986}_{\pm0.008}$ | $\mathbf{0.987}_{\pm0.006}$ |
| phoneme | $\underline{0.988}_{\pm0.004}$ | $0.954_{\pm0.026}$ | $0.871_{\pm0.017}$ | $0.978_{\pm0.007}$ | $0.925_{\pm0.029}$ | $0.963_{\pm0.008}$ | $0.987_{\pm0.005}$ | $\mathbf{0.989}_{\pm0.006}$ |
| shoppers | $0.964_{\pm0.006}$ | $0.938_{\pm0.025}$ | $0.922_{\pm0.075}$ | $0.767_{\pm0.017}$ | $0.851_{\pm0.070}$ | $\underline{0.983}_{\pm0.006}$ | $0.982_{\pm0.006}$ | $\mathbf{0.987}_{\pm0.007}$ |

*Table 18.* Comparison of $\beta$-**Recall scores**. Per dataset, **bold** indicates the best and underline the second best result. We report the average and standard deviation across three training runs and 10 different generated samples each.

| | ARF | TVAE | CTGAN | TabDDPM | TabSyn | TabDiff | CDTD | Ours (DT) |
|---|---|---|---|---|---|---|---|---|
| adult | $0.420_{\pm0.004}$ | $0.196_{\pm0.018}$ | $0.162_{\pm0.039}$ | $0.525_{\pm0.008}$ | $0.397_{\pm0.014}$ | $0.477_{\pm0.003}$ | $\underline{0.573}_{\pm0.004}$ | $\mathbf{0.596}_{\pm0.009}$ |
| airlines | $\mathbf{0.430}_{\pm0.003}$ | $0.010_{\pm0.001}$ | $0.010_{\pm0.004}$ | - | $0.231_{\pm0.018}$ | $0.018_{\pm0.003}$ | $0.335_{\pm0.005}$ | $\underline{0.375}_{\pm0.048}$ |
| beijing | $0.276_{\pm0.005}$ | $0.097_{\pm0.057}$ | $0.143_{\pm0.017}$ | $0.385_{\pm0.012}$ | $0.305_{\pm0.023}$ | $0.365_{\pm0.012}$ | $\underline{0.424}_{\pm0.011}$ | $\mathbf{0.523}_{\pm0.007}$ |
| credit_g | $0.430_{\pm0.024}$ | $0.700_{\pm0.021}$ | $0.327_{\pm0.019}$ | $\mathbf{0.749}_{\pm0.019}$ | $0.334_{\pm0.036}$ | $0.691_{\pm0.025}$ | $\underline{0.737}_{\pm0.019}$ | $0.715_{\pm0.029}$ |
| default | $0.362_{\pm0.007}$ | $0.247_{\pm0.032}$ | $0.304_{\pm0.032}$ | $0.553_{\pm0.004}$ | $0.346_{\pm0.026}$ | $0.482_{\pm0.005}$ | $\mathbf{0.603}_{\pm0.008}$ | $\underline{0.562}_{\pm0.006}$ |
| diabetes | $0.329_{\pm0.006}$ | $0.179_{\pm0.079}$ | $0.178_{\pm0.063}$ | - | $0.176_{\pm0.024}$ | $0.274_{\pm0.010}$ | $\mathbf{0.561}_{\pm0.017}$ | $\underline{0.517}_{\pm0.004}$ |
| electricity | $0.368_{\pm0.004}$ | $0.162_{\pm0.061}$ | $0.121_{\pm0.020}$ | $0.458_{\pm0.003}$ | $0.340_{\pm0.010}$ | $0.438_{\pm0.002}$ | $\underline{0.490}_{\pm0.003}$ | $\mathbf{0.581}_{\pm0.010}$ |
| kc1 | $0.299_{\pm0.011}$ | $0.354_{\pm0.034}$ | $0.223_{\pm0.037}$ | $0.587_{\pm0.015}$ | $0.050_{\pm0.011}$ | $0.079_{\pm0.009}$ | $\underline{0.602}_{\pm0.019}$ | $\mathbf{0.611}_{\pm0.017}$ |
| news | $0.114_{\pm0.003}$ | $0.049_{\pm0.025}$ | $0.339_{\pm0.031}$ | - | $0.035_{\pm0.016}$ | $0.366_{\pm0.018}$ | $\mathbf{0.517}_{\pm0.010}$ | $\underline{0.478}_{\pm0.010}$ |
| nmes | $0.434_{\pm0.008}$ | $0.528_{\pm0.038}$ | $0.344_{\pm0.032}$ | $\mathbf{0.776}_{\pm0.009}$ | $0.360_{\pm0.046}$ | $0.501_{\pm0.012}$ | $\underline{0.761}_{\pm0.008}$ | $0.704_{\pm0.012}$ |
| phoneme | $0.295_{\pm0.010}$ | $0.235_{\pm0.016}$ | $0.094_{\pm0.007}$ | $\underline{0.641}_{\pm0.009}$ | $0.189_{\pm0.025}$ | $0.440_{\pm0.007}$ | $0.633_{\pm0.009}$ | $\mathbf{0.740}_{\pm0.008}$ |
| shoppers | $0.423_{\pm0.006}$ | $0.483_{\pm0.011}$ | $0.326_{\pm0.017}$ | $0.664_{\pm0.024}$ | $0.301_{\pm0.056}$ | $0.477_{\pm0.007}$ | $\mathbf{0.729}_{\pm0.007}$ | $\underline{0.691}_{\pm0.007}$ |

*Table 19.* Comparison of **DCR share scores**. Per dataset, **bold** indicates the best and underline the second best result. We report the average and standard deviation across three training runs and 10 different generated samples each.

| | ARF | TVAE | CTGAN | TabDDPM | TabSyn | TabDiff | CDTD | Ours (DT) |
|---|---|---|---|---|---|---|---|---|
| adult | $0.815_{\pm 0.002}$ | $0.800_{\pm 0.004}$ | $\underline{0.781}_{\pm 0.003}$ | $0.799_{\pm 0.005}$ | $\mathbf{0.780}_{\pm 0.003}$ | $0.786_{\pm 0.003}$ | $0.863_{\pm 0.002}$ | $0.871_{\pm 0.006}$ |
| airlines | $0.833_{\pm 0.002}$ | $0.788_{\pm 0.008}$ | $0.792_{\pm 0.010}$ | - | $\underline{0.776}_{\pm 0.002}$ | $\mathbf{0.771}_{\pm 0.005}$ | $0.800_{\pm 0.002}$ | $0.785_{\pm 0.002}$ |
| beijing | $0.796_{\pm 0.003}$ | $0.810_{\pm 0.015}$ | $\underline{0.784}_{\pm 0.007}$ | $0.792_{\pm 0.002}$ | $\mathbf{0.780}_{\pm 0.003}$ | $0.786_{\pm 0.003}$ | $0.813_{\pm 0.003}$ | $0.835_{\pm 0.002}$ |
| credit_g | $\underline{0.830}_{\pm 0.013}$ | $0.965_{\pm 0.008}$ | $\mathbf{0.801}_{\pm 0.018}$ | $1.000_{\pm 0.001}$ | $0.836_{\pm 0.023}$ | $0.921_{\pm 0.017}$ | $0.999_{\pm 0.001}$ | $1.000_{\pm 0.001}$ |
| default | $0.793_{\pm 0.004}$ | $0.792_{\pm 0.016}$ | $\underline{0.783}_{\pm 0.003}$ | $0.800_{\pm 0.003}$ | $\mathbf{0.780}_{\pm 0.004}$ | $0.786_{\pm 0.003}$ | $0.851_{\pm 0.005}$ | $0.839_{\pm 0.004}$ |
| diabetes | $0.806_{\pm 0.002}$ | $0.787_{\pm 0.009}$ | $0.778_{\pm 0.004}$ | - | $\mathbf{0.775}_{\pm 0.002}$ | $\underline{0.777}_{\pm 0.002}$ | $0.837_{\pm 0.002}$ | $0.799_{\pm 0.002}$ |
| electricity | $0.804_{\pm 0.004}$ | $0.803_{\pm 0.006}$ | $\underline{0.789}_{\pm 0.007}$ | $0.800_{\pm 0.003}$ | $\mathbf{0.781}_{\pm 0.003}$ | $0.791_{\pm 0.003}$ | $0.827_{\pm 0.003}$ | $0.863_{\pm 0.005}$ |
| kc1 | $0.817_{\pm 0.013}$ | $0.890_{\pm 0.011}$ | $\underline{0.800}_{\pm 0.017}$ | $0.897_{\pm 0.011}$ | $0.807_{\pm 0.023}$ | $\mathbf{0.799}_{\pm 0.021}$ | $0.939_{\pm 0.013}$ | $0.971_{\pm 0.011}$ |
| news | $0.785_{\pm 0.003}$ | $0.815_{\pm 0.015}$ | $0.783_{\pm 0.002}$ | - | $\mathbf{0.780}_{\pm 0.005}$ | $\underline{0.782}_{\pm 0.002}$ | $0.818_{\pm 0.004}$ | $0.805_{\pm 0.003}$ |
| nmes | $0.800_{\pm 0.007}$ | $0.851_{\pm 0.013}$ | $\mathbf{0.781}_{\pm 0.016}$ | $0.972_{\pm 0.004}$ | $\underline{0.782}_{\pm 0.008}$ | $0.791_{\pm 0.006}$ | $0.972_{\pm 0.004}$ | $0.982_{\pm 0.002}$ |
| phoneme | $0.808_{\pm 0.008}$ | $0.803_{\pm 0.008}$ | $\mathbf{0.734}_{\pm 0.048}$ | $0.923_{\pm 0.007}$ | $\underline{0.782}_{\pm 0.008}$ | $0.827_{\pm 0.007}$ | $0.921_{\pm 0.006}$ | $0.990_{\pm 0.002}$ |
| shoppers | $0.800_{\pm 0.004}$ | $0.816_{\pm 0.006}$ | $0.784_{\pm 0.006}$ | $0.856_{\pm 0.020}$ | $\mathbf{0.780}_{\pm 0.005}$ | $\underline{0.782}_{\pm 0.005}$ | $0.955_{\pm 0.004}$ | $0.937_{\pm 0.004}$ |

*Table 20.* Comparison of **MIA scores**. Per dataset, **bold** indicates the best and underline the second best result. We report the average and standard deviation across three training runs and 10 different generated samples each.

| | ARF | TVAE | CTGAN | TabDDPM | TabSyn | TabDiff | CDTD | Ours (DT) |
|---|---|---|---|---|---|---|---|---|
| adult | $0.977_{\pm 0.009}$ | $\underline{0.987}_{\pm 0.006}$ | $\mathbf{0.994}_{\pm 0.004}$ | $0.971_{\pm 0.010}$ | $0.985_{\pm 0.007}$ | $0.981_{\pm 0.006}$ | $0.969_{\pm 0.010}$ | $0.960_{\pm 0.008}$ |
| airlines | $0.993_{\pm 0.002}$ | $0.998_{\pm 0.001}$ | $\underline{0.998}_{\pm 0.001}$ | - | $0.997_{\pm 0.002}$ | $\mathbf{0.999}_{\pm 0.001}$ | $0.996_{\pm 0.002}$ | $0.995_{\pm 0.003}$ |
| beijing | $\mathbf{0.988}_{\pm 0.006}$ | $\underline{0.988}_{\pm 0.005}$ | $0.983_{\pm 0.007}$ | $0.976_{\pm 0.006}$ | $0.988_{\pm 0.005}$ | $0.980_{\pm 0.006}$ | $0.979_{\pm 0.007}$ | $0.956_{\pm 0.007}$ |
| credit_g | $0.946_{\pm 0.030}$ | $0.910_{\pm 0.038}$ | $\underline{0.955}_{\pm 0.034}$ | $0.875_{\pm 0.053}$ | $\mathbf{0.957}_{\pm 0.026}$ | $0.947_{\pm 0.030}$ | $0.889_{\pm 0.047}$ | $0.858_{\pm 0.049}$ |
| default | $0.978_{\pm 0.010}$ | $\underline{0.980}_{\pm 0.010}$ | $\mathbf{0.988}_{\pm 0.006}$ | $0.975_{\pm 0.010}$ | $0.978_{\pm 0.008}$ | $0.975_{\pm 0.009}$ | $0.979_{\pm 0.008}$ | $0.949_{\pm 0.010}$ |
| diabetes | $\underline{0.995}_{\pm 0.004}$ | $0.994_{\pm 0.004}$ | $0.992_{\pm 0.003}$ | - | $0.994_{\pm 0.003}$ | $0.995_{\pm 0.003}$ | $\mathbf{0.996}_{\pm 0.002}$ | $0.981_{\pm 0.006}$ |
| electricity | $0.985_{\pm 0.007}$ | $0.986_{\pm 0.011}$ | $\mathbf{0.993}_{\pm 0.004}$ | $0.981_{\pm 0.007}$ | $\underline{0.990}_{\pm 0.005}$ | $0.981_{\pm 0.008}$ | $0.982_{\pm 0.006}$ | $0.957_{\pm 0.010}$ |
| kc1 | $0.946_{\pm 0.028}$ | $0.943_{\pm 0.038}$ | $0.945_{\pm 0.041}$ | $0.922_{\pm 0.037}$ | $\underline{0.972}_{\pm 0.016}$ | $\mathbf{0.974}_{\pm 0.014}$ | $0.920_{\pm 0.042}$ | $0.904_{\pm 0.043}$ |
| news | $\underline{0.993}_{\pm 0.004}$ | $0.986_{\pm 0.007}$ | $0.988_{\pm 0.005}$ | - | $0.993_{\pm 0.005}$ | $\mathbf{0.993}_{\pm 0.005}$ | $0.989_{\pm 0.006}$ | $0.962_{\pm 0.007}$ |
| nmes | $0.965_{\pm 0.021}$ | $0.944_{\pm 0.024}$ | $\mathbf{0.983}_{\pm 0.012}$ | $0.918_{\pm 0.023}$ | $\underline{0.976}_{\pm 0.019}$ | $0.956_{\pm 0.021}$ | $0.924_{\pm 0.021}$ | $0.899_{\pm 0.028}$ |
| phoneme | $0.941_{\pm 0.027}$ | $0.946_{\pm 0.029}$ | $\mathbf{0.975}_{\pm 0.019}$ | $0.914_{\pm 0.023}$ | $\underline{0.958}_{\pm 0.025}$ | $0.937_{\pm 0.027}$ | $0.916_{\pm 0.019}$ | $0.892_{\pm 0.030}$ |
| shoppers | $0.975_{\pm 0.011}$ | $0.975_{\pm 0.011}$ | $\underline{0.983}_{\pm 0.013}$ | $0.963_{\pm 0.014}$ | $\mathbf{0.984}_{\pm 0.011}$ | $0.973_{\pm 0.011}$ | $0.955_{\pm 0.014}$ | $0.907_{\pm 0.014}$ |

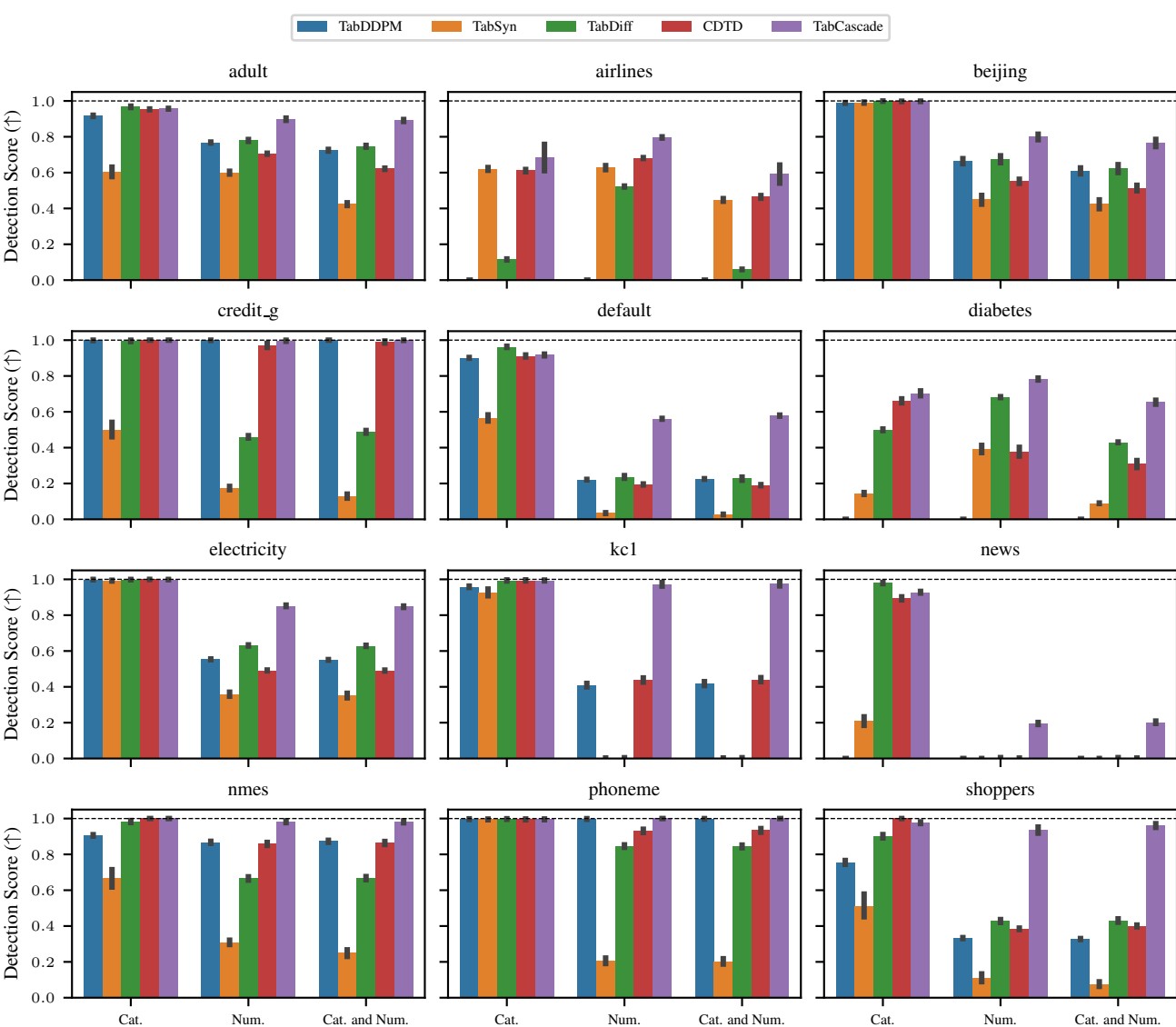

*Figure 23.* Detection scores for all diffusion-based models and all datasets over three training and ten sampling seeds. The Cat. score considers only categorical features, the Num. score only numerical features.

## A.9. Additional Ablation Experiments

### A.9.1. ABLATION EXPERIMENT TRAINING ON DATA WITHOUT MISSING VALUES

*Table 21.* Average results across datasets (**without missing values**) and ten sampling seeds for a single training run. We report the standard deviation of the 12 datasets. The best, row-wise result is indicated in **bold**, the second best is underlined. Unlike for the main results, TabDDPM is able to produce samples for the `news` dataset.

| Metric | ARF | TVAE | CTGAN | TabDDPM | TabSyn | TabDiff | CDTD | Ours (DT) |
|---|---|---|---|---|---|---|---|---|
| Detection Score | $0.373_{\pm0.200}$ | $0.216_{\pm0.248}$ | $0.091_{\pm0.106}$ | $0.550_{\pm0.359}$ | $0.340_{\pm0.218}$ | $0.471_{\pm0.304}$ | $\underline{0.645}_{\pm0.277}$ | $\mathbf{0.773}_{\pm0.272}$ |
| Shape | $0.965_{\pm0.019}$ | $0.893_{\pm0.062}$ | $0.896_{\pm0.051}$ | $0.947_{\pm0.075}$ | $0.955_{\pm0.031}$ | $0.957_{\pm0.053}$ | $\underline{0.980}_{\pm0.011}$ | $\mathbf{0.983}_{\pm0.013}$ |
| Shape (test) | $0.958_{\pm0.022}$ | $0.892_{\pm0.060}$ | $0.894_{\pm0.051}$ | $0.941_{\pm0.073}$ | $0.951_{\pm0.032}$ | $0.951_{\pm0.052}$ | $\underline{0.972}_{\pm0.013}$ | $\mathbf{0.976}_{\pm0.015}$ |
| Shape (cat) | $\mathbf{0.994}_{\pm0.005}$ | $0.900_{\pm0.089}$ | $0.870_{\pm0.084}$ | $0.945_{\pm0.100}$ | $0.974_{\pm0.022}$ | $0.972_{\pm0.066}$ | $\underline{0.986}_{\pm0.018}$ | $0.985_{\pm0.022}$ |
| Shape (num) | $0.943_{\pm0.036}$ | $0.902_{\pm0.041}$ | $0.904_{\pm0.033}$ | $0.952_{\pm0.057}$ | $0.949_{\pm0.035}$ | $0.959_{\pm0.043}$ | $\underline{0.973}_{\pm0.028}$ | $\mathbf{0.986}_{\pm0.007}$ |
| WD (num) | $0.014_{\pm0.014}$ | $0.024_{\pm0.012}$ | $0.023_{\pm0.011}$ | $0.013_{\pm0.015}$ | $0.018_{\pm0.019}$ | $0.017_{\pm0.025}$ | $\underline{0.008}_{\pm0.010}$ | $\mathbf{0.003}_{\pm0.002}$ |
| WD (num, test) | $0.017_{\pm0.017}$ | $0.026_{\pm0.014}$ | $0.024_{\pm0.013}$ | $0.016_{\pm0.015}$ | $0.020_{\pm0.021}$ | $0.020_{\pm0.027}$ | $\underline{0.010}_{\pm0.011}$ | $\mathbf{0.006}_{\pm0.007}$ |
| JSD (cat) | $\mathbf{0.008}_{\pm0.006}$ | $0.119_{\pm0.097}$ | $0.137_{\pm0.082}$ | $0.066_{\pm0.109}$ | $0.035_{\pm0.028}$ | $0.030_{\pm0.066}$ | $\underline{0.017}_{\pm0.019}$ | $0.018_{\pm0.024}$ |
| JSD (cat, test) | $\mathbf{0.016}_{\pm0.013}$ | $0.121_{\pm0.096}$ | $0.138_{\pm0.081}$ | $0.070_{\pm0.108}$ | $0.039_{\pm0.031}$ | $0.037_{\pm0.064}$ | $\underline{0.024}_{\pm0.021}$ | $0.025_{\pm0.026}$ |
| Trend | $0.957_{\pm0.029}$ | $0.855_{\pm0.115}$ | $0.809_{\pm0.118}$ | $0.906_{\pm0.141}$ | $0.925_{\pm0.057}$ | $0.927_{\pm0.105}$ | $\underline{0.961}_{\pm0.036}$ | $\mathbf{0.964}_{\pm0.037}$ |
| Trend (test) | $0.927_{\pm0.041}$ | $0.845_{\pm0.112}$ | $0.806_{\pm0.114}$ | $0.881_{\pm0.133}$ | $0.904_{\pm0.059}$ | $0.899_{\pm0.099}$ | $\underline{0.936}_{\pm0.038}$ | $\mathbf{0.937}_{\pm0.041}$ |
| Trend (mixed) | $\underline{0.945}_{\pm0.039}$ | $0.797_{\pm0.096}$ | $0.685_{\pm0.097}$ | $0.898_{\pm0.142}$ | $0.909_{\pm0.056}$ | $0.923_{\pm0.085}$ | $0.937_{\pm0.050}$ | $\mathbf{0.948}_{\pm0.040}$ |
| MLE | $0.069_{\pm0.069}$ | $0.082_{\pm0.081}$ | $0.144_{\pm0.109}$ | $0.058_{\pm0.056}$ | $0.057_{\pm0.037}$ | $0.048_{\pm0.031}$ | $\underline{0.040}_{\pm0.031}$ | $\mathbf{0.030}_{\pm0.040}$ |
| $\alpha$-Precision | $0.934_{\pm0.071}$ | $0.766_{\pm0.262}$ | $0.839_{\pm0.102}$ | $0.859_{\pm0.203}$ | $0.914_{\pm0.140}$ | $0.875_{\pm0.182}$ | $\underline{0.976}_{\pm0.023}$ | $\mathbf{0.979}_{\pm0.024}$ |
| $\beta$-Recall | $0.331_{\pm0.127}$ | $0.229_{\pm0.201}$ | $0.167_{\pm0.120}$ | $0.438_{\pm0.240}$ | $0.267_{\pm0.151}$ | $0.330_{\pm0.215}$ | $\underline{0.538}_{\pm0.154}$ | $\mathbf{0.592}_{\pm0.126}$ |
| DCR Share | $0.797_{\pm0.024}$ | $0.811_{\pm0.051}$ | $\mathbf{0.779}_{\pm0.021}$ | $0.840_{\pm0.077}$ | $\underline{0.786}_{\pm0.018}$ | $0.801_{\pm0.044}$ | $0.854_{\pm0.078}$ | $0.889_{\pm0.077}$ |
| MIA Score | $0.976_{\pm0.016}$ | $\underline{0.977}_{\pm0.021}$ | $\mathbf{0.982}_{\pm0.026}$ | $0.961_{\pm0.034}$ | $0.977_{\pm0.019}$ | $0.975_{\pm0.019}$ | $0.960_{\pm0.032}$ | $0.944_{\pm0.031}$ |

### A.9.2. ABLATION EXPERIMENT ON VARYING THE MISSINGNESS RATE

*Table 22.* Ablation results on varying the missingness rate averaged over all datasets and sampling seeds for a single training run. We report the standard deviation over the 12 datasetes. A missingness rate of $p = 0.10$ was used for the main results.

| | TabCascade | | | CDTD | | |
|---|---|---|---|---|---|---|
| | $p = 0.10$ | $p = 0.25$ | $p = 0.50$ | $p = 0.10$ | $p = 0.25$ | $p = 0.50$ |
| Detection Score | $0.782_{\pm0.249}$ | $0.765_{\pm0.252}$ | $0.781_{\pm0.250}$ | $0.519_{\pm0.292}$ | $0.503_{\pm0.312}$ | $0.537_{\pm0.295}$ |
| Shape | $0.983_{\pm0.008}$ | $0.979_{\pm0.010}$ | $0.975_{\pm0.012}$ | $0.970_{\pm0.012}$ | $0.962_{\pm0.018}$ | $0.955_{\pm0.021}$ |
| Shape (test) | $0.976_{\pm0.011}$ | $0.971_{\pm0.012}$ | $0.966_{\pm0.013}$ | $0.964_{\pm0.012}$ | $0.956_{\pm0.016}$ | $0.949_{\pm0.021}$ |
| Shape (cat) | $0.986_{\pm0.014}$ | $0.984_{\pm0.018}$ | $0.984_{\pm0.015}$ | $0.984_{\pm0.017}$ | $0.983_{\pm0.017}$ | $0.983_{\pm0.017}$ |
| Shape (num) | $0.985_{\pm0.006}$ | $0.981_{\pm0.007}$ | $0.973_{\pm0.014}$ | $0.962_{\pm0.018}$ | $0.949_{\pm0.032}$ | $0.940_{\pm0.028}$ |
| WD (num) | $0.004_{\pm0.003}$ | $0.005_{\pm0.003}$ | $0.007_{\pm0.005}$ | $0.009_{\pm0.006}$ | $0.011_{\pm0.007}$ | $0.012_{\pm0.006}$ |
| WD (num, test) | $0.007_{\pm0.007}$ | $0.009_{\pm0.008}$ | $0.011_{\pm0.008}$ | $0.012_{\pm0.009}$ | $0.014_{\pm0.008}$ | $0.016_{\pm0.010}$ |
| JSD (cat) | $0.018_{\pm0.014}$ | $0.021_{\pm0.021}$ | $0.021_{\pm0.017}$ | $0.020_{\pm0.018}$ | $0.021_{\pm0.018}$ | $0.024_{\pm0.020}$ |
| JSD (cat, test) | $0.024_{\pm0.017}$ | $0.027_{\pm0.022}$ | $0.027_{\pm0.019}$ | $0.026_{\pm0.020}$ | $0.027_{\pm0.020}$ | $0.028_{\pm0.021}$ |
| Trend | $0.965_{\pm0.027}$ | $0.962_{\pm0.029}$ | $0.963_{\pm0.022}$ | $0.957_{\pm0.032}$ | $0.953_{\pm0.030}$ | $0.954_{\pm0.028}$ |
| Trend (test) | $0.941_{\pm0.032}$ | $0.938_{\pm0.032}$ | $0.941_{\pm0.027}$ | $0.935_{\pm0.034}$ | $0.934_{\pm0.030}$ | $0.937_{\pm0.031}$ |
| Trend (mixed) | $0.947_{\pm0.032}$ | $0.944_{\pm0.029}$ | $0.952_{\pm0.023}$ | $0.930_{\pm0.039}$ | $0.928_{\pm0.036}$ | $0.938_{\pm0.026}$ |
| MLE | $0.025_{\pm0.023}$ | $0.028_{\pm0.026}$ | $0.027_{\pm0.021}$ | $0.038_{\pm0.038}$ | $0.043_{\pm0.058}$ | $0.039_{\pm0.037}$ |
| $\alpha$-Precision | $0.976_{\pm0.025}$ | $0.968_{\pm0.055}$ | $0.966_{\pm0.042}$ | $0.971_{\pm0.044}$ | $0.971_{\pm0.041}$ | $0.965_{\pm0.034}$ |
| $\beta$-Recall | $0.586_{\pm0.110}$ | $0.595_{\pm0.120}$ | $0.579_{\pm0.115}$ | $0.577_{\pm0.127}$ | $0.630_{\pm0.119}$ | $0.600_{\pm0.147}$ |
| DCR Share | $0.889_{\pm0.080}$ | $0.892_{\pm0.084}$ | $0.905_{\pm0.082}$ | $0.883_{\pm0.070}$ | $0.896_{\pm0.076}$ | $0.899_{\pm0.074}$ |
| MIA Score | $0.937_{\pm0.042}$ | $0.938_{\pm0.044}$ | $0.947_{\pm0.036}$ | $0.959_{\pm0.038}$ | $0.960_{\pm0.038}$ | $0.959_{\pm0.033}$ |

### A.9.3. ABLATION EXPERIMENT USING ARF AS LOW-RESOLUTION MODEL

*Table 23.* Ablation results for changing the low-resolution model $p_{\text{low}}^{\theta}$ in TabCascade from CDTD to ARF, while keeping the high-resolution model $p_{\text{high}}^{\theta}$ fixed. The high-resolution model is not re-trained. Results are obtained from a single training run but averaged over all datasets and ten sampling seeds. We report the standard deviation over the 12 datasets. The best, row-wise result is indicated in **bold**, the second best is underlined. With JSD ($\mathbf{x}_{\text{low}}$) we indicate the Jensen-Shannon divergence computed from the true $\mathbf{x}_{\text{low}}$ and samples produced by the respective low-resolution models.

| Metric | ARF | TVAE | CTGAN | TabDDPM | TabSyn | TabDiff | CDTD | Ours (DT) | Ours ($p_{\text{low}}^{\theta}$ = ARF) |
|---|---|---|---|---|---|---|---|---|---|
| Detection Score | $0.290_{\pm0.189}$ | $0.207_{\pm0.264}$ | $0.070_{\pm0.073}$ | $0.475_{\pm0.373}$ | $0.187_{\pm0.164}$ | $0.421_{\pm0.286}$ | $\underline{0.519}_{\pm0.292}$ | $\mathbf{0.782}_{\pm0.249}$ | $0.494_{\pm0.279}$ |
| Shape | $0.957_{\pm0.019}$ | $0.894_{\pm0.053}$ | $0.893_{\pm0.034}$ | $0.938_{\pm0.067}$ | $0.927_{\pm0.037}$ | $0.955_{\pm0.047}$ | $0.970_{\pm0.012}$ | $\underline{0.983}_{\pm0.008}$ | $\mathbf{0.990}_{\pm0.007}$ |
| Shape (cat) | $\underline{0.993}_{\pm0.005}$ | $0.904_{\pm0.085}$ | $0.892_{\pm0.047}$ | $0.935_{\pm0.086}$ | $0.947_{\pm0.040}$ | $0.973_{\pm0.060}$ | $0.984_{\pm0.017}$ | $0.986_{\pm0.014}$ | $\mathbf{0.993}_{\pm0.005}$ |
| Shape (num) | $0.932_{\pm0.028}$ | $0.899_{\pm0.035}$ | $0.896_{\pm0.024}$ | $0.942_{\pm0.055}$ | $0.920_{\pm0.041}$ | $0.952_{\pm0.036}$ | $0.962_{\pm0.018}$ | $\underline{0.985}_{\pm0.006}$ | $\mathbf{0.989}_{\pm0.008}$ |
| WD (num) | $0.016_{\pm0.013}$ | $0.023_{\pm0.011}$ | $0.025_{\pm0.012}$ | $0.015_{\pm0.018}$ | $0.031_{\pm0.031}$ | $0.015_{\pm0.021}$ | $0.009_{\pm0.006}$ | $\underline{0.004}_{\pm0.003}$ | $\mathbf{0.003}_{\pm0.004}$ |
| JSD (cat) | $\underline{0.009}_{\pm0.006}$ | $0.129_{\pm0.110}$ | $0.115_{\pm0.049}$ | $0.085_{\pm0.105}$ | $0.066_{\pm0.049}$ | $0.030_{\pm0.061}$ | $0.020_{\pm0.018}$ | $0.018_{\pm0.014}$ | $\mathbf{0.008}_{\pm0.006}$ |
| Trend | $0.944_{\pm0.040}$ | $0.853_{\pm0.112}$ | $0.819_{\pm0.102}$ | $0.901_{\pm0.126}$ | $0.890_{\pm0.069}$ | $0.924_{\pm0.101}$ | $0.957_{\pm0.032}$ | $\mathbf{0.965}_{\pm0.027}$ | $\underline{0.957}_{\pm0.019}$ |
| Trend (mixed) | $0.935_{\pm0.031}$ | $0.788_{\pm0.115}$ | $0.727_{\pm0.087}$ | $0.869_{\pm0.134}$ | $0.862_{\pm0.059}$ | $0.920_{\pm0.087}$ | $0.930_{\pm0.039}$ | $\mathbf{0.947}_{\pm0.032}$ | $\underline{0.946}_{\pm0.030}$ |
| MLE | $0.065_{\pm0.055}$ | $0.077_{\pm0.080}$ | $0.130_{\pm0.078}$ | $0.327_{\pm0.982}$ | $0.345_{\pm0.975}$ | $0.045_{\pm0.027}$ | $\underline{0.038}_{\pm0.038}$ | $\mathbf{0.025}_{\pm0.023}$ | $0.052_{\pm0.039}$ |
| $\alpha$-Precision | $0.960_{\pm0.029}$ | $0.716_{\pm0.301}$ | $0.847_{\pm0.073}$ | $0.757_{\pm0.294}$ | $0.871_{\pm0.159}$ | $0.917_{\pm0.100}$ | $0.971_{\pm0.044}$ | $\underline{0.976}_{\pm0.025}$ | $\mathbf{0.982}_{\pm0.016}$ |
| $\beta$-Recall | $0.348_{\pm0.095}$ | $0.269_{\pm0.217}$ | $0.201_{\pm0.116}$ | $0.460_{\pm0.273}$ | $0.243_{\pm0.106}$ | $0.384_{\pm0.189}$ | $\underline{0.577}_{\pm0.127}$ | $\mathbf{0.586}_{\pm0.110}$ | $0.380_{\pm0.089}$ |
| DCR Share | $0.807_{\pm0.014}$ | $0.829_{\pm0.051}$ | $\mathbf{0.788}_{\pm0.010}$ | $0.864_{\pm0.073}$ | $\underline{0.788}_{\pm0.024}$ | $0.800_{\pm0.046}$ | $0.883_{\pm0.070}$ | $0.889_{\pm0.080}$ | $0.810_{\pm0.015}$ |
| MIA Score | $0.971_{\pm0.023}$ | $0.970_{\pm0.030}$ | $\mathbf{0.982}_{\pm0.018}$ | $0.954_{\pm0.036}$ | $\underline{0.980}_{\pm0.016}$ | $0.972_{\pm0.021}$ | $0.959_{\pm0.038}$ | $0.937_{\pm0.042}$ | $0.963_{\pm0.027}$ |
| JSD ($\mathbf{x}_{\text{low}}$) | - | - | - | - | - | - | - | $\underline{0.018}_{\pm0.013}$ | $\mathbf{0.008}_{\pm0.006}$ |

### A.9.4. ABLATION EXPERIMENTS VARYING ENCODER COMPLEXITY

*Table 24.* The effect of maximum depth of the DT encoder on various evaluation metrics averaged over datasets and ten sampling seeds for a single training run. We report the standard deviation over the 12 datasets. Grey indicates the maximum depth used for the main results.

| Max. Depth | Detection Score | Shape (num) | WD (num) | Trend | Trend (mixed) | MLE | $\alpha$-Precision | $\beta$-Recall | MIA Score |
|---|---|---|---|---|---|---|---|---|---|
| 3 | $0.664_{\pm0.259}$ | $0.979_{\pm0.006}$ | $0.005_{\pm0.002}$ | $0.960_{\pm0.033}$ | $0.937_{\pm0.041}$ | $0.026_{\pm0.020}$ | $0.977_{\pm0.018}$ | $0.592_{\pm0.105}$ | $0.945_{\pm0.043}$ |
| 4 | $0.695_{\pm0.239}$ | $0.981_{\pm0.006}$ | $0.005_{\pm0.003}$ | $0.962_{\pm0.029}$ | $0.941_{\pm0.038}$ | $0.026_{\pm0.021}$ | $0.982_{\pm0.012}$ | $0.600_{\pm0.107}$ | $0.942_{\pm0.044}$ |
| 5 | $0.701_{\pm0.237}$ | $0.983_{\pm0.005}$ | $0.004_{\pm0.002}$ | $0.960_{\pm0.036}$ | $0.939_{\pm0.040}$ | $0.032_{\pm0.028}$ | $0.982_{\pm0.012}$ | $0.588_{\pm0.111}$ | $0.941_{\pm0.042}$ |
| 6 | $0.737_{\pm0.235}$ | $0.983_{\pm0.006}$ | $0.004_{\pm0.003}$ | $0.960_{\pm0.032}$ | $0.937_{\pm0.036}$ | $0.031_{\pm0.026}$ | $0.981_{\pm0.012}$ | $0.587_{\pm0.115}$ | $0.944_{\pm0.037}$ |
| 7 | $0.759_{\pm0.247}$ | $0.984_{\pm0.006}$ | $0.004_{\pm0.003}$ | $0.961_{\pm0.032}$ | $0.939_{\pm0.035}$ | $0.031_{\pm0.028}$ | $0.979_{\pm0.013}$ | $0.590_{\pm0.111}$ | $0.940_{\pm0.041}$ |
| 8 | $0.788_{\pm0.251}$ | $0.985_{\pm0.006}$ | $0.004_{\pm0.003}$ | $0.965_{\pm0.027}$ | $0.947_{\pm0.032}$ | $0.024_{\pm0.024}$ | $0.976_{\pm0.025}$ | $0.586_{\pm0.110}$ | $0.937_{\pm0.042}$ |
| 9 | $0.774_{\pm0.261}$ | $0.984_{\pm0.006}$ | $0.004_{\pm0.002}$ | $0.964_{\pm0.030}$ | $0.945_{\pm0.033}$ | $0.026_{\pm0.030}$ | $0.972_{\pm0.030}$ | $0.587_{\pm0.119}$ | $0.936_{\pm0.040}$ |

We systematically investigate the effect of the DT encoder's complexity by varying its maximum depth (max_depth) hyperparameter from 3 to 9. Figure 25 shows the impact of increasing max_depth on the proportion of masked inputs to the high-resolution model. For comparison, Figure 24 shows the same for increasing the complexity of the GMM encoder. For features that are integer-valued with few unique values, increasing maximum depth can lead to cases where each unique value is treated as a separate component. In these cases, the feature would be entirely generated by the low-resolution model.

Further, we investigate the effect of max_depth on various sample quality metrics. Table 24 gives the average results over all datasets with 10 different synthetic samples each. For each setting, we adjusted the model parameters to approx. 1 million parameters for the high-resolution model and approx. 2 million parameters for the low-resolution model on the `adult` dataset. We emphasize that the effect of max depth may be different for different architectures but an exhaustive evaluation of all combinations is prohibitively expensive.

Increasing max_depth increases the number of Gaussian components. This appears to make samples substantially more realistic in the eyes of the gradient-boosting-based detection model whereas it has a less pronounced effect on the other metrics. Note that with a max depth of only 3, TabCascade still outperforms the baselines in terms of sample realism, illustrating the general benefit of our cascaded pipeline. Increasing max_depth then enables us to navigate the fidelity-privacy trade-off. Specifically, we can achieve a greater average detection score at the cost of a lower average MIA score. The best choice for max_depth also depends on which metrics are deemed to be most relevant in a given modeling context. If, for instance, $\alpha$-Precision and $\beta$-Recall are presumed to be more important than the detection score, more favorable results could be achieved by lowering max depth to 4.

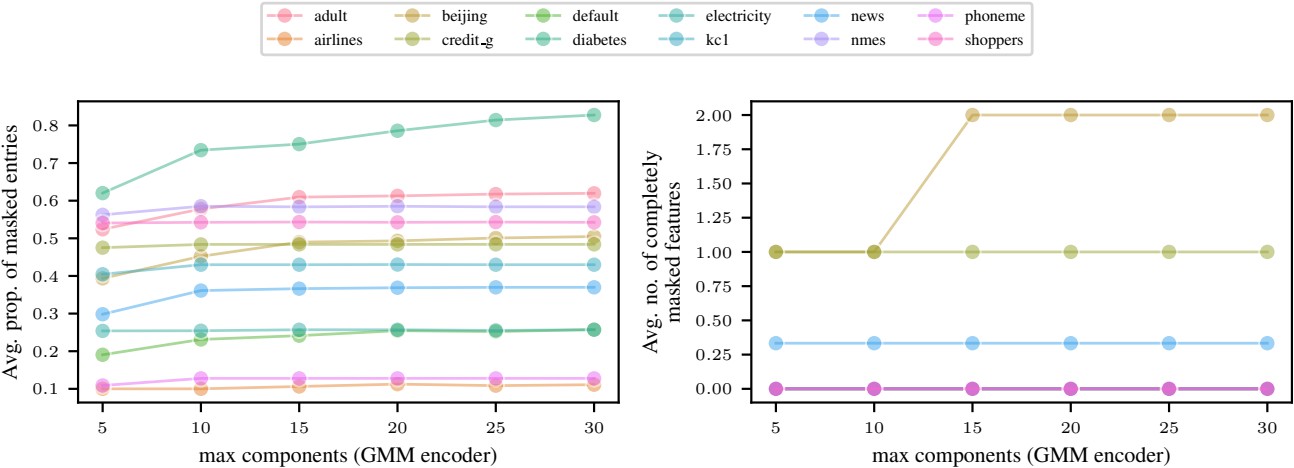

*Figure 24.* Effect of varying the maximum possible number of Gaussian components in the GMM encoder on the average (over three training seeds) proportion of masked inputs to $p_{\text{high}}^{\boldsymbol{\theta}}$ and the average number of completely masked features.

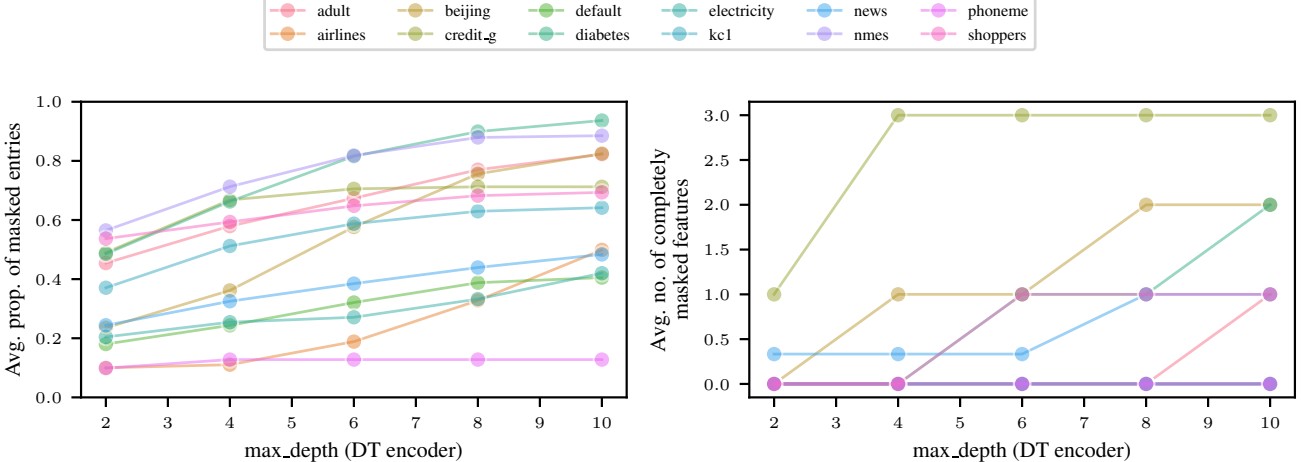

*Figure 25.* Effect of varying the maximum tree depth in the DT encoder on the average (over three training seeds) proportion of masked inputs to $p_{\text{high}}^{\boldsymbol{\theta}}$ and the average number of completely masked features.

## A.10. Training and Sampling Times

*Table 25.* Training times in minutes. For diffusion-based models, the training time was capped at 30 minutes.

|             | ARF  | TVAE | CTGAN | TabDDPM | TabSyn | TabDiff | CDTD | Ours (DT) |
|-------------|------|------|-------|---------|--------|---------|------|-----------|
| adult       | 11.4 | 20.0 | 36.2  | 9.5     | 14.4   | 30.0    | 6.0  | 11.1      |
| airlines    | 97.5 | 20.1 | 48.0  | 10.4    | 6.4    | 15.9    | 5.6  | 10.4      |
| beijing     | 10.6 | 21.5 | 35.3  | 8.1     | 13.2   | 30.0    | 5.6  | 11.3      |
| credit_g    | 0.1  | 8.4  | 22.2  | 9.6     | 9.0    | 30.0    | 5.4  | 10.2      |
| default     | 14.7 | 25.1 | 44.1  | 11.9    | 19.4   | 30.0    | 6.6  | 11.5      |
| diabetes    | 56.0 | 29.5 | 101.8 | 30.0    | 16.2   | 30.0    | 8.0  | 13.2      |
| electricity | 10.9 | 19.7 | 31.0  | 8.1     | 11.6   | 29.7    | 5.6  | 10.4      |
| kc1         | 1.0  | 15.6 | 32.3  | 9.9     | 10.9   | 30.0    | 5.4  | 10.9      |
| news        | 38.7 | 41.7 | 68.2  | 21.1    | 30.0   | 30.0    | 9.2  | 16.8      |
| nmes        | 1.1  | 17.7 | 33.8  | 9.7     | 12.1   | 30.0    | 5.7  | 10.1      |
| phoneme     | 0.2  | 17.6 | 29.8  | 6.7     | 9.0    | 30.0    | 5.1  | 9.1       |
| shoppers    | 3.6  | 24.1 | 39.2  | 10.4    | 14.3   | 30.0    | 6.2  | 11.1      |

*Table 26.* Sample times in seconds per 1000 samples. TabDDPM produces NaNs for `airlines`, `diabetes` and `news` datasets.

|             | ARF  | TVAE | CTGAN | TabDDPM | TabSyn | TabDiff | CDTD | Ours (DT) |
|-------------|------|------|-------|---------|--------|---------|------|-----------|
| adult       | 1.55 | 0.14 | 0.24  | 7.08    | 0.53   | 3.62    | 2.55 | 0.67      |
| airlines    | 2.13 | 0.05 | 0.05  | -       | 0.39   | 3.16    | 0.67 | 0.58      |
| beijing     | 1.09 | 0.14 | 0.23  | 5.32    | 0.55   | 2.24    | 3.76 | 0.61      |
| credit_g    | 1.93 | 0.08 | 0.08  | 8.28    | 0.41   | 1.50    | 1.26 | 0.73      |
| default     | 2.43 | 0.18 | 0.29  | 10.19   | 0.56   | 3.47    | 6.38 | 0.76      |
| diabetes    | 4.46 | 0.22 | 0.32  | -       | 0.53   | 24.11   | 3.54 | 0.88      |
| electricity | 0.94 | 0.06 | 0.05  | 4.98    | 0.38   | 0.63    | 0.73 | 0.61      |
| kc1         | 2.18 | 0.11 | 0.11  | 8.43    | 0.43   | 1.34    | 1.30 | 0.73      |
| news        | 6.76 | 0.34 | 0.44  | -       | 0.60   | 7.69    | 5.26 | 1.14      |
| nmes        | 1.67 | 0.08 | 0.08  | 7.39    | 0.40   | 1.13    | 1.12 | 0.70      |
| phoneme     | 0.56 | 0.04 | 0.04  | 4.07    | 0.38   | 0.44    | 0.58 | 0.56      |
| shoppers    | 1.71 | 0.18 | 0.25  | 7.45    | 0.54   | 3.20    | 2.90 | 0.69      |

