# OpenReview forum: "Cascaded Flow Matching for Heterogeneous Tabular Data with Mixed-Type Features"
_ICML.cc/2026/Conference — ICML 2026 regular_

### Official Review · Reviewer_RrbT · 2026-03-04

**Soundness:** 3
**Presentation:** 3
**Significance:** 3
**Originality:** 3
**Overall Recommendation:** 5
**Confidence:** 4

**Summary:**

This paper proposes TabCascade, a cascaded flow matching framework for tabular data with mixed-type features. The paper identifies two challenges: (1) tabular datasets often contain both categorical and numerical features across columns, and (2) individual numerical features may exhibit mixed-type behavior, where a continuous variable contains probability mass at specific discrete values. To address this, the framework adopts a two-stage generation process. First, a low-resolution model generates categorical features and discretized representations of numerical features using bins learned via Gaussian Mixture Models. Then, a high-resolution flow matching model generates continuous numerical values conditioned on these bins. Extensive experiments and ablation studies are conducted to evaluate the proposed approach.

**Compliance With Llm Reviewing Policy:**

Affirmed.

**Final Justification:**

I am keeping my score (5, accept) because in my original review, I already thought the paper was in very good shape and now that all concerns are addressed, I believe it should be accepted.

**Key Questions For Authors:**

I still need clarification on why the model needs to generate missing values. Are there realistic downstream scenarios where this capability is required?

I would also like to see the individual performance of the low- and high-resolution models in the ablation studies. In particular, how accurately can the low-resolution model assign samples to the correct bins, since this step serves as the precursor to the high-resolution model?

The authors mention that DT identify more bins and therefore outperform GMMs. It is unclear whether the improvement mainly comes from having more bins. If so, would simply increasing the number of bins (e.g., using a fixed large number) lead to similar improvements (and does that lead to more compute overhead)?

**Limitations:**

Yes

**Strengths And Weaknesses:**

The overall framework is well presented. The challenge of learning generative models for tabular data is clearly articulated: tabular datasets often contain both continuous and discrete features that may be coupled, and even continuous features may exhibit mixed-type behavior. Given this motivation, the proposed framework follows naturally and is intuitive, resembling the idea of super-resolution models used in other machine learning modalities.

The method also appears technically sound. The low-resolution and high-resolution components are clearly defined, and different design choices and parameters are studied through ablation experiments.

Overall, the paper addresses a meaningful challenge, and the proposed approach is both significant and original.

---

> ### Author Rebuttal · Authors · 2026-03-30
>
> We sincerely thank you for your time investment and the high effort you put into understanding our work. We are also thankful that you acknowledge that our paper addresses an important and meaningful challenge in tabular data generation. Below, we address your questions:
>
> > I still need clarification on why the model needs to generate missing values. Are there realistic downstream scenarios where this capability is required?
>
> The short answer is: This is required whenever missing values may represent informative signal. For example, researchers in the social sciences and economics, medicine or psychology often face data with missing values that actually convey meaning. It might be that a missing entry indicates that a medical test was not conducted at all because a patient canceled their appointment. This missingness will not be completely random, a patient is more likely to skip an appointment when they either have no acute health issue or are very ill. Another use case appears when researchers would like to use synthetic data to develop or test a model that should work in the presence of missing values. In this case the synthetic data must include realistic missing values as well. A final example would be the development of imputation techniques on sensitive data, which cannot be shared directly (e.g., on financial information). For further details, we refer to the seminal works [1], [2], [3] and [4], which we will add to the introduction in our final version. Lastly, note that in Table 17 we show that our model also outperforms the baselines when there are no missings in the data. Hence, our model *can* generate missing values but it has value beyond that.
>
>
> [1] Molenberghs, G., Fitzmaurice, G., Kenward, M.G., Tsiatis, A., & Verbeke, G. (Eds.) (2014) Handbook of Missing Data Methodology.
>
> [2] Little, A. and Rubin, D.B. (2019) Statistical Analysis with Missing Data. Third Edition.
>
> [3] Daniels, M.J. and Hogan, J.W. (2008) Missing Data in Longitudinal Studies: Strategies for Bayesian Modeling and Sensitivity Analysis.
>
> [4] Molenberghs, G. and Kenward, M.G. (2007) Missing Data in Clinical Studies.
>
>
> > I would also like to see the individual performance of the low- and high-resolution models in the ablation studies. In particular, how accurately can the low-resolution model assign samples to the correct bins [...] ?
>
> Indirectly, this evaluation is already provided in the main results. If the low-resolution does not allocate the sample to the correct bin, the high-resolution model starts from a very sub-optimal prior distribution due to the data-dependent coupling. In such a case, we would see reduced performance when comparing CDTD to TabCascasde, which is not the case.
>
> Additionally, we computed the average (over all datasets and sampling seeds) Jensen-Shannon divergence (JSD) for $x_{\text{low}} = (x_{\text{cat}}, z)$, comparing the true distribution and the distribution learned by the ARF-based and CDTD-based low-resolution models in Table 19. For the ARF-based model we find a JSD of 0.008 (+- 0.006), whereas for the CDTD-based model we find 0.018 (+- 0.013). These are virtually identical to the JSD (cat) metric in the same table, which is only computed on categorical features and excludes the latent representations. This indicates that both models very accurately track the true distribution of the latents. It is not possible to evaluate the individual performance of the high-resolution model, as we necessarily need low-resolution samples for that.
>
>
> > The authors mention that DT identify more bins and therefore outperform GMMs. It is unclear whether the improvement mainly comes from having more bins. If so, would simply increasing the number of bins (e.g., using a fixed large number) lead to similar improvements (and does that lead to more compute overhead)?
>
> We mention some important differences in section A.5.2. Accordingly, DT directly optimizes for a hard clustering of samples into Gaussian components and constructs components on partitions of the data, which significantly reduce the overlap of the components compared to GMM. We use these characteristics to prove our theorem that DT encoding and data-dependent coupling reduces the transport cost bound. It is not possible to prove this for GMM due to the overlap of the Gaussian components. The benefits of DT therefore not only lie in the higher number of bins. We could also use GMM and force it to use a greater number of bins but this comes with two downsides: 1) GMM will take much longer to train than DT and 2) many components will be ``redundant'', leading to significant overlap. Overlap reduces the benefit of data-dependent coupling since it reduces the precision with which we can move a sample towards its target. In particular, note that GMM does soft-clustering and together with overlapping components this can lead to groups which are actually disconnected on the domain of $x_i$ (hence their mean is not informative of the true group mean).

---

> > ### Author Rebuttal · Reviewer_RrbT · 2026-04-01
> >
> > My concerns are resolved. I want to keep my score as 5-Accept.

---

### Official Review · Reviewer_fuNz · 2026-03-07

**Soundness:** 3
**Presentation:** 4
**Significance:** 2
**Originality:** 3
**Overall Recommendation:** 5
**Confidence:** 4

**Summary:**

The paper tackles the problem of generative modeling for tabular data. The authors make a case that the main difficulty in generating mixed-type data is the generation of numerical features and propose a cascaded approach whereby categorical features are generated first, together with a discrete "low-resolution" version of the numerical features, using a diffusion approach. Then, the numerical features are generated, conditioned on the categorical features and the low-resolution numerical representation using a flow matching approach with a non-independent coupling.
The proposed approach is compared with prior works on a wide selection of datasets and evaluation metrics, showing improved synthetic data utility metrics.

**Compliance With Llm Reviewing Policy:**

Affirmed.

**Final Justification:**

I would like to thank the authors for their efforts in getting to the bottom of the issues I raised. Since all of my remaining concerns have been addressed I would like to increase my score to 5, recommending acceptance.

**Key Questions For Authors:**

1. Would careful post-processing (e.g., rounding) of generated numerical features have a high impact on detection score (and other metrics) for the different methods?
2. Why do the DCR share scores reported in Table 15 differ so drastically from those presented in previous works [1, 2]
3. Is the proposed method (or the other baselines) overfitting to the training data? How would the synthetic data quality metrics change if they were compared to test data instead of training data?

[[1]](https://arxiv.org/abs/2310.09656) Zhang et al. Mixed-Type Tabular Data Synthesis with Score-based Diffusion in Latent Space.
[[2]](https://arxiv.org/abs/2310.09656) Shi et al. TabDiff: a Mixed-type Diffusion Model for Tabular Data Generation.

**Limitations:**

I believe the authors could improve the discussion around the privacy results presented in the paper (see Strengths and Weaknesses section).

**Strengths And Weaknesses:**

### Strengths:
1. The paper is well-written and polished. It presents a coherent story and the proposed approach is clearly explained and well-motivated.
2. The main idea of the paper is very sensible and natural. As far as I am aware, it is the first time these ideas are applied within the context of tabular generative models.
3. The experimental section is, for the most part, very solid. The baseline selection includes most of the relevant recent tabular generative models and the dataset selection goes beyond the typical recent benchmarks in this domain. The set of evaluation metrics is also extensive, covering most of the metrics that have been used to measure the quality of synthetic tabular data.
4. While the experimental results are summarized in the main paper, there are lots of details and extra results in appendices and the authors provide fully reproducible code.

### Weaknesses
I have two major concerns about the paper (1-2) and a minor one (3).
1. **Pathologies of numerical features.** I suspect that the apparent difficulty in modeling numerical features (as evidenced by the detection score) is not necessarily due to them being harder to model, but due the many pathologies that affect this type of data. While the authors explicitly tackle some of these pathologies (such as the existence of atoms in otherwise continuous features) they don’t address other known ones such as those detailed in [1] or, perhaps more importantly, (originally) continuous features that are recorded/stored in finite precision in csv formats, effectively making them discrete. The latter can make it very easy for a model such as LightGBM to distinguish generated from real data, unless the former is appropriately rounded or directly generated as discrete (which does not seems to be the case looking at the provided code). The detection scores presented in Figure 23 of the Appendix, make me question the central thesis of the paper, particularly given that the proposed solution does not meaningfully improve the detection scores when including numerical features for most of the datasets where the drop appears pathological. Since this can be easily mitigated by rounding the generated samples in post processing (an common option offered in libraries such as SDV), I would like to see, at least, how this affects results, even if it doesn’t cover the full range of pathologies.

2. **Privacy.** While I agree with the author’s statement that methods must be specifically designed with privacy in mind, the authors’ stance towards the privacy metrics seems dismissive. This mainly raises two questions:
    - Prior works have been framed as providing a better trade-off between utility and privacy (compared to SMOTE for example). My interpretation of the papers’ results is that, while the proposed approach provides higher utility, it also seems to provide the worst privacy of the baselines. In particular the results for DCR share make me concerned about the privacy of the generated samples. What applications do the authors have in mind for synthetic data that does not require privacy. Or, to put it another way, if one doesn’t care about the privacy metrics, why not simply use the original data directly?
    - To what extent is the apparent improvement in utility only due to overfitting? Is the proposed method simply learning a distribution that is closer to the empirical training distribution, rather than the underlying  p(x) of the process that generated it? This could be checked by comparing quality metrics to held out (test) data rather than training as done, for example, in [2].

3. **Standard errors.** While the authors provide standard errors for all metrics, these are only with respect to the synthetic data generation and evaluation process itself. That is, the generative model is only trained once on a single fold and evaluated several times by generating different sets of synthetic samples. I would prefer to have a sense of the variance w.r.t. the original generative model initialization and training process (ideally with a k-fold evaluation to also include variation of the training data sampling process). While I am aware that this is not common practice in prior works and I can understand that such an evaluation is more expensive, in my experience, this variance can be wider than that w.r.t. the synthetic sample generating process.

[[1]](https://table-representation-learning.github.io/assets/papers/tabular_data_generation_can_we.pdf) Zein and Urvoy. Tabular Data Generation: Can We Fool XGBoost ? 36th Table Representation Learning workshop (NeurIPS 2022)

[[2]](https://arxiv.org/abs/2309.09968). Jolicoeur-Martineau et al. Generating and Imputing Tabular Data via Diffusion and Flow-based Gradient-Boosted Trees. AISTATS 2024

---

> ### Author Rebuttal · Authors · 2026-03-30
>
> We sincerely thank you for your time investment and high quality review. Below, we address your questions:
>
> > [...] they don’t address other known ones such as those detailed in [1] or [...] continuous features that are recorded/stored in finite precision [...].
>
> The problem of finite precision holds for all types of data. It is not clear what other pathologies you are referring to by citing [1]. Similar to existing work, we use a Quantile Transformer for numerical features. This is similar to the CDF encoder in [1], but this only affects how the input to the *generative* model. For the detection score, we transform all features to their original format, including the precision.
>
>
> > The detection scores presented in Figure 23, make me question the central thesis of the paper [...].
>
> We think there is a slight misinterpretation of this figure. The different "categories" (Cat, Num, Cat and Num) refer to the variables that are included in the *detector*. Across these, the same generative models are used.
> For "Cat and Num", the detector has access to all features, so we expect the score equal to or worse than a detection model that only considers categorical features. We draw two conclusions: (1) categorical features do not give much information to a fake/true detector; (2) our model always outperforms CDTD, regardless of which features the detection model has access to. We will modify the caption of this figure to avoid confusion.
>
>
> > [...] the authors’ stance towards the privacy metrics seems dismissive.
>
> We apologize for how we formulated this. We would describe our stance as *realistic* rather than dismissive. Practical applications of generative models to sensitive data *always* need additional privacy mechanisms tailored to the problem at hand, such that the resulting inference is defendable in court. This constitutes a different part of the literature and is usually model-agnostic. In our work, we focus on designing a new generative model. We will adapt the corresponding text in the paper.
>
>
> > [...] the results for DCR share make me concerned about the privacy [...].
>
> This comes down to the well-known trade-off with fidelity. Producing more realistic samples necessarily means that samples will be closer to the data distribution. We only report DCR for completeness. It has been shown to be uninformative about privacy [1]. Other metrics, such as MIA, should be used. Hence, DCR should not be seen as a privacy metric.
>
> [1] Yao, et al. (2025) The DCR Delusion: Measuring the Privacy Risk of Synthetic Data.
>
>
> > Is the proposed method (or the other baselines) overfitting to the training data?
>
> Evidence against overfitting is that the MIA scores remain high: The gradient-boosting model is not able to predict whether a data sample was used for training the model. If a model overfits, the MIA score would be close to 0, since the synthetic samples would be exact copies of the true data. Below, we also show some metrics computed on the *test* data. They are very similar to the metrics computed on the training data in Table 1, this is consistent with no overfitting. We see similar results for the baselines.
>
> |              |   Ours (DT) |
> |:-------------|------------:|
> | Shape |       0.957 |
> | WD     |       0.008 |
> | JSD   |       0.023 |
> | Trend |       0.939 |
>
>
>
> > Standard errors [...].
>
> We train each generative model three times with different seeds. For each training run, we sample 10 synthetic datasets with different seeds (see section 5.2). The training data also slightly changes due to the simulation process for the missing values. Hence, the variance of model initialization and the training process is already accounted for in the reported standard errors. We will improve the table notes to clarify this.
>
>
> > Would careful post-processing (e.g., rounding) of generated numerical features have a high impact on detection score [...] for the different methods?
>
> We apply the same post-processing to all models to ensure a fair comparison. This includes a rounding operation for numerical features (which can differ in the number of digits per feature). We ensure that all generated data has the *exact same precision* as the original data. Hence, the effect of rounding is fully accounted for.
>
>
> > Why do the DCR share scores reported in Table 15 differ so drastically from those presented in previous works
>
> To investigate this, we changed our DCR evaluation code to match their code exactly. This gives us results that are virtually identical to our original results. From the TabSyn paper, it appears that they re-train the generative model on a 50/50 split, such that the optimal DCR share is 0.5. In our case, the optimal DCR share is size(train) / (size(train) + size(test)) = 0.778, using the formula from their github readme. **This fully explains the discrepancy in reported DCR shares.** The TabSyn results in Table 15 are therefore qualitatively the same as in their paper.

---

> > ### Author Rebuttal · Reviewer_fuNz · 2026-04-02
> >
> > Thank you for the thorough response. This partially resolves my concerns:
> >
> > > It is not clear what other pathologies you are referring to by citing [1]
> >
> > I meant pathologies such as clipped values or numerical features that take only a finite number of discrete values (such as the examples in Figure 4).
> >
> > > For the detection score, we transform all features to their original format, including the precision.
> >
> > Thank you for the clarification. This step escaped me when skimming the code. However, upon closer inspection, I believe there might be a bug in the current implementation. I tried the Beijing dataset and I see that the data preprocessor infers that the TEMP column has 9 digits of precision when it is an integer column. PRES is another integer column that is marked as continuous in the config file and is detected as having 6 digits of precision. I believe this type of errors can have a large impact on the detection score results.
> >
> > > We think there is a slight misinterpretation of this figure. The different "categories" (Cat, Num, Cat and Num) refer to the variables that are included in the detector. Across these, the same generative models are used.
> >
> > That was already my interpretation and in my opinion the legend is clear. To clarify, my point was that the difference between including all columns (Cat + Num) and only categoricals (Cat) appears very large in some datasets (e.g., Beijing, electricity, kc1, news, nmes). While TabCascade does improve this, I would expect it to have a larger impact on the detection score given the main narrative of the paper (making synthetic numericals not as trivial to distinguish from real ones). The fact that it does not is what made me speculate that there might be an easy “defect” in the numeric synthetic data that LightGBM can latch on, such as precision differences.
> >
> > > We train each generative model three times with different seeds
> >
> > Thank you for the clarification. This fully resolves my concern with standard errors.
> >
> > > This fully explains the discrepancy in reported DCR shares
> >
> > Thank you for the clarification. This also resolves my question of the DCR score discrepancy.
> >
> > > In our work, we focus on designing a new generative model.
> >
> > I don't think one can look at utility and fidelity metrics in isolation because a generative model that simply memorizes the training samples is useless. If the proposed approach achieves higher fidelity and utility at the cost of privacy, there is a risk that it has crossed the threshold where it is no longer useful. Hence, why I believe it is important to discuss this as a potential limitation.

---

> > > ### Author Response · Authors · 2026-04-07
> > >
> > > > [...] there might be a bug in the current implementation. I tried the Beijing dataset and I see that the data preprocessor infers that the TEMP column has 9 digits of precision when it is an integer column. [...] this type of errors can have a large impact on the detection score
> > >
> > > Thank you for this follow-up question it is indeed important, and we investigated this thoroughly.
> > >
> > > The TEMP column in the beijing dataset *does* have 9 digits of precision. Even though the data config file indicates TEMP as an integer, we never use this fact in the data processing (lines 71-73 in data/data_preprocess.py). The config file is an artifact from relying on the CDTD github code at the start of our project. This precision issue originates from the original data source. To derive new results, we implemented the following heuristic: For each feature, we retrieve the distribution of decimal digits over all observations. If there are less than 10 values with non-zero decimal digits, we treat the feature as integer-valued. For example, PRES and TEMP only have 4 and 2 observations with non-zero decimal digits, respectively. **This addresses the issue of correctly treating integers**. Note that this does *not* impact model training but only the postprocessing of generated samples.
> > >
> > > Furthermore, **you were right in that precision differences explain the very low detection scores** on some datasets. **We solved this issue entirely**. We traced this back to unexpected behavior of the round function in combination with float64 numbers. We now ensure that the *maximum* precision is 6 decimal digits for both training and synthetic data at time of evaluation. 6 decimal digits can still be represented in float32, which was used for training the models.
> > >
> > > These changes to our processing equalize precision of synthetic and generated data and boost detection scores dramatically. Below, we compare the average detection score for TabCascade under the old and new processing. The news dataset remains difficult, since it includes many continuous features with many decimal digits.
> > >
> > > | | Old | New |
> > > |:------------|------:|------:|
> > > | adult | 0.891 | 0.891 |
> > > | beijing | 0.111 | 0.765 |
> > > | default | 0.579 | 0.579 |
> > > | diabetes | 0.654 | 0.654 |
> > > | news| 0.001 | 0.202 |
> > > | shoppers | 0.389 | 0.961 |
> > > | credit_g | 0.999 | 1 |
> > > | electricity| 0.008 | 0.847 |
> > > | airlines | 0.589 | 0.595 |
> > > | kc1 | 0.029 | 0.974 |
> > > | phoneme | 0.768 | 1 |
> > > | nmes| 0.064 | 0.982 |
> > >
> > > Except for WD, the other metrics are barely impacted. The qualitative results do not change: **TabCascade significantly outperforms all other generative models**. For brevity, we report the updated results for changed metrics and for most diffusion-based models below (see Table 1 in the paper for the original results).
> > >
> > > ||TabSyn |TabDiff | CDTD | Ours (DT) |
> > > |:-------------------|---------:|----------:|-------:|------------:|
> > > | Detection Score | 0.202 |  0.43  |0.518 | 0.787 |
> > > | Shape | 0.927 | 0.954 | 0.97 | 0.984 |
> > > | Shape (num)|0.918 | 0.952 |  0.962 | 0.985 |
> > > | WD (num)| 0.031 | 0.016 | 0.009 | 0.004 |
> > > | MIA Score| 0.981 | 0.974 | 0.958 | 0.935 |
> > >
> > > Since CDTD is the second-best performing model in terms of the detection score, it implies that **using our cascade pipeline improves the average detection score by 51.9%**.
> > >
> > >
> > >
> > > > If the proposed approach achieves higher fidelity and utility at the cost of privacy, there is a risk that it has crossed the threshold where it is no longer useful.
> > >
> > > We agree, this is why we included the MIA score on top of the DCR share. We will add a discussion on the fidelity-privacy trade-off to the results section. We will emphasize the potential negative impact on privacy due to generating more realistic samples and the necessity to carefully examine sample quality with respect to the relevant problem-specific privacy metric in practice. We will cite [1] to indicate that the DCR share should not be interpreted as a privacy metric. We will also highlight that if a practitioner is mainly concerned with privacy in terms of membership inference attacks (MIA), TabCascade only exhibits slightly lower values than existing methods, which also indicates no overfitting.
> > >
> > > [1] Yao, et al. (2025) The DCR Delusion: Measuring the Privacy Risk of Synthetic Data.
> > >
> > >
> > > > I would expect it to have a larger impact on the detection score [...]. The fact that it does not is what made me speculate that there might be an easy “defect” in the numeric synthetic data [...], such as precision differences.
> > >
> > > Indeed, your intuition was correct! With the corrections discussed above, the performance of TabCascade in the "Cat + Num" category gets much closer to that in the "Cat" category compared to CDTD. In the table below, we report the average detection score for each category. Our premise that categorical features are easier to generate accurately still holds.
> > >
> > > | |CDTD | Ours (DT)|
> > > |:----------|-------:|------------:|
> > > |cat + num |0.518 |0.787 |
> > > |cat | 0.918|0.929 |
> > > |num | 0.549| 0.814 |

---

### Official Review · Reviewer_vYYE · 2026-03-12

**Soundness:** 3
**Presentation:** 3
**Significance:** 3
**Originality:** 2
**Overall Recommendation:** 4
**Confidence:** 2

**Summary:**

This paper proposes TabCascade, a cascaded flow-matching framework for heterogeneous tabular data generation that decomposes the task into two stages: a low-resolution stage for modeling categorical variables and discretized numerical surrogates, and a high-resolution conditional stage for generating the final numerical values based on the coarse representation. This design allows the model to naturally handle mixed-type numerical features, such as variables containing both discrete states and continuous values, by encoding discrete regimes early and generating continuous values only when needed. Technically, the high-resolution generator is trained with flow matching under a guided conditional probability path, together with data-dependent coupling and feature-specific nonlinear time schedules to simplify the transport process. Experiments on 12 tabular datasets show that TabCascade consistently outperforms strong baselines, including CTGAN, TVAE, TabDDPM, TabSyn, TabDiff, and CDTD, across multiple evaluation metrics such as detection score, Wasserstein distance for numerical features, and downstream machine-learning utility.

**Compliance With Llm Reviewing Policy:**

Affirmed.

**Final Justification:**

I would first like to thank the authors for their considerable effort during the rebuttal period. Given that some of my concerns have been addressed, I will maintain my original score.

**Key Questions For Authors:**

1. How sensitive is the method to the choice of discretization scheme used to construct the low-resolution representation?

2. Would further hierarchical decomposition (e.g., multi-level cascades) provide additional benefits?

3. How does TabCascade differ from applying existing coarse-to-fine diffusion pipelines directly to tabular data?

4. What is the training and inference overhead relative to TabDiff or TabDDPM?

**Limitations:**

The main weakness is that TabCascade introduces a strong modeling dependence on the quality of the low-resolution representation. Because the high-resolution stage is conditioned on this discretized latent representation, the framework may be sensitive to the choice of encoder and discretization strategy, which could limit its robustness and make the method feel less like a fully general solution.

A secondary limitation is that the overall idea is somewhat incremental from a conceptual standpoint: it adapts a coarse-to-fine cascade to the tabular domain, even though the paper applies this idea effectively to mixed-type features. While the empirical gains are strong, the conceptual advance may not appear substantial enough for a top-tier machine learning venue.

**Strengths And Weaknesses:**

### Strengths

1. The paper identifies a practical yet underexplored issue in tabular data generation: numerical variables often exhibit mixed discrete–continuous structures, such as missing values or zero inflation, which are difficult for standard diffusion- or GAN-based tabular generators to model.

2. The use of conditional probability paths, data-dependent coupling, and learnable time schedules provides a principled way to incorporate the coarse representation into the transport process.

3. The paper evaluates the method on a reasonably large benchmark suite of 12 datasets and compares it against several recent baselines. Multiple metrics are reported, including distributional similarity and downstream machine-learning utility.


### Weaknesses

1. The cascade design is conceptually similar to hierarchical or coarse-to-fine generative modeling, which has appeared in many domains, such as image super-resolution diffusion and hierarchical VAEs. The novelty therefore lies mainly in its adaptation to tabular data rather than in a fundamentally new modeling principle.

2. Although the paper motivates data-dependent coupling and transport simplification, the theoretical discussion remains relatively high-level. A deeper analysis of how the cascade affects flow-matching training dynamics or convergence would strengthen the paper.

3. The method relies on constructing a low-resolution discretization of numerical features. Its performance may depend on how this discretization is implemented, yet the paper does not thoroughly analyze this sensitivity.

4. Training two models—a low-resolution model and a conditional high-resolution model—may increase training complexity compared with single-stage generators. The paper does not fully discuss the computational cost or scalability.

---

> ### Author Rebuttal · Authors · 2026-03-30
>
> We sincerely thank you for your time investment and high quality review. Below, we give answers to your questions:
>
> > How sensitive is the method to the choice of discretization scheme used to construct the low-resolution representation?
>
> We investigate two encoders in the paper: A distribution tree [DT] encoder and a Gaussian mixture model. Both are optimized towards allocating signal to Gaussian components. Hence, not any arbitrary encoder can be chosen. The mean and variance of the Gaussian components is used for the data-dependent coupling. Our results show that the method is sensitive to the choice of discretization scheme, with the DT encoder performing best. Note also that our Theorem only applies to this encoder. The overlap of Gaussian components in the GMM encoder makes a theoretical conclusion impossible. Our DT encoder is universally applicable and shows good empirical performance, as well as provable benefits for lowering the transport cost bound.
>
> > Would further hierarchical decomposition (e.g., multi-level cascades) provide additional benefits?
>
> If the goal is to model mixed-type features, then modeling a categorical resolution and a numerical resolution are enough. An additional level would increase complexity without adding any benefit. However, it is possible to, for example, further summarize the categorical data to have an even coarser structure. This would be beneficial if the categorical data were particularly difficult to model. For other data types, more levels might be beneficial. This is for instance the case, when dealing with datasets with dependent rows, e.g., due to an added longitudinal dimension. In this case, another level that models the pooled data first could be a great coarse signal for the model to later fill in the details. This type of data represents interesting future work.
>
> > How does TabCascade differ from applying existing coarse-to-fine diffusion pipelines directly to tabular data?
>
> Tabular data includes numerical and categorical feature types. Typical cascaded diffusion pipelines are usually derived with the goal of generating images and cannot be applied to tabular data. In these cases, the notion of low- versus high-resolution structure is fundamentally different from ours. Also, TabCascade necessarily requires a diffusion process that works on categorical data in its low-resolution stage. This is not the case for existing cascaded pipelines. Lastly, the idea of using the DT encoder to downsample the resolution of numerical features via discretization is not something that is done by any existing cascaded model. We will include our discussed points in a separate section in the appendix.
>
> > What is the training and inference overhead relative to TabDiff or TabDDPM?
>
> We provide training and sampling times in appendix A.10. As expected, TabCascade doubles the training time relative to CDTD but is very competitive compared to TabDDPM. TabDiff is known to be much slower to train than the other diffusion-based models. We believe that the increase in training time is worth it, considering the substantial increase in realism. Due to the generality of our model, you could choose a faster low-resolution model for categorical data to further improve training times.
>
> > The main weakness is that TabCascade introduces a strong modeling dependence on the quality of the low-resolution representation. Because the high-resolution stage is conditioned on this discretized latent representation, the framework may be sensitive to the choice of encoder and discretization strategy [...].
>
> That TabCascade depends on high quality low-resolution samples is true. This is a known drawback of all cascaded models and a consequence of the teacher forcing during training. We see no reason to believe that the DT encoder would not work on any other numerical feature or dataset. We also see no reason to make the choice of encoder a hyperparameter, as our results clearly demonstrate the strength of using the DT encoder. We prove that the denoising task for numerical features becomes simpler under the cascaded model, which holds in general for all datasets.
>
> > A secondary limitation is that the overall idea is somewhat incremental from a conceptual standpoint.
>
> We politely disagree on this point. Existing cascaded models focus on generating image data. Our work is more than a mere application of an existing model to tabular data. Vice versa, the idea of deriving low-resolution latents for numerical features the way we do is not as powerful for image generation, since the power of the encoding relies on the fact that the data matrix is fixed, such that the meaning of the columns does not change. Lastly, mixed-type features, in particular missing values, are not of concern for other data modalities. The cascaded structure required for their modeling follows naturally, but is far from an incremental application of an existing framework.

---

> > ### Author Rebuttal · Reviewer_vYYE · 2026-04-04
> >
> > Thank you to the authors for their efforts in addressing the questions. I maintain my original score

---

### Decision · Program_Chairs · 2026-04-30

**Decision:**

Accept (regular)

**Comment:**

The reviewers agree that the paper proposes an intuitive and technically sound cascaded flow matching framework for handling mixed-type tabular data. Decomposing the generation task into low-resolution and high-resolution stages effectively addresses the practical challenges of modeling features with mixed discrete and continuous states. The empirical evaluation across a diverse set of benchmarks is thorough and demonstrates clear utility improvements over existing baselines. Furthermore, the authors actively engaged with the reviewers during the rebuttal, successfully resolving critical concerns regarding numerical precision anomalies, standard error reporting, and privacy metric interpretations. Given the solid technical contribution and strong empirical validation, this paper is useful to the tabular generative modeling community and is recommended for acceptance